# Extending the Modular Earth Submodel System (MESSy v2.54) model hierarchy: The ECHAM/MESSy idealized (EMIL) model set-up

Hella Garny[1,2], Roland Walz[1], Matthias Nützel[1], and Thomas Birner[2]

[1]Deutsches Zentrum für Luft- und Raumfahrt (DLR), Institut für Physik der Atmosphäre, Oberpfaffenhofen, Germany
[2]Ludwig Maximilians University of Munich, Meteorological Institute Munich, Munich, Germany

**Correspondence:** Hella Garny (Hella.Garny@dlr.de)

**Abstract.** As models of the Earth system grow in complexity, a need emerges to connect them with simplified systems through model hierarchies in order to improve process understanding. The Modular Earth Submodel System (MESSy) was developed to incorporate chemical processes into an Earth System model. It provides an environment to allow for model configurations and set-ups of varying complexity, and as of now the hierarchy ranges from a chemical box model to a fully coupled Chemistry-Climate model. Here, we present a newly implemented dry dynamical core model set-up within the MESSy framework, denoted as ECHAM/MESSy IdeaLized (EMIL) model set-up. EMIL is developed with the aim to provide an easily accessible idealized model set-up that is consistently integrated in the MESSy model hierarchy. The implementation in MESSy further enables the utilization of diagnostic chemical tracers. The set-up is achieved by the implementation of a new submodel for relaxation of temperature and horizontal winds to given background values, which replaces all other "physics" submodels in the EMIL set-up. The submodel incorporates options to set the needed parameters (e.g., equilibrium temperature, relaxation time and damping coefficient) to functions used frequently in the past. This study consists of three parts: In the first part, test simulations with the EMIL model set-up are shown to reproduce benchmarks provided by earlier dry dynamical core studies. In the second part, the sensitivity of the coupled troposphere-stratosphere dynamics to various modifications of the set-up is studied. We find a non-linear response of the polar vortex strength to the prescribed meridional temperature gradient in the extratropical stratosphere, that is indicative of a regime transition. In agreement with earlier studies, we find that the tropospheric jet moves poleward in response to the increase in the polar vortex strength, but at a rate that strongly depends on the specifics of the set-up. When replacing the idealized topography to generate planetary waves by mid-tropospheric wave-like heating, the response of the tropospheric jet to changes in the polar vortex is strongly damped in the free troposphere. However, near the surface, the jet shifts poleward at a higher rate than in the topographically forced simulations. Those results indicate that the wave-like heating might have to be used with care when studying troposphere-stratosphere coupling. In the third part, examples for possible applications of the model system are presented. The first example are simulations with simplified chemistry to study the impact of dynamical variability and idealized changes on tracer transport, and the second example are simulations of idealized monsoon circulations forced by localized heating. The ability to incorporate passive and chemically active tracers in the EMIL set-up demonstrates the potential for future studies of tracer transport in the idealized dynamical model.

# 1 Introduction

Earth system models continue to incorporate more processes to enable a more complete simulation of the climate system, and thus produce the best possible climate projections. In practice, this increases the complexity of model codes as new compartments are added to represent new processes and interactions. However, with models gaining more and more complexity, it becomes difficult to isolate and understand the role of individual processes. This "gap between simulation and understanding in climate modeling" was pointed out by Held (2005), and it was suggested that the way forward is to work with a hierarchy of models with reduced to full complexity. Two recent overview papers (Jeevanjee et al., 2017; Maher et al., 2019) give surveys of current concepts and activities in building hierarchical model systems.

The basic concept in constructing a simplified model is to include only those processes, that are (absolutely) relevant for the question to be addressed. Thereby, the behavior of those processes can be isolated in an idealized environment, and the interaction of the limited number of processes chosen can be investigated.

A frequently used idealized model set-up for studying global large-scale dynamics is the dry dynamical core model proposed by Held and Suarez (1994, HS94 hereafter). While originally developed and used for testing dynamical cores of atmospheric models, the elegance of the model makes it an ideal tool for dynamical process studies, and it is widely used for this purpose (see Maher et al., 2019, for a review of applications). This "Held-Suarez"-type model uses the full dynamical core of a general circulation model (GCM), but replaces all thermodynamical processes (e.g., radiation, convection) by relaxation towards a prescribed equilibrium temperature and the surface boundary layer by relaxation of near-surface winds towards zero (as described in detail in Sec. 2). Thus, with this model set-up the thermodynamic forcing of the atmosphere can be easily modified and the response of the large-scale circulation to those isolated modifications can be studied. Examples are changes in equilibrium meridional temperature gradient or thermal damping time scale (Gerber and Vallis, 2007), or changes in surface friction (Chen et al., 2007).

The functions for the equilibrium temperature and relaxation coefficients suggested in HS94 are widely used, and the HS94 model set-up was extended to study the dynamics of the stratosphere-troposphere system by modifying the equilibrium temperature of the stratosphere (Polvani and Kushner, 2002, PK02 hereafter) and later by adding topography to include planetary wave generation that is essential for the stratospheric circulation (Gerber and Polvani, 2009). This model set-up was used among others to study stratosphere-troposphere coupling (Gerber and Polvani, 2009), the structure of the Brewer-Dobson circulation (Gerber, 2012), and the circulation's response to idealized heating resembling the thermal response to greenhouse forcing (e.g., Butler et al., 2010; Wang et al., 2012). Recently, it was suggested that the forcing of the planetary waves relevant for stratospheric dynamics can also be achieved by inserting diabatic heating in the mid- to upper troposphere (Lindgren et al., 2018), which leads to a similar climatology as the topographically forced simulations, but to changes in the sudden stratospheric warming properties.

While the dry dynamical core model has proven useful in advancing our understanding of the dynamical response to given thermodynamic forcing, the application of the model hinges on a realistic representation of the Earth's atmosphere's behavior of the modeled dynamics. Gerber and Polvani (2009) and Chan and Plumb (2009) showed that the strong response of the

surface jet location to stratospheric polar vortex changes found in the original study by PK02 resulted from a regime shift of the tropospheric jet. With a changed set-up, e.g., by including topography (Gerber and Polvani, 2009), or with enhanced meridional temperature gradients in the winter hemisphere (Chan and Plumb, 2009), the regime-like behavior of the jet location is suppressed, and thus the response of the jet location to stratospheric polar vortex changes is damped strongly. However, the

regime shift can re-emerge for experiments with strong additional forcing (e.g., tropical heating, as shown by Wang et al., 2012). Overall, those results indicate that the dynamical response to a given forcing is highly (non-linearly) dependent on the basic state of the model. Whether this sensitivity to the basic state due to dynamical regimes is relevant for the real atmosphere will have to be evaluated with care. If the regime behavior proves to be an artifact of the idealized models, this would impede its application to advance the understanding of dynamical processes of the real atmosphere.

Beyond the purely dry dynamical core models, which are useful to study aspects of the global circulation, a question that motivates the expansion to another level of complexity, is the interaction of moisture with large-scale dynamics, either by latent heat release or by its role as greenhouse gas. Frierson et al. (2006) expanded the dry dynamical core ("Held-Suarez") model by adding moisture and convection with latent heat release to the model, including simplified (gray) radiation that is insensitive to water vapor, thus tackling the question of the role of latent heat release for large scale dynamics. This model set-up has

been extended by including the radiative effects of ozone in an idealized manner, resulting in a more realistic simulation of stratospheric dynamics (Davis and Birner, 2019). In a step further, the role of water vapor as radiatively active gas is included by using more comprehensive radiation schemes, as done by e.g., Merlis et al. (2013); Jucker and Gerber (2017); Tan et al. (2019). In those set-ups, treatment of radiatively relevant fields as clouds, ozone and aerosol forcing is mostly based on simple assumptions such as constant values.

As stated above, the nature of the hierarchy that is to be constructed depends on the scientific question at hand. Our aim is to study the large-scale dynamical variability of the stratosphere-troposphere system and its response to idealized forcings, and in particular the impact of dynamical variability and forced changes on the transport of passive and chemically active trace gases. The latter is motivated by a variety of research questions on the distribution of trace species in the atmosphere, for example on how changes in the circulation in a changing climate will affect stratospheric ozone. This question has received

a lot of attention recently in the light of observed lower stratospheric ozone trends that are not fully understood (Ball et al., 2018). Another question we aim to tackle with the idealized model is the efficiency of troposphere-stratosphere transport in monsoonal circulation systems via different pathways. The idealized set-up allows to study the role of different transport pathways depending on the details of the forcing of the circulation system. To enable those studies, a well suited model set-up is a dry dynamical core model with the utilities for tracer transport and the possibility to include chosen chemical reactions

(simplified to the needs of the user). Therefore, we implement such a model set-up within the Modular Earth Submodel System (MESSy, Jöckel et al., 2005) framework, which provides the needed utilities in a modular manner.

Several initiatives are aiming to build modeling frameworks with set-ups of varying complexity within the same model system (Vallis et al., 2018; Polvani et al., 2017), an approach that will advance both the usability of idealized models as well as the connectedness of the simple and the more complex model set-ups. In the same spirit, the MESSy framework was developed ex-

plicitly with the goal to provide "a framework for a standardized, bottom-up implementation of Earth System Models (or parts

of those) with flexible complexity" (see https://www.messy-interface.org/). The motivation of the MESSy framework was originally to incorporate chemical processes of varying complexity into an Earth System model. The MESSy framework couples a base model (dynamical core) to submodels, that contain the physical parametrizations as well as diagnostics. Among other base models, the ECHAM dynamical core is available in MESSy. The MESSy framework includes model configurations rang-

ing from a 0-dimensional box model of atmospheric chemistry (Sander et al., 2019) to the complex chemistry-climate model ECHAM/MESSy Atmsopheric Chemistry (EMAC), coupled to a deep ocean model (Jöckel et al., 2016). An illustration of a selection of available model complexities is shown in Fig. 1, as function of the complexity in physical processes/ compartments included (horizontal axis) and of the complexity of atmospheric chemical processes included (vertical axis). The lowest complexity on the chemical axis are prescribed concentrations for radiatively active species (e.g., ozone), followed by a simplified

parametrization to include effects of methane oxidation on stratospheric water vapor. The chemistry module MECCA (Module Efficiently Calculating the Chemistry of the Atmosphere, Sander et al., 2019) contains a large set of reactions relevant in the troposphere and stratosphere, but it can be configured to the user's needs by choosing any subset of reactions, thus allowing for simplified to very comprehensive chemical set-ups. The chemical calculations can be performed as a box model (denoted CAABA, see Sander et al., 2019), or within a full general circulation model either without feedback between dynamics and

chemistry (the so-called "Quasi Chemistry-Transport Model" (QCTM), see Deckert et al., 2011) or with feedback, i.e., as full chemistry-climate model (Jöckel et al., 2006; Jöckel et al., 2010, 2016). Besides the prescribed sea surface temperature set-up, a mixed-layer ocean (Dietmüller et al., 2014) or a full ocean model (Pozzer et al., 2011) can be used.

One advantage of the MESSy framework is its modular nature, i.e., individual processes are implemented as independent submodels that can be easily exchanged or complemented by new processes, and each submodel can be easily switched on or

off (by namelist choice). Therewith, the hurdle of code modifications to build a model tailored to the necessary complexity is rather low. Moreover, the design of the model system allows the creation of model hierarchies in which the same code can be used in a simple model set-up as well as in the full Earth-System model. Any developments in model components can be transferred easily up- and downward in the model hierarchy.

**Aims and structure of paper**

The aim of this paper is three-fold: Firstly, it serves the documentation of the model and its performance (Sec. 2 and 3). Secondly, we study the sensitivity of the simulated troposphere-stratosphere dynamics to the model set-up (Sec. 4), and thirdly we present application examples that serve to highlight the capabilities with the EMIL implementation (Sec. 5). These three parts are stand-alone sections, and the reader may choose to focus on the section of her/his interest.

In the first part (Sec. 2 and 3), we document the implementation of the dynamical core set-up within the MESSy frame-

work and its performance. While "Held-Suarez" test simulations with the same dynamical core (ECHAM) were previously performed to study the resolution sensitivity of the model core (Wan et al., 2008), the here presented implementation is new in that it is part of the MESSy framework. The implementation within MESSy ensures an easily accessible idealized model set-up that is consistently integrated in the MESSy model hierarchy, and that enables the use of all tracer utilities, including the utilization of diagnostic chemical tracers. The implementation is achieved by adding a simple submodel for Newtonian cooling

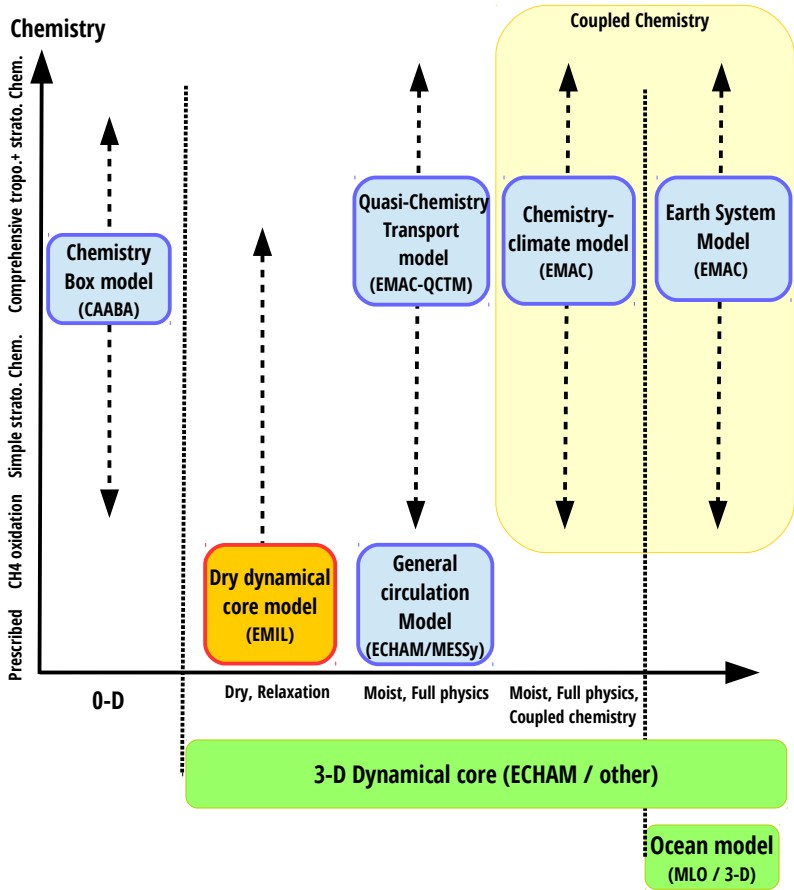

**Figure 1.** Schematic of the MESSy model hierarchy with existing (blue) model set-ups and the model set-up described in this paper (red). Model set-ups are displayed as function of their complexity in dynamics/physics/compartments (horizontal axis) versus complexity in chemical mechanism (vertical axis). The horizontal axis ranges from (left) a 0-dimensional box model to (middle) models with an atmospheric dynamical core (ECHAM or other implemented dynamical cores in MESSy), but with varying physical complexity, and to (right) models with an additional ocean model (Full 3-d or mixed-layer ocean). The vertical axis displays the chemical complexity, that can gradually be increased from prescribed tracer concentrations for the radiation scheme to a more and more comprehensive set of chemical reactions. The chemistry can be used diagnostically only, or in a coupled manner (yellow box).

and Rayleigh friction, that replaces the complex physics (see Sec. 2 and the Supplement for technical details including a user manual). We present standard test cases with forcings given by Held and Suarez (1994) and its stratospheric extension (Polvani and Kushner, 2002) in Sec. 3.

In the second part (Sec. 4), we study the sensitivity of the simulated troposphere-stratosphere dynamics with respect to several modifications to previously used set-ups. Most importantly, this includes modifications of the equilibrium temperatures in the tropical troposphere and in the winter high latitudes (Sec. 4.2). The latter leads to more realistic temperature profiles in the lower stratosphere. We further test the sensitivity of the simulated dynamics to the generation of large-scale waves by zonally asymmetric heating instead of idealized topography, as suggested recently by Lindgren et al. (2018). We end the section with a discussion on the different states of the tropospheric jet and of the stratospheric polar vortex, as well as their relation, in the suite of different sensitivity experiments (Sec. 4.4).

Finally, in the third part (Sec. 5), we present two application examples of the model: first, we present simulations including a small set of chemical reactions (namely photolysis of Chlorofluorocarbons) and demonstrate the potential of the model to study the role of dynamical variability and idealized changes on tracer transport (Sec. 5.1). Secondly, the simulation of an upper tropospheric anticyclone forced by simple, localized constrained heating that resembles the Asian monsoon anticyclone is presented in Sec. 5.2.

## 2   Model description

The ECHAM/MESSy IdeaLized (EMIL) model set-up is based on MESSy version 2.54 (Jöckel et al., 2006; Jöckel et al., 2010, 2016), and will be available for users in the next release, i.e.,version 2.55. In the idealized, "Held-Suarez"-type, model set-up, all physics (radiation, clouds, convection and surface processes) are switched off, and are replaced by the newly implemented submodel "RELAX", that relaxes the variables temperature and horizontal winds to given background values. The submodel RELAX is described in the next subsection, and we provide technical details of the model set-up (namelist choices etc.) and implementation in the Supplement.

### 2.1   The submodel RELAX

The submodel RELAX calculates

1. Newtonian cooling, i.e., temperature relaxation towards a given equilibrium temperature with a given relaxation time scale

2. Rayleigh friction, i.e., horizontal wind relaxation towards zero with a given damping coefficient

3. additional diabatic heating over selected regions

The three processes are switched on/off via namelist parameters, as described in the Supplement. The submodel is called from the physics routine *physc* through *messy_physc*. The full call tree including all subroutines is provided in the Supplement. In the following, the implemented options in the routines are described, with the full equations given in Appendix A.

**Newtonian cooling**

The temperature tendency calculated by Newtonian cooling is given by $\delta T/\delta t = -\kappa(T - T_{eq})$, where $\kappa$ is the inverse relaxation time scale, $T$ the actual temperature calculated by the model, and $T_{eq}$ the prescribed equilibrium temperature. The inverse relaxation time scale and the equilibrium temperature have to be specified in the model set-up, either by setting them to fields imported from an external file, or by setting them to values given by pre-implemented functions. Currently, the implemented functions for the inverse relaxation time scale and the equilibrium temperature are firstly those given by HS94 (option 'HS', see Eq. A1), but with the possibility to include hemispheric asymmetry, and, secondly, those given by PK02 (option 'PK', see Eq. A4), but with the following extension: we include the possibility to vary the transition pressure between tropospheric and stratospheric temperature from summer to winter hemisphere. This latitudinal variation is implemented by using the same weighting function as is used for the transition to the polar vortex equilibrium temperatures (see Eqs. A5 and A6). The transition pressure in the remaining area is held constant at $p_{Ts} = 100\,\mathrm{hPa}$, as in the original 'PK' set-up. Fig. 2 shows an example of the equilibrium temperature with modified winter transition pressure, here for $p_{Tw} = 400\,\mathrm{hPa}$ and $\gamma = 2\,\mathrm{K\,km^{-1}}$. In section 4.2.2, sensitivity simulations with respect to variations in the transition pressure over winter high latitudes ($p_{Tw}$) are presented.

Additionally, we performed simulations with a modified prescribed tropospheric equilibrium temperature, differing in the strength of the tropical tropospheric vertical temperature gradient. In this set-up of $T_{eq}$, in the formulation for the tropospheric equilibrium temperature, the term that reduces the vertical temperature gradient in the tropics (see Eq. A1, 4th term) was inadvertently implemented with a logarithm with base 10 instead of the natural logarithm. As a result, the control parameter of the tropical vertical temperature gradient $\delta_z = 10\,\mathrm{K}$ is reduced by a factor of $1/\log(10) \approx 0.43$, so that in the sensitivity experiments $\delta_z = 0.43 \times 10\,\mathrm{K} = 4.3\,\mathrm{K}$. The resulting difference in the equilibrium temperature, as displayed in Fig. 2 (right), maximizes at around 5.5 K in the tropical upper troposphere. The simulated temperatures in the same region are about 3.5 K lower. While the set of simulations with this modified set-up of $T_{eq}$ was produced inadvertently by an implementation oversight, they provide an interesting sensitivity to the original PK set-up, and will thus be used in this study to test the sensitivity of the tropospheric jet response to forced polar vortex changes (see Sec. 4.2).

**Rayleigh friction**

Horizontal winds are relaxed to zero (i.e., damped) with a given damping coefficient $k_{damp}$ by $\delta\boldsymbol{v}/\delta t = -k_{damp}\boldsymbol{v}$. As for the Newtonian cooling, the damping coefficient can be selected via the namelist with the same options. The implemented functions that can be chosen are:

1. the surface layer damping as specified by HS94 (option *'HS'*, see Eq. A7)

2. the damping of a layer at the model top as specified by PK02 (option *'PK'*, see Eq. A8)

3. a newly introduced option for damping of a layer at the model top that follows the function as implemented in the original ECHAM code (option *'EH'*, see Eq. A9).

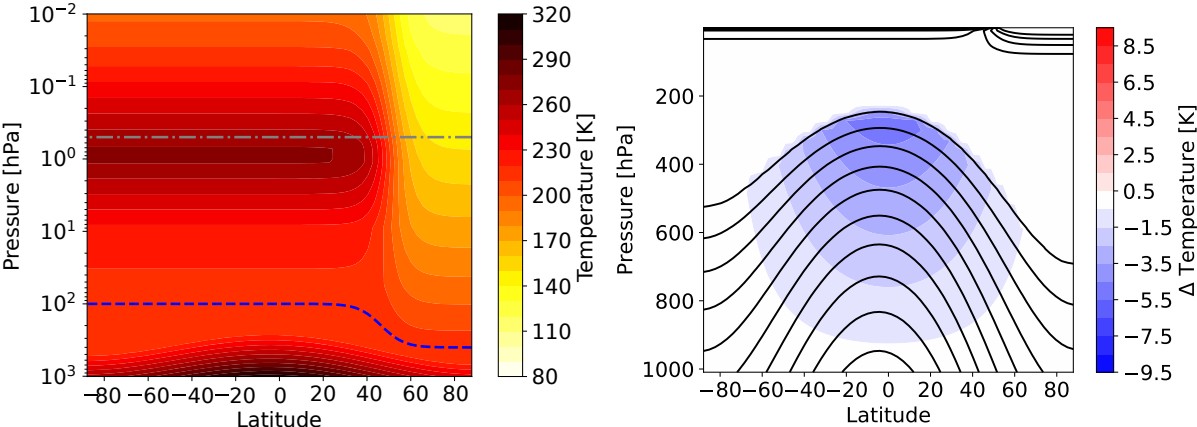

**Figure 2.** Left: Equilibrium temperature (in K) for $p_{Tw} = 400\,\text{hPa}$ and $\gamma = 2\,\text{K}\,\text{km}^{-1}$ together with the transition pressure $p_T(\phi)$ (blue dashed line) and the pressure above which damping sets in (gray dashed-dotted line). Right: $T_{eq}$ as given by the original PK02 implementation (black contours) and difference in $T_{eq}$ between simulations with reduced $\delta_z$=4.3 K and with standard $\delta_z$=10 K (colors).

For the *'EH'* option, the drag is enhanced by a given factor for each level going upward. Sensitivity simulations with respect to the newly implemented form of the upper level damping are presented in Section 4.1. Note that as damping at the model surface (option 1) and at the upper layers (options 2 or 3) are complementary, more than one option can be chosen, in which case the profiles of the damping coefficients are added.

## 5 Diabatic heating routines

Next to the zonally symmetric temperature tendency calculated by Newtonian cooling, additional temperature tendencies (diabatic heating and cooling) can be added to mimic particular thermodynamic forcings in the atmosphere. Currently, three options are implemented:

1. A function for zonal mean heating (*tteh_cc_tropics*), that allows the user to apply a temperature tendency with a Gaussian shape in latitude and pressure, as detailed in Eq. A10. This zonal mean heating can be applied to mimic greenhouse gas induced temperature changes, as has been done by e.g., Butler et al. (2010).

2. A wave-like heating varying with longitude (*tteh_waves*), that can be used for diabatic generation of planetary waves, as introduced by Lindgren et al. (2018) and detailed in Eq. A11.

3. A function for localized heating (*tteh_mons*), that can be applied to simulate monsoon-like circulation systems, as previously by done by e.g., Siu and Bowman (2019). The formulation for the localized heating is given by Eqs. A12 to A16.

## 3 Model benchmark tests

In this section, results obtained with the EMIL set-up are compared to results of earlier studies with identical set-ups (both with the Held-Suarez set up, Sec. 3.1 and the Polvani-Kushner set-up, Sec. 3.2) to test whether the EMIL implementation is able to reproduce the results of those earlier studies.

The benchmark simulations presented in this section are performed for 1825 days for the Held-Suarez forcing, and for 10950 days for the Polvani-Kushner forcing (see Figure captions and Table B1 for details).

### 3.1 Held-Suarez forcing

"Held-Suarez" test simulations with the same dynamical core (ECHAM) as used in EMIL were previously performed (Wan et al., 2008). We ran a simulation with identical set-up and resolution to test whether our implementation of the Held-Suarez

set-up with the same base model can reproduce the results of Wan et al. (2008). The resolution consists of a spectral horizontal resolution of T63 and 19 vertical levels extending up to 10 hPa, denoted as T63L19. As shown in Fig. 3 the climatologies of zonal wind, temperature and eddy fluxes are closely reproduced when compared to Fig. 1 of Wan et al. (2008). In both model set-ups, the wind jet maxima are around 30 ms$^{-1}$, the eddy heat flux maxima around 20 K m s$^{-1}$, and the eddy momentum flux maxima around 70 m$^2$s$^{-2}$. Furthermore, the EMIL temperature and wind climatologies also compare well to the simulations

shown in the original HS94 study (see their Fig. 1 and 2).

    In the remainder of the paper, a vertical resolution with higher top (0.01 hPa) and with 90 levels (L90MA, where MA stand for Middle Atmosphere) will be used together with T42 as spectral resolution (one of the standard resolutions of EMAC, see Jöckel et al., 2016). The differences in the climatologies between the T42L90MA and the T63L19 simulation (for the HS set-up) are shown in Fig. 3 (bottom). The jets are shifted equatorward with T42L90MA resolution, and the eddy variance is

generally reduced. This is likely a combined effect of lower horizontal and higher vertical resolution, consistent with the results by Wan et al. (2008): They reported an equatorward shift of the jet and reduced eddy variance both with decreasing horizontal resolution, as well as with increasing vertical resolution. However, in our experiments not only the vertical resolution, but also the model top is increased in the L90MA experiment, and the latter might play a role here. The issue of resolution sensitivity will not be touched further as it is not the subject of this paper, but it should be kept in mind that the results might in general

be dependent on the chosen resolution.

### 3.2 Polvani-Kushner set-up

In the study by PK02, an equilibrium temperature is introduced that enables the simulation of an active stratosphere with a polar vortex in the winter hemisphere.

    As a test case, EMIL simulations are performed with identical forcing as in PK02, i.e., with the same choice of the prescribed

equilibrium temperature and the damping layer at the top of the model. The results for simulations with the polar temperature lapse rate $\gamma$ set to 4 K km$^{-1}$ are shown in Fig. 4 (left). The polar vortex strength maximizes at around 90 ms$^{-1}$ for $\gamma$=4 K km$^{-1}$, and at 30 ms$^{-1}$ for $\gamma$=1 K km$^{-1}$ (not shown), similar to the wind maxima shown in PK02. Also the structure of the polar vortex,

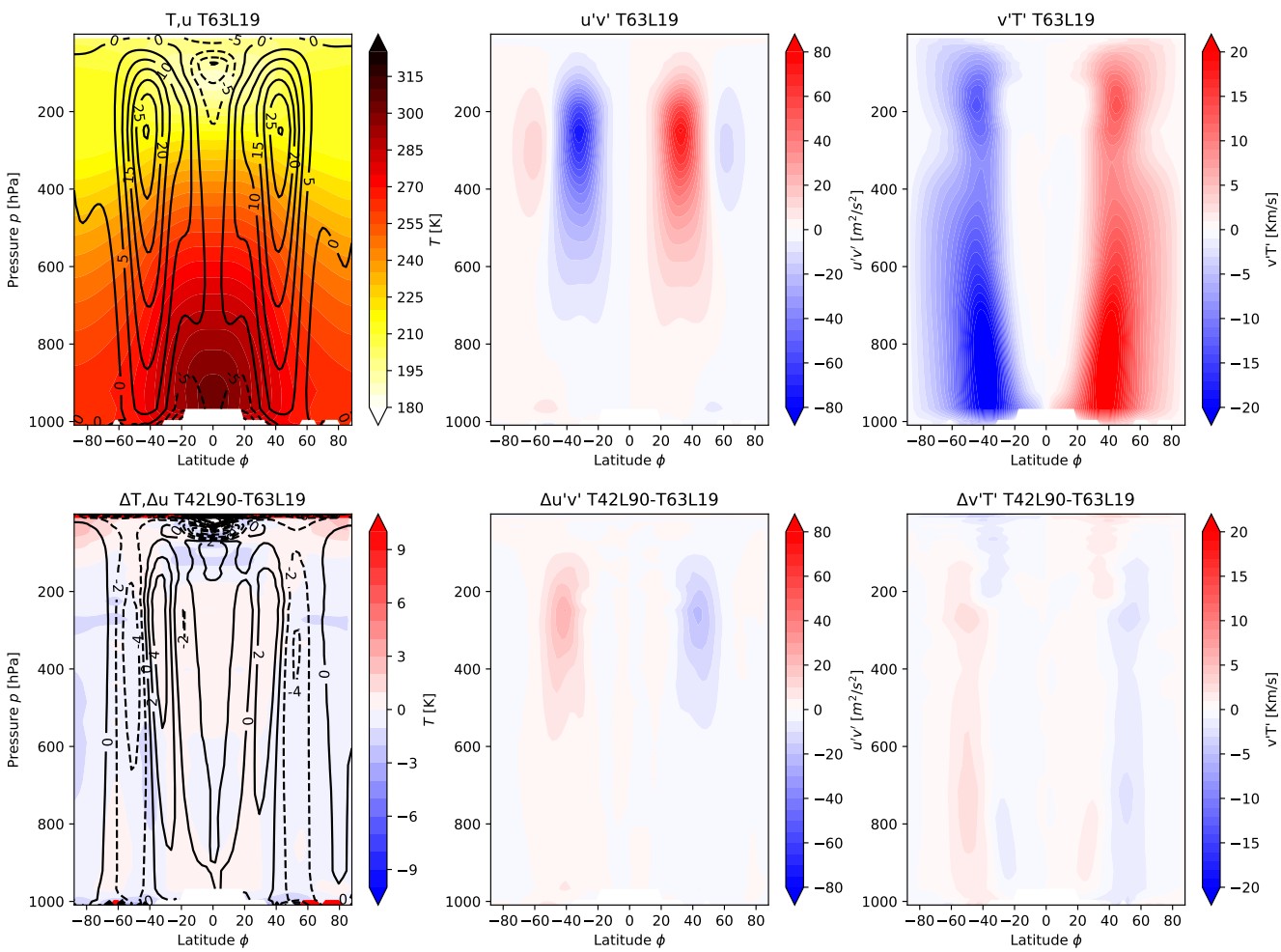

**Figure 3.** Top: Results from a HS simulation at T63L19 resolution, showing mean temperature [K] and zonal mean zonal wind [m s$^{-1}$, contour interval 5 m s$^{-1}$] (left), mean eddy momentum fluxes [m$^2$ s$^{-2}$] (middle) and mean eddy heat fluxes [K m s$^{-1}$] (right) averaged over 1500 days (after spin-up of 325 days). Bottom: As above, but difference of a simulation at T42L90MA resolution minus the T63L19 simulation (with wind contour interval 2 m s$^{-1}$).

and the subtropical jets agree well between our simulation and the ones presented by PK02. Based on the same model as used by PK02 (namely GFDL's spectral dynamical core), Jucker et al. (2013) show climatologies of wind and temperature for the PK02 set-up with $\gamma$=4 K km$^{-1}$. The temperature climatology of the EMIL simulation with $\gamma$=4 K km$^{-1}$ agrees well with the one shown by Jucker et al. (2013), with both models simulating a tropical lower stratospheric temperature minimum of 210 K

5   and a pronounced minimum in temperature (T < 180 K) at the winter pole around 10 hPa.

For a second test case, we include the generation of planetary waves by an idealized topography, as proposed by Gerber and Polvani (2009). Fig. 4 (right) shows the simulated climatologies with a wavenumber 2 (WN2) mountain with amplitude h =

km and $\gamma$=4 K km$^{-1}$. Following Gerber and Polvani (2009), the mountain is centered at 45°N and falls off to zero at 25°N and 65°N (see their Eq. 1). This set-up of the mountain was found to lead to most realistic simulation of the mean state of the polar vortex and its variability by Gerber and Polvani (2009). The resulting climatologies of zonal wind, with a polar vortex strength of about 50 ms$^{-1}$, and of temperature, with a minimum temperature over the winter pole at 10 hPa of around 180 K

again closely reproduce the results by Gerber and Polvani (2009) and the equivalent simulation shown by Jucker et al. (2013).

The variability of the polar vortex is diagnosed by the time-series of the zonal mean zonal wind at 10 hPa and 60N for the PK simulation with $\gamma$=4 K km$^{-1}$ and a WN2 mountain amplitude of h=3 km in Fig. 5 (black line). The polar vortex is highly variable with winds between -10 to 60 ms$^{-1}$, with sudden decreases in the wind speeds, known as sudden stratospheric warmings. The time series of the EMIL simulation presented here closely resembles that shown by Gerber and Polvani (2009)

in terms of variability.

In the study by PK02 it was shown that an increased polar vortex strength, forced by an enhanced stratospheric meridional temperature gradient (i.e., via parameter $\gamma$), induces a poleward shift of the tropospheric jet. The tropospheric jet response to stratospheric polar vortex changes will further be discussed in Sec. 4, but it is noted here that EMIL model simulations with the same set-up as in the PK02 study reproduce the behavior of the poleward shift of the tropospheric jet with increasing polar

vortex strength (see Fig. 8, left, yellow solid line).

Overall, the results of this section show that the EMIL set-up is able to reproduce earlier results of simulations performed with dynamical core models under identical set-up of equilibrium temperature, relaxation time, the damping layer and topographically generated planetary waves.

## 4  Sensitivity of coupled troposphere-stratosphere dynamics to modified set-ups

In this Section, the response of the simulated troposphere-stratosphere dynamics to three different types of modifications are studied: (1) modifications of the shape of the upper atmospheric sponge layer, see Sec. 4.1, (2) modifications of the equilibrium temperature, namely of the tropical tropospheric vertical temperature gradient (Sec. 4.2.1) and of the winter high-latitude equilibrium temperature profile (Sec. 4.2.2), and (3) planetary wave generation by wave-like heating instead of topography (Sec. 4.3). The section concludes with a discussion of the sensitivities of the stratospheric polar vortex and the tropospheric jet

to the different kinds of modifications.

The simulations presented here are performed for at least 1825 days, and a number of simulations are extended up to 10950 days. The simulation length is specified for each simulation in Table B1 and in the figure captions. To reduce the uncertainty of the results, it would be favorable to extend each simulation until convergence of the climatologies is reached. In particular for climate states with multiple dynamical regimes, this would, however, require very long integration times.

To reduce computational and data storage costs, we used the strategy of variable simulation length, i.e., we extended only a chosen set of simulations to test for the robustness of the results. In the shorter simulations, considerable uncertainty in the climatologies due to variability can be present. However, as shown in the following, the results are qualitatively robust when comparing the short and long simulations. Details on the simulation set-up and integration length can be found in Table B1.

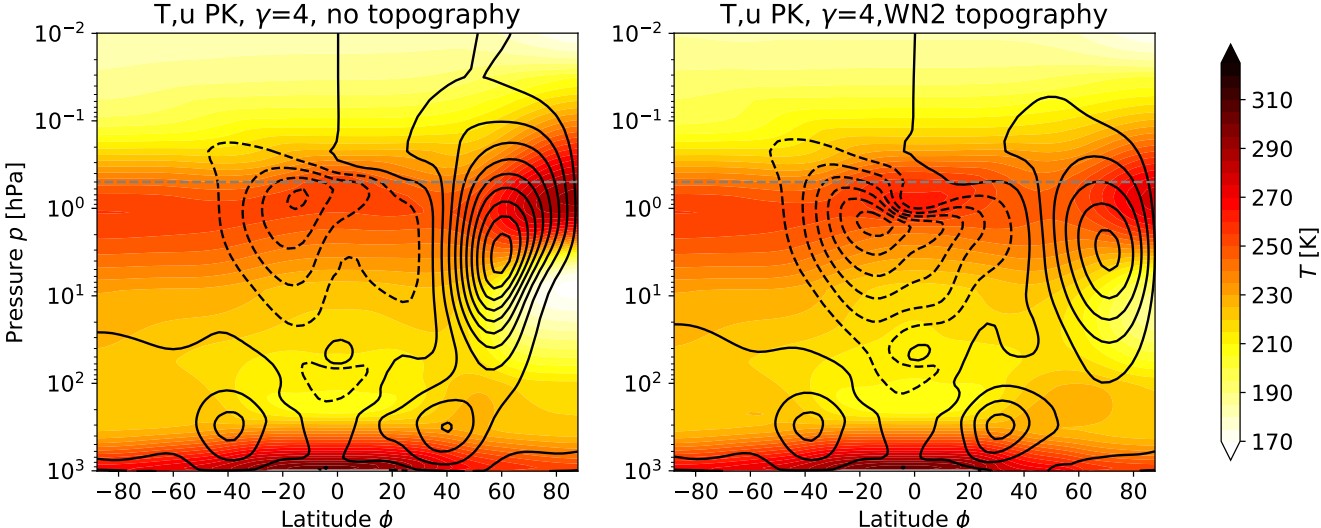

**Figure 4.** Climatologies of wind (black contours, contour interval 10 ms$^{-1}$, solid positive, dashed negative) and temperature [K] (colored contours) of (left) an EMIL simulation with the PK02 set-up with $\gamma$=4 K km$^{-1}$, and (right) an EMIL simulation with the PK02 set-up with $\gamma$=4 K km$^{-1}$ and with WN2 topography with h=3 km. The gray dashed horizontal lines in the EMIL climatologies mark the lower boundary of the damping layer. Averages are performed over 10000 days.

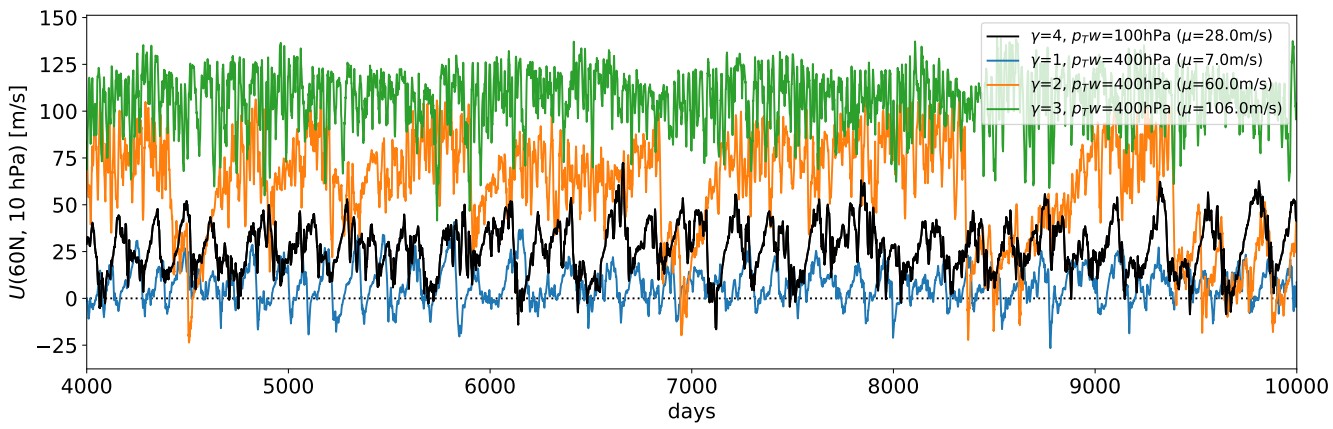

**Figure 5.** Timeseries of zonal mean zonal wind [ms$^{-1}$] at 10 hPa and 60°N for different configurations of the PK set-up with a WN2 topography with h=3 km. The black line displays the reference simulation with $\gamma$ =4 K km$^{-1}$ as in Fig. 4 (right). The colored lines display the sensitivity simulations discussed in Sec. 4.2, with lowered winter transition pressure $p_{Tw}$ and for $\gamma$ =1, 2 and 3 K km$^{-1}$ (see legend). In the legend, the average windspeed at this location over the whole simulation is given (denoted $\mu$).

## 4.1 Sensitivity to the shape of the upper atmospheric damping layer

The damping layer at levels above 0.5 hPa is included to account for the strong damping of winds that in the real atmosphere (or the full model) is due to drag by breaking gravity waves (GW). The simplified manner of damping the entire horizontal wind fields introduces a non-physical sink of momentum, as not only the zonal mean wind, but also all waves are damped. When analyzing results obtained with the model, this has to be kept in mind.

The damping layer as introduced by PK02 uses a damping coefficient that increases quadratically with decreasing pressure. The profile of the PK02 damping coefficient is shown in Fig. 6 together with the profile of zonal mean zonal wind tendencies due to parametrized gravity waves divided by the zonal mean wind (averaged over 40-60°N/S) from a model simulation with the full atmospheric EMAC set-up, i.e., an equivalent damping coefficient of the zonal mean wind by the parametrized GW drag. The "damping" by GW drag varies between years and hemispheres, but generally increases exponentially with decreasing pressure, not quadratically. Therefore, we argue that a damping coefficient with exponential increase mimics the net effects of parametrized GW drag better.

A sensitivity simulation is performed in which the damping coefficient in the upper model domain follows the exponential function given by Eq. (A9) (option *EH*; this is the shape of the "sponge" layer originally implemented in the ECHAM model). The damping coefficient of this sensitivity simulation is shown in Fig. 6 as red line.

The simulated climate states with the two different set-ups naturally differ within the sponge layer, with maximum differences in zonal winds of $30\,\mathrm{m\,s^{-1}}$ around $0.5\,\mathrm{hPa}$ (see Fig. 7). Considerable differences in wind and temperature also extend below the damping layer, in particular at high latitudes, but also in the tropics. In the tropics, alternating jets form, reminiscent of the Quasi-Biennial Oscillation, but with very long periods (about 5 years). As the tropical winds have thus not converged for the given simulation length, and differences are not significant, we will not regard them here.

Differences are mostly insignificant below $10\,\mathrm{hPa}$, however small (significant) differences in zonal winds of $2\,\mathrm{m\,s^{-1}}$ extend down into the troposphere. The increase in high-latitude zonal winds, which maximizes at the lower bound of the damping layer, is accompanied by an upward shift of the temperature maximum at the stratopause. This brings the temperature maximum closer to realistic values, as evident from the comparison to the "SPARC" climatology (Randel et al., 2004; SPARC, 2002) in the right panel of Fig. 7.

Since the *EH* sponge is weaker, the increase in zonal mean winds within the damping layer can be expected. The weaker sponge and changed zonal wind structure modifies planetary wave propagation (stronger upward propagation between from about $3\,\mathrm{hPa}$ upward, not shown), thus influencing the mean climate also below the damping layer (decreased wave driving, leading to stronger zonal winds and lower polar temperatures). The effect of the modified damping coefficients is similar in simulations with WN2 topography (albeit with weaker absolute differences, not shown).

As the exponentially increasing damping coefficient (*EH*) resembles the vertical structure of GW drag, and since for both a flat surface and idealized topography, the height at which the polar winter temperature profile reaches its maximum is more realistic in case of the *EH* damping layer (see Fig. 7 right), we chose to use the exponentially increasing damping coefficient (*EH*) in the following as our reference set-up.

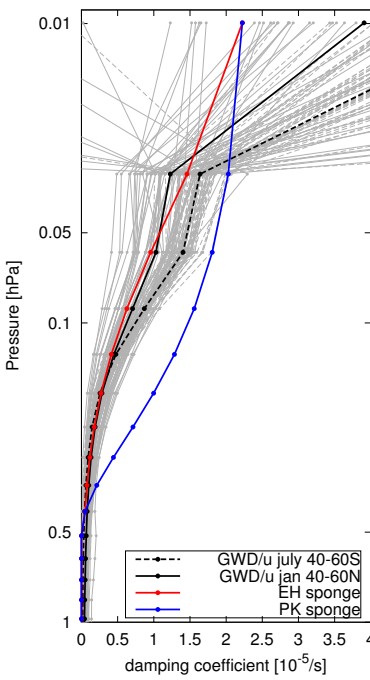

**Figure 6.** Damping coefficient $[s^{-1}]$ of the sponge layer in the "EH" (red) and in the PK02 set-up (blue) together with the effective damping time scale of the zonal winds by gravity wave drag (i.e., -GWD/u) from an ECHAM simulation with a non-orographic and orographic GW scheme, averaged over 40-60°S in July 1960 to 2010 (gray dashed, average over all years shown as black dashed) and averaged over 40-60°N in January 1960 to 2010 (gray solid, average over all years shown as black solid).

## 4.2 Sensitivity to modification of the equilibrium temperature

In addition to the benchmark simulations with identical set-up as in PK02, we performed sensitivity simulations with modified prescribed equilibrium temperature, with an altered tropospheric vertical temperature gradient (Sec. 4.2.1) and with reduced winter high-latitude lower stratospheric temperatures (Sec. 4.2.2).

### 4.2.1 Sensitivity to reduced tropical tropospheric vertical temperature gradient

In the standard HS94 set-up, the vertical temperature gradient in the (sub-)tropical troposphere is reduced to mimic the effects of latent heat release. This reduction is controlled by the parameter $\delta_z$ in the 4th term of Equ. A1, and in the sensitivity simulations presented here, $\delta_z$ was reduced from 10 K to 4.3 K, as described in Sec. 2 (see also Fig. 2, right for the resulting difference in $T_{\text{eq}}$). These simulations with reduced static stability in the (sub-)tropics resulted from an implementation oversight, but proved to provide an interesting sensitivity to the original PK02 set-up.

EMIL model simulations with identical set-up as in PK02 reproduce the strong poleward shift of the tropospheric jet in response to a forced increase in the polar vortex strength via the prescribed stratospheric meridional temperature gradient (i.e.,

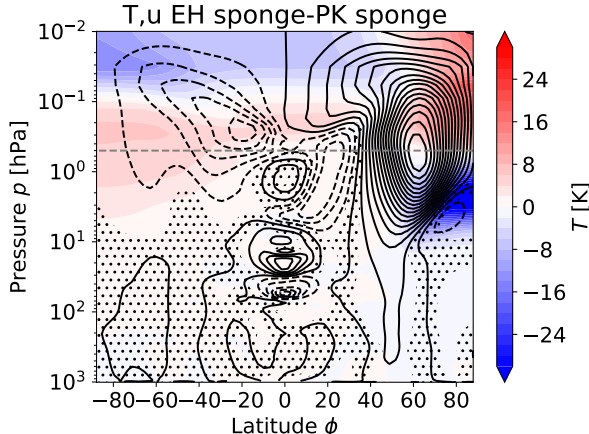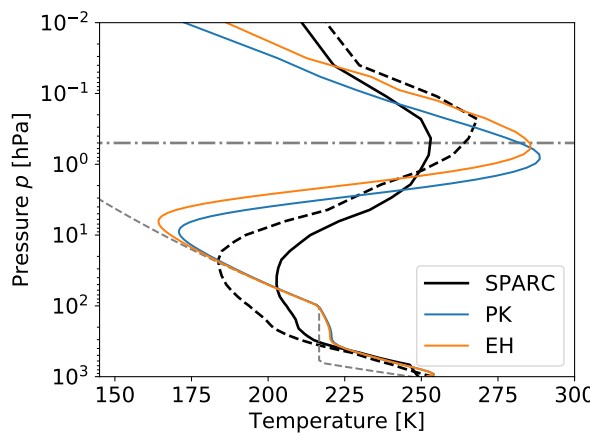

**Figure 7.** Left: Differences in zonal mean temperature (contour interval is $2\,\mathrm{K}$) and zonal mean zonal wind (contour interval is $2\,\mathrm{m\,s^{-1}}$) between a model simulation with exponentially increasing damping coefficient (*EH*) and a model simulation with quadratically increasing damping coefficient (*PK*) for $p_{\mathrm{Tw}} = 100\,\mathrm{hPa}$ and $\gamma = 4\,\mathrm{K\,km^{-1}}$ without topography. Averages are performed over 1500 days of simulation. Stippling indicates non-significant wind differences (on a 95% level, based on a T-test performed on 30-day means). Right: Polar winter temperature profiles of same model simulations averaged from $70°\mathrm{N}$ to $90°\mathrm{N}$, with temperature profiles from the SPARC climatology in northern hemispheric winter conditions (black solid line) and southern hemispheric winter conditions (black dashed line) as well as the equilibrium temperature profile (gray dashed line). The dash-dotted line marks the lower boundary of the sponge layer.

via parameter $\gamma$). The increase in polar vortex strength (measured by the zonal mean wind speed at $60°\mathrm{N}$ and $10\,\mathrm{hPa}$) with increasing values of the polar vertical lapse rate $\gamma$ is shown in Fig. 8 (bottom left), together with the location of the tropospheric jet (Fig. 8, second from bottom), both for the simulations with original PK02 set-up and the set-up with the altered tropical tropospheric temperature gradient (dashed lines). The tropospheric jet location is measured here as the latitude of the maximum

of climatological mean zonal mean zonal winds at $500\,\mathrm{hPa}$. Note that this value can differ from the time-mean of the jet latitude determined at individual days (compare to mean values given in the Appendix, Figs. C1 to C3).

For increasing polar lapse rates $\gamma$, the polar vortex strengthens at a similar rate in the original set-up and the simulations with reduced $\delta_z$ (this also applies to variability, as evident from probability distributions, see Fig. C1). Consistently, also the residual circulation changes in a similar manner in the two sets of simulations. This is diagnosed here as deviation of temperature from

10 the equilibrium temperature, a valid measure of the strength of the residual meridional circulation in the idealized model (see e.g., Jucker et al., 2013). We choose to average this temperature difference from $40°\mathrm{N}$ to $90°\mathrm{N}$, as this is the region of diabatic heating associated with downwelling. Larger values of this temperature difference therefore imply a stronger circulation. In Fig. 8 (top two rows), these temperature differences are displayed for $10\,\mathrm{hPa}$ and $100\,\mathrm{hPa}$ to represent the strength of the circulation in the middle and lower stratosphere, respectively.

Despite the similar response of stratospheric dynamics and polar vortex strength in the simulations with reduced $\delta_z$ compared to the simulations with original PK set-up, the response of the tropospheric jet is strongly damped compared to the original

PK02 set-up. At 500 hPa, the maximum wind location shifts only by a few degrees from the simulation without polar vortex to the one with $\gamma$=5 K km$^{-1}$, while in the original PK02 set-up this shift amounts to more than 10 degrees latitude (see Fig. 8). Note that in the reduced $\delta_z$ simulation, the tropospheric jet is shifted slightly equatorward compared to the reference simulations already for the basic state without polar vortex (see Fig. 8 ), which might be the reason for the different response, as will be discussed in Sec. 4.4.

As has been shown by Gerber and Polvani (2009), the response of the tropospheric jet location to stratospheric forcing is strongly damped in simulations with idealized topography compared to those without topography. The EMIL model simulations presented here reproduce the damped response to stratospheric forcing (changes in $\gamma$) under same set-up as in Gerber and Polvani (2009), as shown in Fig. 8 (middle, yellow line). In the simulations including topography, the tropospheric jet is likewise shifted equatorward with reduced $\delta_z$. However, we detect little difference in the tropospheric jet response to stratospheric polar vortex changes between the two sets of simulations with topography. Thus, in the presence of planetary wave forcing, the different tropospheric equilibrium temperatures appear to play a smaller role for the stratosphere-troposphere coupling. This will further be discussed in Sec. 4.4.

### 4.2.2 Sensitivity to modification of the equilibrium temperature in the winter high latitude lower stratosphere

The simulated winter high-latitude temperature profiles for EMIL simulations with *PK* set-up and WN2 topography are shown in Fig. 9 (left) for varying $\gamma$, compared to temperature profiles from the observationally based "SPARC" climatology for northern winter. The comparison of the simulations to the observational climatology reveals a positive temperature bias in the upper troposphere / lower stratosphere (UTLS) region of the winter high latitudes ($70°$N to $90°$N), when using the standard *PK* set-up with a constant transition pressure of $p_T(\phi) \equiv 100$ hPa. The positive temperature bias remains unchanged for different polar vortex lapse rates $\gamma$. Even for strong decreases of the equilibrium temperature above the $100$ hPa level, the positive temperature bias in the UTLS region cannot be compensated. This is essentially because the equilibrium temperature already exceeds the observational temperatures in that region. Due to the general-circulation transport of heat from the tropics to polar regions throughout the troposphere and stratosphere, the temperature bias even increases. Therefore, every simulation with $p_{Tw} = 100$ hPa necessarily has a too warm UTLS region in the winter high latitudes compared to observations. The warm bias is associated with an unrealistic "step" in the temperature profile, forced by the constant equilibrium temperature profile in the UTLS up to 100 hPa.

In order to approach a more realistic temperature profile in the UTLS region of the winter high latitudes, the transition pressure $p_{Tw}$ is increased. A similar approach was used by Sheshadri et al. (2015), who lowered the transition pressure globally to 200 hPa and showed that this led to an improvement in lower stratospheric zonal winds. Here, we systematically vary the the transition pressure in polar winter high latitudes across $p_{Tw} = 100$ to $450$ hPa as well as $\gamma$ across $\gamma = 1$ to $4$ K km$^{-1}$ (see Table B1).

As discussed in the previous section, a modified version of the tropospheric equilibrium temperature function with a changed vertical temperature gradient was implemented in a previous model version. The whole parameter sweep was performed in this modified model set-up with reduced $\delta_z$, and we repeated simulations for $p_{Tw} = 100$ hPa and $400$ hPa with the standard set-up

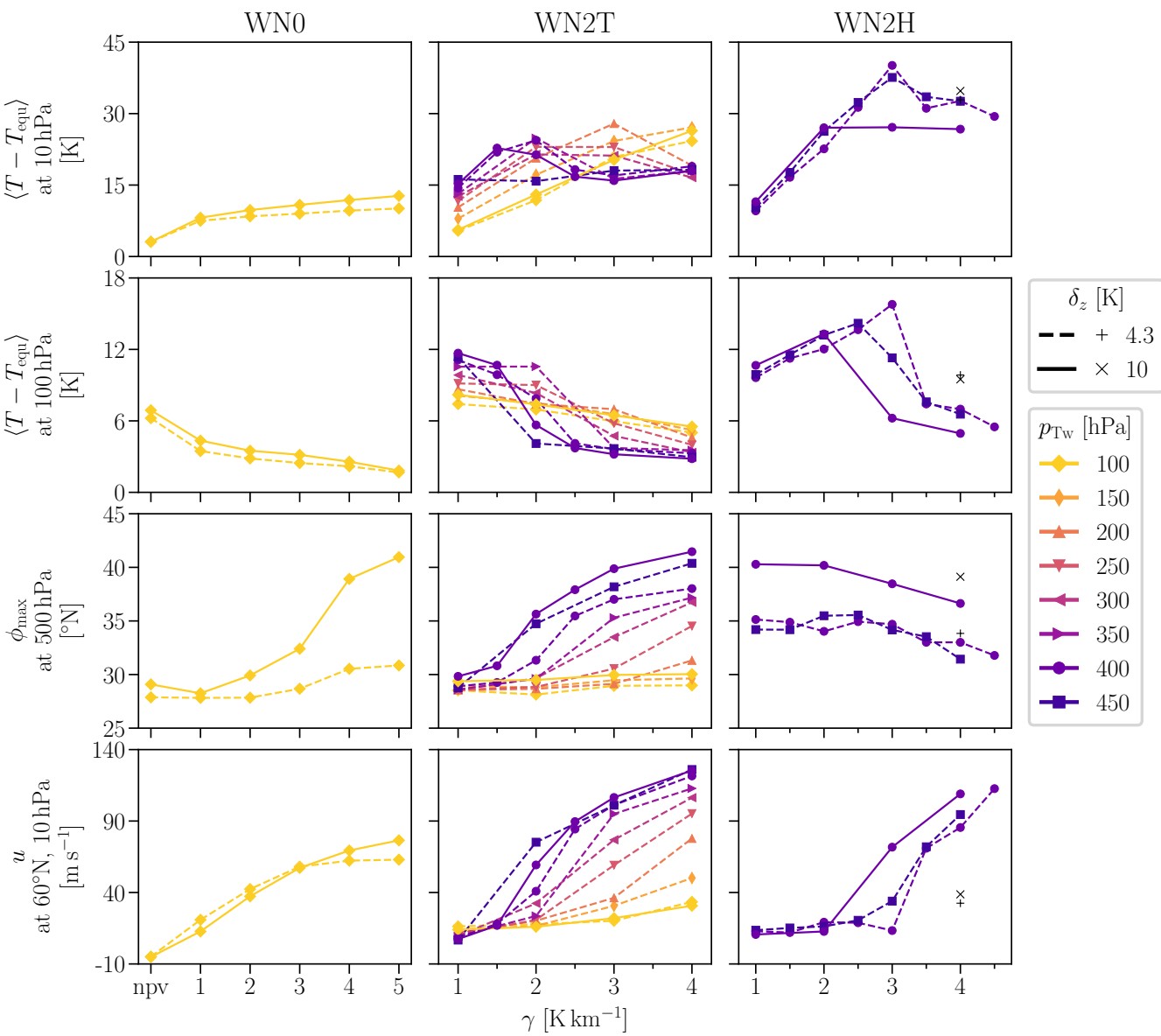

**Figure 8.** Top row: Difference of temperature and equilibrium temperature $T - T_{eq}$ averaged from $40°$N to $90°$N at $10$ hPa. Second row: same, but at $100$ hPa. Third row: Latitude $\phi_{max}$ of the zonal mean zonal wind speed maximum of the tropospheric jet. Fourth row: zonal mean zonal wind $u$ at $60°$N and $10$ hPa. The left column shows results from model simulations without planetary wave forcing, the middle column with WN2 topography of height $h = 3$ km and the right column with WN2 tropospheric heating of amplitude $q_0 = 6$ K day$^{-1}$. Solid lines represent simulations with standard $\delta_z = 10$ K, dashed lines with reduced $\delta_z = 4.3$ K.

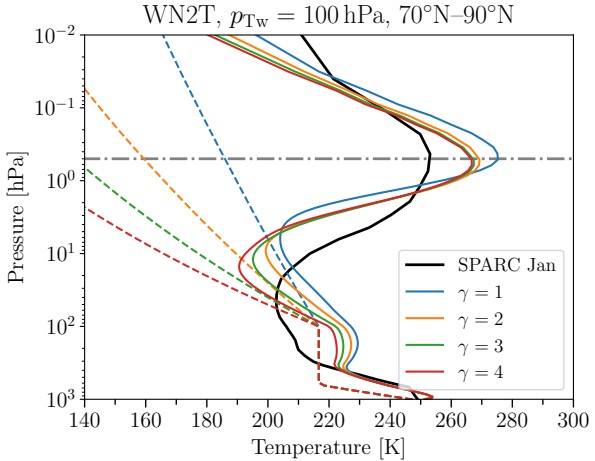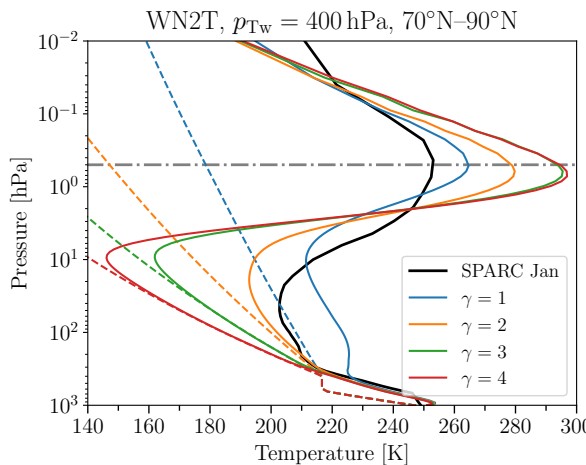

**Figure 9.** Polar winter temperature profiles of model simulations with WN2 topography of height $h = 3\,\mathrm{km}$ and different polar vortex lapse rates $\gamma$ for $p_{\mathrm{Tw}} = 100\,\mathrm{hPa}$ (left) and $p_{\mathrm{Tw}} = 400\,\mathrm{hPa}$ (right), together with the temperature profiles obtained from the SPARC climatology (black line) as well as the equilibrium temperature profiles (colored dashed lines). Averages are based on about 10000 days.

to test the combined sensitivity of the results on modifications in $p_{\mathrm{Tw}}$ and on the tropical tropospheric equilibrium temperature. In Fig. 8, the results of both set-ups are shown (with the modified simulations as dashed lines). The simulations with reduced $\delta_z$ are performed for 1825 days (with the first 300 days considered as spin-up), while the simulations under standard set-up were extended to 10950 days (with 1000 days of spin-up). While the 1825-day simulations are mostly too short to establish
convergence of the climatologies, the qualitative behavior in those simulations is in general in agreement with the results from the extended simulations (as presented in the following).

The polar temperature profiles shown in Fig. 9 for the simulations with standard set-up are very similar to those for the simulations with the modified set-up (not shown). In both set-ups, at $p_{\mathrm{Tw}} = 400\,\mathrm{hPa}$, all equilibrium temperatures in the polar winter UTLS region fall below the corresponding temperatures obtained from observations except the one for $\gamma = 1\,\mathrm{K\,km^{-1}}$
(see the right panel of Fig. 9). For the simulations with $\gamma = 3\,\mathrm{K\,km^{-1}}$ and $\gamma = 4\,\mathrm{K\,km^{-1}}$, the winter high-latitude temperatures are lower than observational temperatures throughout the UTLS region, and follow the equilibrium temperature up to about 30 hPa. Above, the temperature increases strongly, reaching a maximum at around 0.7 hPa. In contrast, the UTLS temperature in the simulation with $\gamma = 1\,\mathrm{K\,km^{-1}}$ is well above the corresponding equilibrium temperature in the UTLS, and the temperature maximum at around 0.5 hPa is weaker. The simulation with $\gamma = 2\,\mathrm{K\,km^{-1}}$ lies in between: Its temperature in the UTLS is
higher than the equilibrium temperature, but less so than for $\gamma = 1\,\mathrm{K\,km^{-1}}$.

The non-linear behavior of the deviation from the equilibrium temperature is illustrated for a variety of values of $\gamma$ and $p_{\mathrm{Tw}}$ in Fig. 8. For low values of $\gamma$, we find an increase in $T - T_{\mathrm{eq}}$ with $\gamma$ in the mid-stratosphere, but a decrease in the lower stratosphere, in agreement with the result of Gerber (2012) that a stronger polar vortex leads to a strengthened circulation in the upper stratosphere and to a weakened circulation in the lower stratosphere (see Fig. 3 of Gerber, 2012, for comparison). However,
above a certain threshold of $\gamma$, the circulation strength decreases with $\gamma$ and then stagnates also in the mid-stratosphere (10 hPa).

This critical value of $\gamma$ depends on $p_{\mathrm{Tw}}$, in line with lower polar $T_{\mathrm{eq}}$ values for higher values of $p_{\mathrm{Tw}}$. In the upper stratosphere (1 hPa), a monotonic increase in $T - T_{\mathrm{eq}}$ both with larger $\gamma$ and $p_{\mathrm{Tw}}$ is found (not shown), but we exclude this analysis because of the likely influence of the upper damping layer.

The strength of the polar vortex increases for larger $\gamma$ and $p_{\mathrm{Tw}}$ values, as expected from the stronger meridional temperature gradient (see Fig. 8). However, the polar vortex increases non-linearly with increasing $\gamma$, with stronger acceleration above a critical value. This is in line with the change in behavior of $T - T_{\mathrm{eq}}$ at 10 hPa when reaching this critical value (e.g., for $p_{\mathrm{Tw}} = 400\,\mathrm{hPa}$, between $\gamma = 2\,\mathrm{K\,km^{-1}}$ to $2.5\,\mathrm{K\,km^{-1}}$ in the modified set-up). Thus, for increases in the prescribed meridional temperature gradient in the polar stratosphere (i.e., via $\gamma$) below a certain threshold, the polar vortex strength increases only very weakly. At the same time, mid-to high-latitude temperatures increase above the corresponding equilibrium temperature (i.e., $T - T_{\mathrm{eq}}$ increases with $\gamma$). Thus, the residual circulation is strengthened, and the associated high-latitude warming counteracts the increase in the prescribed meridional temperature gradient, explaining the weak changes of the polar vortex. Once a certain threshold in the prescribed meridional equilibrium temperature is reached, the polar vortex increases strongly, and at the same time $T - T_{\mathrm{eq}}$ decreases, indicating a reduction in wave driving and thus additional dynamical strengthening of the meridional temperature gradient.

In response to the polar vortex increase for larger $\gamma$ and $p_{\mathrm{Tw}}$ values, the tropospheric jet shifts poleward (see Fig. 8 ). This poleward shift is found to be similarly strong in the standard set-up and the modified (reduced $\delta_z$) set-up. Thus, the strong dependence of the strength of the tropospheric jet response on the set-up found for the simulations without topography (see Sec. 4.2.1 and Fig. 8, left) is not present in the simulations with WN2 forcing, even under the stronger stratospheric forcing in the simulations with increased $p_{\mathrm{Tw}}$. The sensitivity of the rate of the tropospheric jet response will further be discussed in Sec. 4.4.

Overall, the stratospheric circulation responds non-linearly to modifications of the winter equilibrium temperature profile. Lowering the height at which the equilibrium temperature starts to decrease can diminish the high-latitude lower stratospheric temperature bias. To more or less completely remove the warm bias and the associated unrealistic "step", $p_{\mathrm{Tw}}$ has to be increased to $400\,\mathrm{hPa}$. In the simulation set-up with $p_{\mathrm{Tw}} = 400\,\mathrm{hPa}$ and $\gamma = 2\,\mathrm{K\,km^{-1}}$, the winter high-latitude temperature profile is close to reanalysis data (SPARC climatology and ERA-Interim, the latter not shown) in the UTLS region and a moderate oscillation of the temperature in the upper atmosphere is simulated.

## 4.3 Planetary wave generation with topography versus heating

In the experiments presented in the preceding subsection 4.2, an idealized topography was used to generate planetary waves. Recently, Lindgren et al. (2018) suggested an alternative method to generate planetary waves in a set-up with an active stratosphere: they introduced a tropospheric wave-like thermal forcing of the form of Eq. (A11), which is added to the temperature tendency of Newtonian cooling.

For the equilibrium temperature, Lindgren et al. (2018) employ a constant transition pressure of $p_{\mathrm{T}}(\phi) = 200\,\mathrm{hPa}$, i.e. $p_{\mathrm{Ts}} = p_{\mathrm{Tw}} = 200\,\mathrm{hPa}$, and $\epsilon = 0$, i.e., a hemispherically symmetric temperature distribution in the troposphere. Fig. 10 shows the temperature profiles in the winter high latitudes for different simulations that were thermally forced by Eq. (A11). The

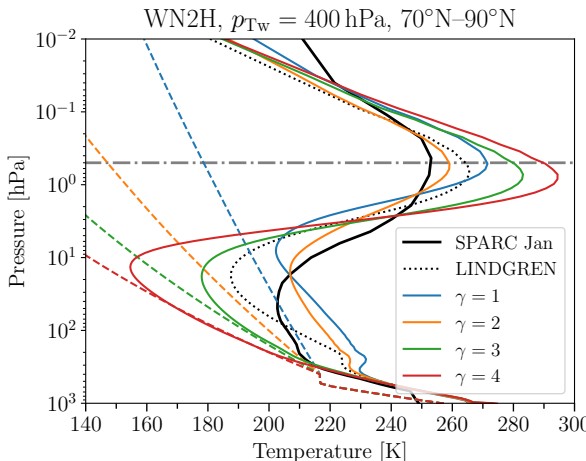

**Figure 10.** Same as Fig. 9, but for model simulations with WN2 tropospheric heating of amplitude $q_0 = 6\,\mathrm{K\,day^{-1}}$ and different polar vortex lapse rates $\gamma$ for $p_{\mathrm{Tw}} = 400\,\mathrm{hPa}$. The temperature profiles obtained from a simulation with the original Lindgren set-up (black dotted line) are added for comparison.

model simulation with the original Lindgren set-up exhibits a too high temperature in the winter high-latitude UTLS region compared to the SPARC climatology for the same reason as was explained for the topographically forced simulations with $p_{\mathrm{T}}(\phi) = 100\,\mathrm{hPa}$ ("PK" set-up) in the previous subsection: the decrease of the equilibrium temperature due to $\gamma$ starts too high to be able to compensate the warm bias. This motivated the investigation of model simulations with a larger transition pressure

$p_{\mathrm{Tw}}$ in the winter high latitudes for the thermally forced simulations as well.

In our model simulations with WN2 tropospheric heating, we similarly use $\epsilon = 0$, but return to $p_{\mathrm{Ts}} = 100\,\mathrm{hPa}$ and vary $p_{\mathrm{Tw}}$.[1] In addition to the profile obtained from the Lindgren set-up, Fig. 10 contains the winter high-latitude temperature profiles for different polar vortex lapse rates, $\gamma$ and for $p_{\mathrm{Tw}} = 400\,\mathrm{hPa}$.

Fig. 8 (right) shows results from the simulations with thermally forced planetary waves, and again both the modified (reduced

$\delta_z$) simulations as well as simulations with the standard tropospheric equilibrium temperatures are included. As discussed for the topographically forced simulations, we also find a non-linear behavior of the stratospheric circulation in the thermally forced simulations: A state with a weak polar vortex (for $\gamma$ smaller than about $3\,\mathrm{K\,km^{-1}}$, see Fig. 8 bottom right) manifests in a positive temperature bias in the UTLS region of the winter high latitudes (Fig. 10), and increasing temperature deviation from the equilibrium temperature with increasing $\gamma$ (Fig. 8 top right). A state with a strong polar vortex arises for $\gamma \geq 3$ to

$3.5\,\mathrm{K\,km^{-1}}$ (Fig. 8 bottom right), with temperature following the equilibrium temperature very closely in the UTLS region (see Fig. 8).

---

[1]The difference in the equilibrium temperature between $p_{\mathrm{Ts}} = 100\,\mathrm{hPa}$ and $p_{\mathrm{Ts}} = 200\,\mathrm{hPa}$ is marginal since the US standard atmosphere between about 55 hPa and 226 hPa is isothermal at 216.65 K. Thus, for different values of $p_{\mathrm{Tw}}$ only the lower region of the polar vortex lapse rate around $\phi_0 = 50°\mathrm{N}$ experiences a slight change when employing $p_{\mathrm{Ts}} = 100\,\mathrm{hPa}$ instead of $p_{\mathrm{Ts}} = 200\,\mathrm{hPa}$.

The response of the polar vortex to changes in the equilibrium temperature is similar between the topographically versus thermally forced model simulations in that a transition from a weak to a strong polar vortex is found for both cases. Thermally forced model simulations also show an increase of the strength of the meridional circulation at $10\,\mathrm{hPa}$ up to a certain threshold of $\gamma$, similar than for the topographically forced simulations (see Fig. 8, top). Note, however, that the threshold is higher for the thermally forced simulations for identical equilibrium temperature. The change in the behavior of the meridional circulation in the model simulations with $p_{\mathrm{Tw}} = 450\,\mathrm{hPa}$ and $p_{\mathrm{Tw}} = 400\,\mathrm{hPa}$ (both for the modified and standard set-up) appears at the same polar vortex lapse rates, at which the polar vortex starts to strengthen. At $100\,\mathrm{hPa}$, the topographically forced simulations show a (non-linear) decrease of the circulation strength with increasing $\gamma$ for all values of $p_{\mathrm{Tw}}$, while in the thermally forced simulations the circulation in the lower stratosphere responds in a similar non-linear way as at $10\,\mathrm{hPa}$.

Further, we compare the response of the tropospheric jet to changed equilibrium temperatures in topographically forced simulations to the response in the thermally forced simulations. As discussed in the last section, in the case of the topographically forced model simulations, the location of the free tropospheric jet shifts poleward in the simulations with a stronger stratospheric polar vortex. However, when the planetary waves are thermally forced, the free tropospheric jet maximum remains at an almost constant latitudinal location, or even moves equatorward (see lower panels of Fig. 8). Even strong increases in the stratospheric polar vortex for $\gamma > 3\,\mathrm{K\,km^{-1}}$ at $p_{\mathrm{Tw}} = 400\,\mathrm{hPa}$ and for $\gamma > 2.5\,\mathrm{K\,km^{-1}}$ at $p_{\mathrm{Tw}} = 450\,\mathrm{hPa}$, respectively, are not accompanied by a northward shift of the free tropospheric jet maximum.

Thus, while the non-linear increase in the stratospheric polar vortex strength is overall similar in the topographically and diabatically forced simulations, other aspects of the circulation response show distinct differences. Overall, the different behavior of model simulations with topographically and thermally forced circulations outlined here indicates that the thermally forced simulations might have to be used with caution, in particular for studying troposphere-stratosphere coupling.

## 4.4 Discussion on dynamical states of the stratospheric polar vortex and the tropospheric jet

The sensitivity simulations with respect to modifications of the equilibrium temperature (Sec. 4.2) and to planetary wave generation (Sec. 4.3) revealed the following: Firstly, the stratospheric polar vortex responds non-linearly to the enhancement of the meridional temperature gradient in the simulations with planetary wave forcing. Secondly, the strength of the tropospheric jet response to this stratospheric vortex strengthening depends on the model's basic state. These two results are discussed in the following.

**Stratospheric polar vortex regimes**

We found strong non-linear strengthening of the polar vortex with enhanced meridional temperature gradients (via increasing $\gamma$ and/or $p_{\mathrm{Tw}}$), i.e., the polar vortex transitions from a weak state to a strong state. In between, climate states are found in which the vortex appears to alternate between those two states, indicative of a transition between different vortex regimes. This is reflected in changes in the polar vortex variability, as shown in Fig. 5 for the topographically forced simulations with $p_{\mathrm{Tw}} = 400\,\mathrm{hPa}$ and $\gamma = 1$, 2, and $3\,\mathrm{K\,km^{-1}}$. While the simulation with the weakest polar vortex ($\gamma = 1\,\mathrm{K\,km^{-1}}$) exhibits large variability with frequent crossings of the zero-wind line (indicative of sudden stratospheric warmings), in the simulation

with $\gamma = 3\,\mathrm{K\,km^{-1}}$ the wind oscillates around its large mean value without crossing the zero wind line. With an intermediate polar lapse rate ($\gamma = 2\,\mathrm{K\,km^{-1}}$), episodes with strong stable winds are disrupted by sudden decelerations, and the polar vortex remains in a weak state for an extended period (up to a few hundred days) thereafter, i.e., the vortex alternates between a strong and a weak regime. The regime behavior is further supported by the shape of the probability distribution of the maximum wind at 10 hPa for those three simulations (see Fig. C2): while in the simulation with $\gamma = 1\,\mathrm{K\,km^{-1}}$, the polar vortex strength is bound below $50\,\mathrm{m\,s^{-1}}$, and in the simulation with $\gamma = 3\,\mathrm{K\,km^{-1}}$, the vortex strength is always above $75\,\mathrm{m\,s^{-1}}$, for $\gamma = 2\,\mathrm{K\,km^{-1}}$ a broad, nearly bimodal, distribution is found. The distribution functions for the modified (reduced $\delta_z$) simulations (not shown) are noisier due to the shorter simulation length, but show a similar behavior than those shown in Fig. C2. Further, also the diabatically forced simulations indicate a regime shift of the polar vortex: the probability distribution functions of the polar vortex strength change strongly between the weak and strong vortex state (e.g., change in sign of skewness, see Fig. C3), supporting further that we see a regime transition. In terms of polar vortex changes with increasing prescribed polar stratospheric temperature gradient, the simulations with different set-up show only minor differences (both for modifications in $\delta_z$ and in planetary wave generation).

Analysis of the residual circulation strength (as measured by deviation from equilibrium temperature, see Sec. 4.2) indicates, that the sudden strengthening of the polar vortex is associated with a strong decrease of the residual circulation in the mid stratosphere. In the weak vortex regime, the weak westerly winds allow for vertical planetary wave propagation, as expected from linear wave theory (Charney and Drazin, 1961). The waves dissipate in the vicinity of the polar vortex, and drive the residual circulation, that in turn reduces the meridional temperature gradient. When increasing the prescribed meridional equilibrium temperature gradient up to a certain threshold, the wave forcing appears to be increasing. Thus, the stronger residual circulation counteracts the decreasing equilibrium temperatures in the polar stratosphere, and the polar vortex changes little. However, when lowering polar equilibrium temperatures to a certain threshold, the wave driving seems not to be able to counteract the strengthening meridional temperature gradient any longer: the polar vortex strengthens, and thereby vertical wave propagation is inhibited by the strong winds (again by simple arguments following Charney and Drazin, 1961). The positive feedback between strong winds and suppressed wave forcing might explain the suddenness of the polar vortex strengthening, or in other words, explains why the vortex behaves regime-like. The non-linear interaction of planetary waves and the mean flow is less pronounced in simulations without planetary wave forcing, explaining that the polar vortex responds more linearly to the prescribed temperature changes. However, the above line of argument will have to be tested by more thorough analysis of wave fluxes in the simulations.

**Tropospheric jet location and its response to stratospheric forcing**

The strength of the tropospheric jet response to the stratospheric vortex strengthening is summarized in Fig. 11, where the tropospheric jet location is shown as a function of the polar vortex strength for increasing polar lapse rate $\gamma$ for a number of different experiment set-ups. The slope of the lines in Fig. 11 thus indicates the sensitivity of the tropospheric jet location to stratospheric polar vortex changes. The strongest response is found in the experiments with the set-up as in the original PK02 study (black line), while the response of the tropospheric jet is considerably weaker in the set-up with reduced (sub-)tropical

static stability (reduced $\delta_z$, black dashed line). This holds for locating the tropospheric jet in the mid-troposphere (Fig. 11 left) as well as near the surface (Fig. 11 right).

The inclusion of topographically generated planetary waves dampens the response of the tropospheric jet (as previously shown by Gerber and Polvani, 2009), both in the original and the reduced $\delta_z$ set-up (see blue lines). With reduced polar LS

temperatures (i.e., $p_{\mathrm{Tw}}$=400 hPa), the stronger polar lapse rate leads to enhanced polar vortex strengths, and the tropospheric jet is shifted northward with an intermediate, almost linear rate (red lines). Again, the jet diagnosed near the surface and in the mid-troposphere reveal similar rates of change (compare Fig. 11 left and right). Under same set-up of $T_{\mathrm{eq}}$ but with diabatically forced planetary waves, the tropospheric jet diagnosed in the mid-troposphere is almost insensitive to the stratospheric polar vortex increase. However, when diagnosed near the surface, the jet shifts poleward at an even higher rate than in the topographically

forced simulations (Fig. 11, purple lines).

As mentioned in the Introduction, the strong poleward jet displacement in the original PK02 experiments has been shown to be associated with a shift between subtropical and mid-latitude jet regimes (e.g., Chan and Plumb, 2009). While we do not find the bimodal distribution of the near-surface jet location as shown by Chan and Plumb (2009), a broadening of the probability distribution of the tropospheric jet location in the simulation with $\gamma$=3 K km$^{-1}$, and a change in skewness of the

distributions (from positive to negative between $\gamma$=3 K km$^{-1}$ and $\gamma$=4 K km$^{-1}$) indicates a regime shift in the jet location also in our simulations. The probability distributions are appended for reference, see Fig. C1.

In the simulations without topography and with reduced $\delta_z$, the tropospheric jet shifts slightly poleward with increasing $\gamma$, but remains in the subtropical regime in (see Fig. 11 and Fig. C1). We presume that the more equatorward location of the tropospheric jet in the basic state inhibits the regime transition to a poleward located tropospheric jet in the reduced $\delta_z$

simulations in response to the stratospheric forcing. The equatorward shift of the jet in the basic state is consistent with the previously reported jet response to the insertion of diabatic heating in the tropical upper troposphere (e.g., Butler et al., 2010). The anomalies in the equilibrium temperature extend into the subtropics (see Fig. 2), so that the static stability is reduced in the tropical and subtropical troposphere. The reduced static stability effectively enhances the tendency for baroclinic instability in the subtropics (e.g., Lu et al., 2008), which could favor a subtropical eddy-driven jet location. This is consistent with the

persistent subtropical jet regime in the simulations with reduced $\delta_z$. Whether the jet would move to the higher-latitude regime in our reduced $\delta_z$ simulations under stronger stratospheric forcing remains to be investigated.

In the simulations with topography, the jet is likewise shifted equatorward with reduced $\delta_z$. However, the jet response to the strengthening stratospheric polar vortex is not different in the simulations with reduced $\delta_z$ (compare red solid and dashed lines in Fig. 11). This could be due to the additional effects of planetary waves, again consistent with the result by Lu et al. (2008)

that changes in the jet location are more tightly related to the subtropical stability in the SH (with little planetary wave activity) than in the NH.

When the planetary waves are forced diabatically, the basic state mid-tropospheric jet is likewise shifted equatorward in the simulations with reduced $\delta_z$. The response to stratospheric forcing is again unaltered, as in both sets of simulations the location of maximum winds in the mid-troposphere appears to be fixed in the mid-latitude regime throughout the range of polar vortex

strengths. Consistent with the static stability argument, the diabatically forced simulations exhibit the highest gross static

stability in the subtropical troposphere compared to the other sets of simulations (not shown). Thus, the reduced tendency towards baroclinic instability in the subtropics favors the mid-latitude jet regime. However, as the equilibrium temperature is identical in the topographically and diabatically forced simulations, there is no obvious reason for the enhanced stability and the different basic states of the tropospheric jet.

Overall, the set of simulations presented here confirms that the tropospheric jet in the dry dynamical core model tends to fall into either a subtropical or a mid-latitude regime. This extends the result of Chan and Plumb (2009), in that different kinds of modifications of the set-up can lead to states with a strongly damped response of the tropospheric jet location to the stratospheric forcing. In Chan and Plumb (2009), enhancement of the surface equilibrium temperature equator-to-pole gradient led to states with a weak jet response, because the tropospheric jet is located in the mid-latitude regime already for a weak

stratospheric polar vortex, similar to our diabatically forced simulations. However, in the diabatically forced simulations the wind maximum in the mid-troposphere and near the surface seem to be de-coupled: while the former remains at a rather constant latitude, the latter strongly moves poleward under strong stratospheric forcing, with signs of a regime transitions to an even higher latitude regime (indicated by bimodality, see Fig. C3). Next to this state with the jet remaining in the mid-latitude regime, we also found a state in which the tropospheric jet remains in the subtropical regime even under strong forcing (namely

in the set-up with reduced $\delta_z$ and no planetary wave forcing). However, the jet might likely move to the mid-latitude regime if the stratospheric forcing was increased further, but that remains to be investigated.

## 5    Application examples

In the previous sections, the implementation of the EMIL model was documented and tested, and modified set-ups were introduced, showing that the model is well suited for further applications. In the following, two examples of research applications

with the dynamical core model are shown. First, variability and changes in tracer transport in response to changes in the polar vortex are analyzed, using the simulation set-up with the modified equilibrium temperature (see Sec. 4.2). Secondly, the localized heating routine (see Sec. 2.1) is used to force an idealized monsoon circulation system.

### 5.1    Chemistry and tracer transport

With the implementation of the idealized model set-up in the MESSy framework, all tracer utility and chemistry submodels

can be easily used to study the tracer transport in the idealized model. Within the chemistry submodel MECCA (Sander et al., 2019), tailor made chemical mechanisms can be selected to the users' needs, allowing for selection of simplified chemistry set-ups. As a proof of concept, we present results from simulations where the only selected chemical reactions are the photolysis of Chlorofluorocarbons (CFCs, namely CFC-11 and CFC-12).

Technically, this simulation set-up requires, in addition to the "standard" EMIL set-up, to switch on submodels for solving

chemical kinetics (MECCA, Sander et al., 2019), for calculating photolysis rates (JVAL, Sander et al., 2014), for determining orbital parameters (ORBIT, Dietmüller et al., 2016) and submodels for tracer definition (TRACER and PTRAC, Jöckel et al., 2008) and tracer boundary condition nudging (TNUDGE, Kerkweg et al., 2006). CFC mixing ratios were set to values

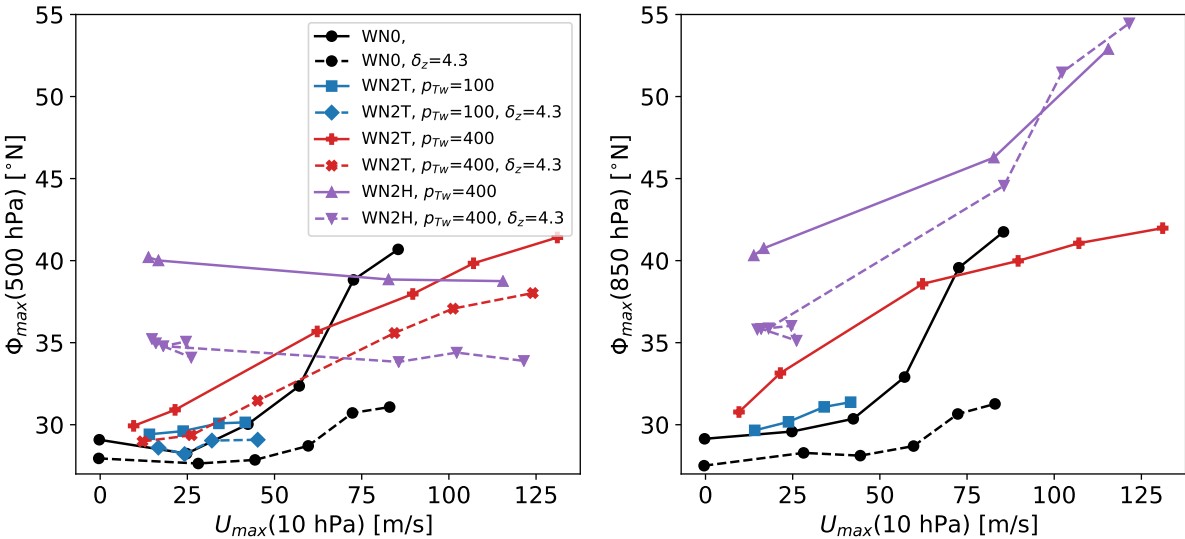

**Figure 11.** Latitude of the zonal mean zonal wind maximum at 500 hPa (left) and 850 hPa (right) displayed against the maximal zonal mean zonal wind $U_{max}$ at 10 hPa for various simulation set-ups: simulations under PK set-up without topography (labeled WN0), and with WN2 planetary waves forced topographically (labeled WN2T) and diabatically (labeled WN2H) for various values of the winter transition pressure $p_{Tw}$ and tropospheric tropical vertical temperature gradient $\delta_z$ (see legend; if no value is given, the parameter is set to default). Each symbol displays the value diagnosed from the climatology of simulations with varying polar vortex lapse rate $\gamma$. The values for the WN2T simulations with $\delta_z$=4.3 K are not shown on the right, because data at 850 hPa was not saved appropriately.

representative of year 2000 at the surface, and tracers were initialized with a mean distribution from an earlier EMAC simulation. To obtain constant January conditions of solar irradiance (compatible with the idealized thermodynamical forcing of the dynamics), in the TIMER namelist, a perpetual month simulation can be selected.

With the given model set-up including chemical tracers, the influence of idealized dynamical variability on chemically active
species can be studied. Shown in Fig. 12 are zonal mean CFC-11 mixing ratios at 50 hPa as function of latitude and time in a simulation with *PK* set-up, with the reduced value of $\delta_z$=4.3 K and with $p_{Tw}$=400 hPa and $\gamma$=2 K km$^{-1}$. The polar vortex variability leads to variability in CFC-11 mixing ratios in particular at high latitudes. As diagnosed from the time series of zonal mean zonal wind at 60°N and 10 hPa (top panel in Fig. 12, black line) sudden stratospheric warming (SSW) events occur at around day 600 and day 1350, as indicated by the gray dashed lines. While the most common definition of the zero-crossing
of the 10 hPa zonal wind is met a few days later, the lines are inserted at the dates of the strongest deceleration of the wind. Both events are followed by an extended period with a weak polar vortex.

For both SSW events, the CFC-11 mixing ratios drop at high latitudes simultaneously with the drop of zonal winds at 10 hPa. However, around 200 days after the SSW events, high latitudes mixing ratios have increased again. This behavior can be explained as follows: Simultaneously with the SSW, strong downwelling occurs at high latitudes (north of 60°N), driven by
the strong wave dissipation that effected the SSW (see red line in top panel in Fig. 12). The enhanced downwelling transports

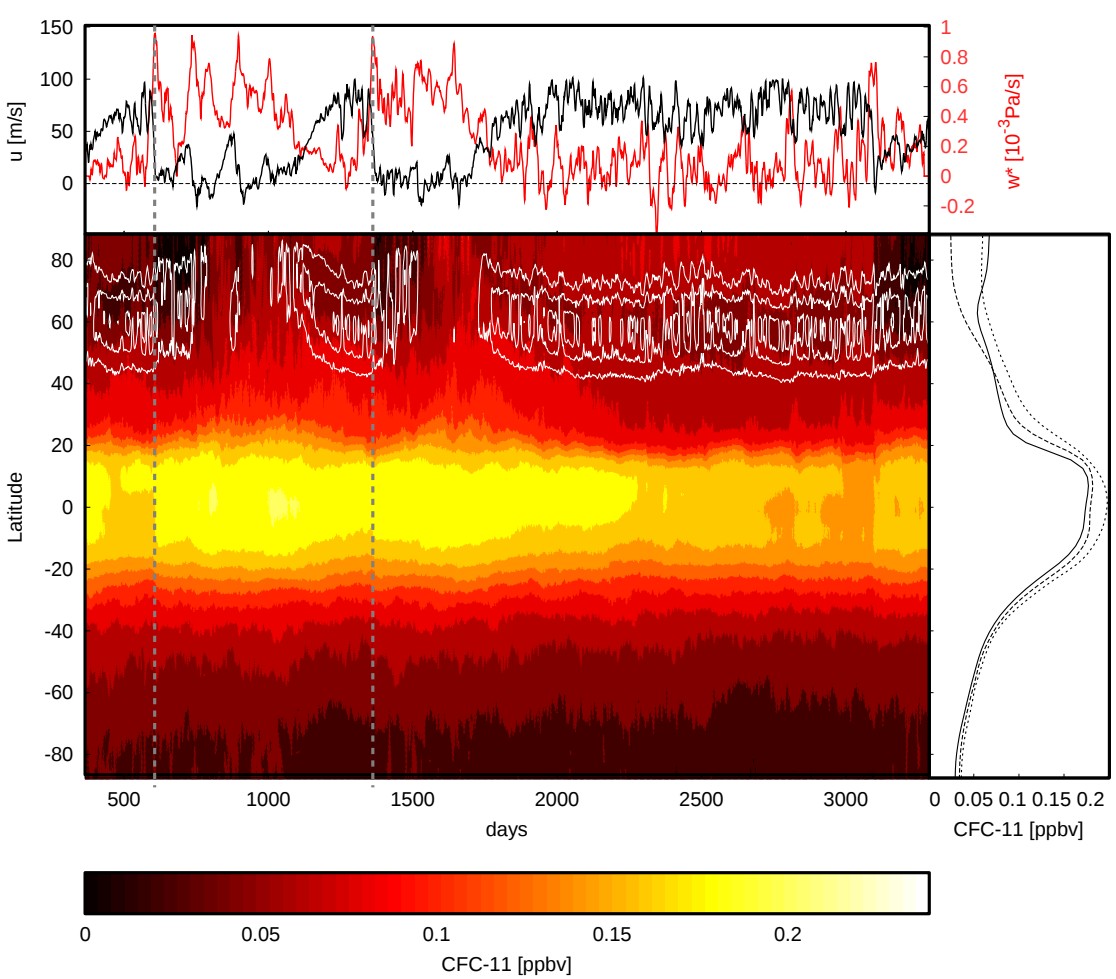

**Figure 12.** Top: time series of zonal mean zonal wind $\overline{u}$ at $60°$N and $10$ hPa (black) and mean $\overline{w^*}$ at $50$ hPa and $60°$-$90°$N (red). Middle panel: zonal-mean CFC-11 mixing ratios (color, in ppbv) at $50$ hPa as function of simulated day and latitude, and zonal mean zonal wind at $50$ hPa (white contours, interval $15$ ms$^{-1}$). Vertical gray lines mark dates of SSW (centered here at the dates of the strongest wind decelerations). Right: time-mean CFC-11 mixing ratios as function of latitude over days with strong vortex (days 400-600; 1200-1350; 2000-3000, black solid), over days following SSW with strong downwelling (day 600-780; 1380-1500; 3100-3280, dashed) and over days with eroded polar vortex (day 800-1000 and 1580-1700, dotted).

CFC-depleted air from higher altitudes downward. However, due to the diminished vortex in the period after the SSW, air from mid-latitudes with higher CFC mixing ratios can be mixed towards the pole, thus leading to an enhancement of CFC mixing ratios at high latitudes. This is evident in Fig. 12 around days 800-1000 and days 1500-1700, when zonal winds are below $15$ ms$^{-1}$.

The transport anomalies are evident in the latitudinal profiles of CFC-11 mixing ratios, as shown in the right panel of Fig. 12: during episodes with a strong polar vortex (solid line), there is a local minimum in mixing ratios close to the polar vortex edge (in agreement with strongest downwelling at the vortex edge, see Fig. 13, third panel), denoting the separation between mid-latitude and high-latitude air by the vortex. Just at and after the SSW events, CFC mixing ratios drop at high latitudes (dashed line), while in the episodes with eroded vortex, CFC-11 mixing ratios are enhanced at mid- to high latitude and no mixing barrier can be identified (dotted line).

Two additional simulations were performed with idealized changes in the polar vortex (intermediate: $\gamma = 2\,\mathrm{K\,km}^{-1}$, weak vortex: $\gamma = 1\,\mathrm{K\,km}^{-1}$, strong vortex $\gamma = 3\,\mathrm{K\,km}^{-1}$). The resulting climatological mean CFC-11 mixing ratios at $50\,\mathrm{hPa}$ are shown in Fig. 13 (top). The differing dynamical states of the three simulations are clearly reflected in the tracer mixing ratios: The simulation with the weak vortex ($\gamma = 1\,\mathrm{K\,km}^{-1}$, red) shows highest CFC-11 mixing ratios in the tropics to mid-latitudes, with a smooth transition from tropics to high-latitudes, in line with strongest upwelling (see Fig. 13c; see also results in Sec. 4.2) and strong mid-latitudes wave driving that results in mixing (see Fig. 13d). In the simulation with a strong vortex ($\gamma = 3\,\mathrm{K\,km}^{-1}$, blue), mixing ratios in the tropics are lower, due to weaker upwelling in the lower stratosphere, and the gradient to mid-latitudes is steep. This can be explained both due to weaker mixing (see Fig. 13d), but also because the region of downwelling is shifted towards lower latitudes (see Fig. 13c): In the simulations with stronger polar vortex ($\gamma = 2\,\mathrm{K\,km}^{-1}$ and $3\,\mathrm{K\,km}^{-1}$), downwelling is maximized at the equatorward flank of the polar vortex, and is weak within the vortex. In contrast, in the simulation with a weak polar vortex ($\gamma = 1\,\mathrm{K\,km}^{-1}$), downwelling is maximized more poleward and is stronger also at high latitudes. The maximum of downwelling in the mid-latitudes as well as the high isolation of vortex air in the strong vortex case likely explains why CFC-11 mixing ratios are elevated within the vortex. The intermediate simulation with $\gamma = 2\,\mathrm{K\,km}^{-1}$ lies in between the other two simulations, but shows highest variability (largest standard deviation) in most quantities, as expected, since this simulation oscillates between states with a weak and strong vortex (see Fig. 12 and Sec.4.2).

As demonstrated here, the idealized set-up of the simulation allows to study the role of vortex variability or specifically forced polar vortex strength changes on tracer mixing ratios in an isolated manner, i.e., the absence of other chemical processes or variability like the annual cycle.

## 5.2 Monsoon anticyclone forced by localized idealized heating

Idealized models have been widely used to understand the basic processes occuring in the monsoon regions (e.g., Gill, 1980; Yano and McBride, 1998; Bordoni and Schneider, 2008). In particular, the development and dynamics of the monsoon anticyclones in the UTLS over Asia (e.g., Gill, 1980; Hoskins and Rodwell, 1995; Liu et al., 2007; Wei et al., 2014, 2015; Hsu and Plumb, 2000; Amemiya and Sato, 2018; Siu and Bowman, 2019) and North America (Siu and Bowman, 2019) have been investigated using simplified modelling approaches. Here we impose an idealized heating field to force monsoonal anticyclones. The analyses presented here document the capability of the model system to apply such a forcing and to simulate anticyclones with realistic properties. This will enable more rigorous, in-depth analyses of the dynamics and of transport processes in such idealized monsoon simulations. These future analyses will exploit further capabilities of the MESSy-infrastructure, in particular the inclusion of idealized and realistic tracers to study transport processes.

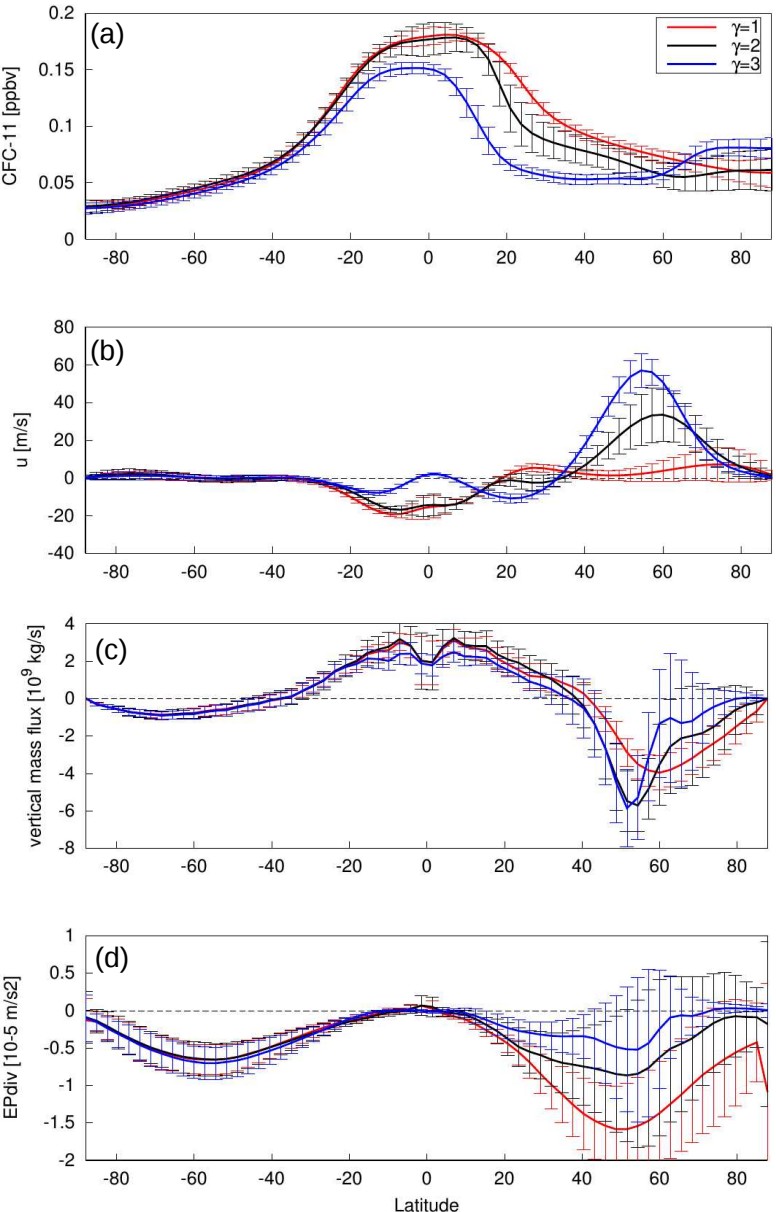

**Figure 13.** a) Zonal mean CFC-11 mixing ratios , b) zonal mean zonal wind , c) mean vertical mass flux and d) EP flux divergence, all at 50 hPa as function of latitude for EMIL simulations with *PK* set-up with $p_{Tw}$=400 hPa and $\gamma$ =1 (red) , 2 (black) and 3 K km$^{-1}$ (blue).

In the following we show results from a T42L90MA simulation with the standard *'HS'* set-up of equilibrium temperature and NH summer constellation, i.e., hemispheric asymmetry is caused by setting the asymmetry factor $\epsilon$ to -10 K. The first two years of this simulation have been neglected and here results from the third simulation year are presented. On top of the basic state a regionally confined heating source is imposed in the NH tropics to subtropics (following Eqs. A12-A16 with $\phi_0^m = 20°$ N, $\lambda_0^m = 90°$ E, $\delta\phi^m = 10°$, $\delta\lambda^m = 30°$). In the vertical, the heating extends from $p_{bot}^m = 800\,\mathrm{hPa}$ to $p_{top}^m = 100\,\mathrm{hPa}$. The heating is turned on at day 0 of the simulation with a spin up of $t_s^m = 20\,\mathrm{days}$. Other temporal variations are not considered as $q_{\mathrm{temp}}^m = 0\,\mathrm{K\,day^{-1}}$. The temporally constant (neglecting the spin-up period) heating is imposed with $q_0^m = 8\,\mathrm{K\,day^{-1}}$. After the spin-up period, the average total energy per day that is added into the model due to this additional heat source (deduced from 6 h model output) is slightly below $21 \times 10^{19}\,\mathrm{J/day}$. This heating is of the same order of magnitude as the idealized heat source of $6 \times 10^{19}\,\mathrm{J/day}$ prescribed in Siu and Bowman (2019) to model the North American monsoon anticyclone (see their experiments 5a-5e).

The mean geopotential height field at 100 hPa for this T42L90MA simulation with the described idealized heating is shown in Fig. 14(a). A clear anticyclone is produced in response to the additional heating. This anticyclone is similar to the Asian monsoon anticyclone (e.g., Hoskins and Rodwell, 1995; Zhang et al., 2002; Randel and Park, 2006; Nützel et al., 2016). Fig. 14(b) shows a latitude vs. pressure cross section of zonal winds averaged over all longitudes. The zonal winds averaged over the anticyclone region are overlayed in black contours. The positive wind speed in the north and the negative wind speeds towards the equator marking the edges of the anticyclone are clearly visible (cf. Figs. 2 and 1 in Randel and Park, 2006; Garny and Randel, 2016, respectively).

Fig. 15 shows the temporal variation of the monsoon anticyclone during a 5-day period of the simulation. The daily geopotential height fields in Fig. 15 show an example of a splitting event of the anticyclone. On the first day of the depicted period, the anticyclone is elongated (Fig. 15a). After that the anticyclone splits and two days later two anticyclone centers can be identified (red dots in Fig. 15b). Four days after the elongated phase, the western center has decayed and the eastern center has moved slightly westwards to roughly 90° E (Fig. 15c). Such splitting events (sometimes also denoted westward eddy sheddings, Figs. 15 and 16 in Hsu and Plumb, 2000), as shown in Fig. 15 are (typical) features during the monsoon period (e.g., Fig. 13 in Garny and Randel, 2013; Vogel et al., 2015; Nützel et al., 2016; Pan et al., 2016).

Coincidentally, an example of eastward eddy shedding was found during the same 5-day period and is indicated via arrows in Fig. 15. This phenomenon has been previously investigated in several publications (e.g., Dethof et al., 1999; Garny and Randel, 2013; Vogel et al., 2014; Nützel et al., 2016) and constitutes a major mode of variability observed in the monsoon anticyclone. During the depicted period on the eastern edge of the anticyclone a filament is torn off: On the first day, the anticyclone stretches to the east (Fig. 15a). Two days later this development is even more pronounced (Fig. 15b) and again two days later a filament is separated from the main anticyclone (Fig. 15c).

Those examples show that the EMIL model implementation is suited to simulate a monsoon-like anticyclone reminiscent of the realistic mean state by imposing an idealized localized heating. Also typical features of the variability of the monsoon anticyclone are reproduced in the presented EMIL simulation. Details of the variability of the idealized anticyclone, also under

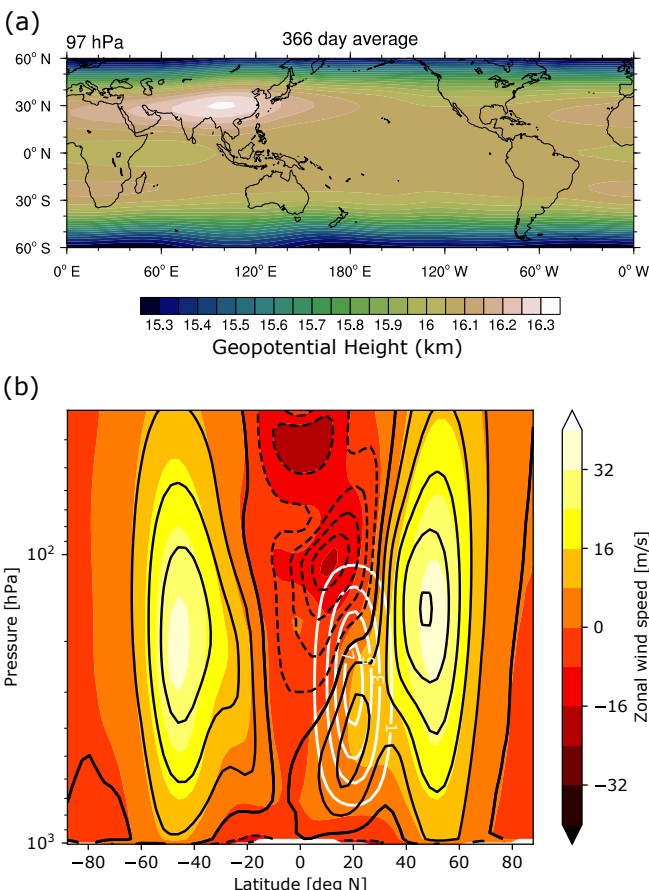

**Figure 14. (a)** Mean anticyclone structure via geopotential height (km) at ∼100 hPa from 366 days of the simulation (map included for orientation and scale purposes only; i.e., the simulation features no orography etc.). **(b)** Vertical cross section of zonal mean wind (colour-coded) and wind in the anticyclone region averaged over 60-120°E (black contours; in steps of $8\,\mathrm{m\,s^{-1}}$; negative values dashed). White contours show the maximum along the longitudes of the implied heating function (in $\mathrm{K\,day^{-1}}$).

constant versus time-varying forcing, and its impact on troposphere-stratosphere tracer transport will be the subject of future studies.

## 6 Summary and Outlook

The implementation of a dry dynamical core model set-up within the MESSy framework is documented. This set-up, denoted
5  EMIL (ECHAM/MESSy IdeaLized), is shown to perform consistently with established dry dynamical core benchmarks, both earlier configurations of the ECHAM core, and those developed by other modeling centers.

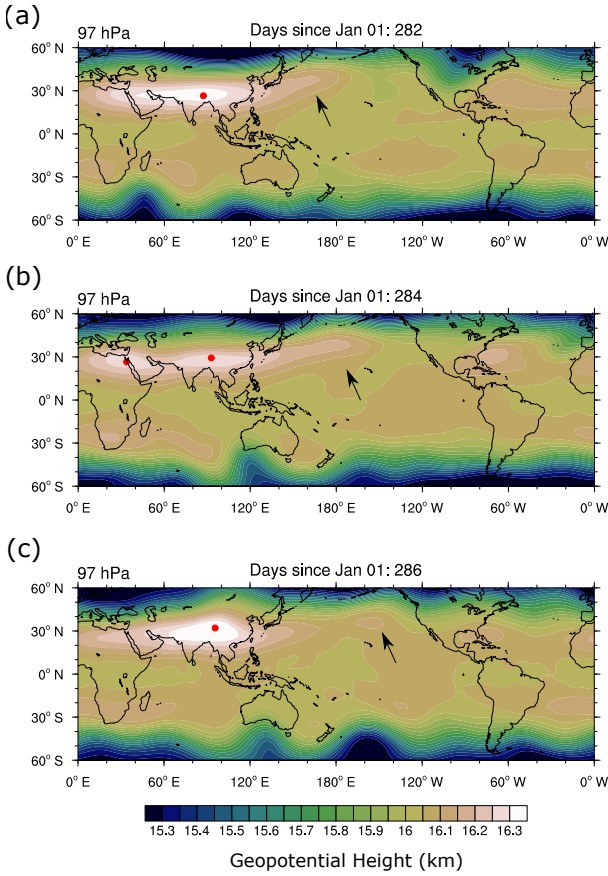

**Figure 15.** Evolution of geopotential height at ∼100 hPa showing an example of a splitting event and an eastward shedding event over a 5-day period. Red dots are idicating the approximate positions of the anticyclone centers, while arrows highlight the eddy shedding event.

The implementation of the submodel for temperature and wind relaxation (submodel RELAX), necessary for the dynamical core set-up, includes pre-implemented functions for the parameters for Newtonian cooling and Rayleigh friction as described in HS94 and PK02. Extensions to those functions are added, namely the option to change the transition pressure between tropospheric to stratospheric equilibrium temperatures in the winter hemisphere. Further, the submodel includes the possibility to include additional diabatic heating either by pre-implemented functions for zonal mean, localized or wave-like heating, or by externally read files. Thus, the implementation provides a tool-kit for the users to choose model set-ups to their needs.

Modifications to the set-up by PK02 and Gerber and Polvani (2009), which were used frequently in the past, are presented with respect to the shape of the upper sponge layer and with respect to the equilibrium temperature profile in the winter stratosphere. The damping coefficient of the upper sponge layer is set to increase exponentially with height instead of quadratically, resembling more closely parametrized drag by GW in the full model and leading to more realistic temperature profiles in the stratopause region. However, the impact outside the sponge layer is considerable only above 10 hPa at high latitudes. Modifications of the equilibrium temperature in the high latitude UTLS region are performed by increasing the transition pressure ($p_{\mathrm{Tw}}$)

between troposphere and stratosphere at high-latitudes (thus, the decrease of temperatures forming the polar vortex starts at lower altitudes). We find that increasing the transition pressure from 100 hPa to 400 hPa results in a realistic mean temperature profile (with the polar lapse rate $\gamma=2\,\mathrm{K\,km^{-1}}$), thus correcting for the UTLS warm bias of the PK02/ Gerber and Polvani (2009) simulations. With the increased transition pressure, we find a regime-like behavior of the polar vortex. The polar vortex

appears to transition from a weak to a strong regime with increasing stratospheric polar lapse rate (i.e., increasing $\gamma$ and $p_{\mathrm{Tw}}$). The simulation with the most realistic mean temperature profile is at the transition point between those regimes. While we presented evidence here that the polar vortex changes reflect a regime transition, we will address the polar vortex regimes in more detail in a follow-up study.

In the past, regime behavior of the tropospheric jet has been discussed (Chan and Plumb, 2009; Gerber and Polvani, 2009;

Wang et al., 2012), that led to a very strong response of the tropospheric jet to changes in the polar vortex in the original work by PK02. We find a similarly strong response of the tropospheric jet to stratospheric forcing in our model under same set-up as in PK02. In line with previous results, that have shown sensitivity of the tropospheric jet response to the tropospheric equilibrium temperatures (Chan and Plumb, 2009), we found a strongly damped poleward shift of the tropospheric jets in response to stratospheric forcing under a set-up with lower tropical upper tropospheric temperatures. We hypothesize that the

more equatorward position of the tropospheric jet in those simulations, possibly related to decreased static stability in the subtropics, leads to the damped response, but this remains to be analyzed in more detail. In simulations with topographically forced planetary waves, we find a similar tropospheric jet response to stratospheric forcing than in Gerber and Polvani (2009), as can be expected from the previously shown robustness of the tropospheric jet behavior in different model configurations (resolution, and different dynamical cores, see Wang et al., 2012). The sensitivity of the jet response to the tropical tropospheric

equilibrium temperatures is lower in the topographically forced simulations, possibly because the presence of planetary waves limits the control of the subtropical static stability on the jet location. In general, it remains to be understood to what extend the regime changes of the polar vortex are connected to regime changes of the tropospheric jet, and whether those regime changes found in the idealized models are also relevant for the real atmosphere. If this behavior occurs only in idealized models, this would put their application to advance the understanding of the jet's locations in our atmosphere into question.

Simulations with planetary wave generation by topography and by wave-like heating (as suggested by Lindgren et al., 2018) are contrasted. Generally, similar basic states can be simulated with the two different set-ups, and in both cases increases in $\gamma$ lead to increases in the polar vortex strength. However, the heating-forced simulations respond more non-linearly to increases in $\gamma$ both in terms of polar vortex strength and lower stratospheric downwelling. Furthermore, while in the topographically forced simulations, both the jet maximum in the free troposphere and near the surface move poleward with a stronger polar

vortex (in agreement with Gerber and Polvani, 2009), the wind maximum in the free troposphere remains at an almost constant latitude in the simulations with wave-like heating. Despite this almost constant location of the jet in the free troposphere, the jet is strongly displaced poleward near the surface in response to stratospheric forcing, transitioning from a regime of jet locations around 35-40°N to a regime with jet locations north of 50°N for strong polar vortex increases (similar to previously reported regime transitions, e.g., by Chan and Plumb, 2009). The prescribed wave-like heating extends throughout the free troposphere

and into the lower stratosphere, leading to the hypothesis that the prescribed heating damps the ability of the wind maximum

in the free troposphere to shift. Another possible explanation of the constant location of the free tropospheric jet is that it is located already at far higher latitudes than in the topographically forced simulations for weak polar vortices (see Fig. 11). Available observational evidence on troposphere-stratosphere coupling indicates that zonal wind anomalies usually occur in a vertically coherent manner, for example due to thermal forcing by stratospheric ozone depletion (e.g., Son et al., 2010), or in connection with SSW events (e.g., Baldwin and Dunkerton, 1999). This puts the behavior of the wave-like heating experiments into question, and further work will be necessary to understand the different behavior of troposphere-stratosphere coupling in the different model versions and set-ups. Overall, we recommend to use the thermally forced wave generation with caution.

As a first application example of the dry dynamical core model we present, as a proof-of-concept, a simulation with very basic chemistry (here only photolysis of CFCs), and the potential of such simulation set-ups to study the impact of dynamical variability and changes on tracer transport in an idealized fashion is shown. The set of chemical reactions can be expanded to the user's needs to study transport of more complex chemical tracers, such as ozone.

Secondly, we present a simulation of a monsoon-like upper tropospheric anticyclonic circulation with realistic variability forced by imposed localized heating. Such a set-up can be used to study the dynamics of diabatically forced circulation systems such as monsoon anticyclones under different forcings and background states.

With the dry dynamical core model set-up, the model hierarchy within the MESSy framework is extended by a commonly used model set-up for studying dynamical processes. With the implementation in MESSy, the tracer utilities including the possibility to consider diagnostic chemically active tracers are available in the dry dynamical core model. As a next step, we envision an expansion to account for chemistry-dynamics interaction in a simplified manner as an intermediate step between the dry dynamical core model and the full Chemistry-Climate model. This next step in constructing a consistent model hierarchy of chemistry-dynamical coupling is motivated by the research question on how circulation-induced anomalies in radiative trace gases (e.g., ozone) feed back on the dynamics. This question is relevant both on climate time-scales as well as on intra-seasonal timescales (e.g., during sudden stratospheric warmings). This extended set-up would require radiative calculations depending on the actual tracer concentrations. While this expansion of the coupled idealized set-up will be subject of future work, we note here that all necessary components are available already in the MESSy framework: the radiation scheme from the full EMAC model (Dietmüller et al., 2016) can be used with setting the input to either the online simulated values of the trace gas of interest (i.e., ozone), while the other relevant species can be set to climatological values (e.g., water vapor) or zero (e.g., clouds and aerosols). The envisioned model set-up, basically an idealized "chemistry-dynamical model", would thus consist of a dry dynamical core with thermodynamic forcing by an idealized prescribed latent heating and radiative calculations that depend on the chemical species of interest (e.g., ozone).

*Code and data availability.* The Modular Earth Submodel System (MESSy) is continuously further developed and applied by a consortium of institutions. The usage of MESSy and access to the source code is licenced to all affiliates of institutions which are members of the MESSy Consortium. Institutions can become a member of the MESSy Consortium by signing the MESSy Memorandum of Understanding. More information can be found on the MESSy Consortium Website (http://www.messy-interface.org). The code presented here has been based on MESSy version 2.54 and will be available in the next official release (version 2.55).

The data of the simulations presented in this study is freely available under http://doi.org/10.5281/zenodo.3768731

## Appendix A: Implemented functions in the RELAX submodel

### A1  Newtonian cooling

The inverse relaxation time scale $\kappa$ and the equilibrium temperature $T_{\mathrm{eq}}$ have to be specified in the model set-up via the RELAX namelist file (see Supplement). The following pre-implemented functions are available:

The functions for the 'HS' set-up, as defined by Held and Suarez (1994) but including the option of hemispheric asymmetry (as introduced by PK02), are

$$T_{\mathrm{eq}}^{\mathrm{HS}} = \max\left\{ T_0, \left[ T_1 - \delta_y \sin^2\phi - \epsilon\sin\phi - \delta_z \log\left(\frac{p}{p_0}\right)\cos^2\phi \right] \left(\frac{p}{p_0}\right)^k \right\}, \tag{A1}$$

$$\kappa = \kappa_{\mathrm{a}} + (\kappa_{\mathrm{s}} - \kappa_{\mathrm{a}})\max\left( 0, \frac{p/p_s - \sigma_{\mathrm{b}}}{1 - \sigma_{\mathrm{b}}} \right)\cos^4\phi \tag{A2}$$

where $\phi$ is the geographical latitude, $p$ is the actual pressure, $p_s$ is the current surface pressure and $k = R/c_p = 2/7$. All constants can be set via namelist entries, with defaults set to the values given in HS94 (see Supplement, Table 1 for description of parameters and default values). The parameter $\epsilon$ sets the hemispheric asymmetry, and its sign is controlled by the namelist parameter $h_{\mathrm{fac}}$. If $h_{\mathrm{fac}}$ is zero, the equilibrium temperature is symmetric between the hemispheres (i.e., $\epsilon = 0$). If $h_{\mathrm{fac}} \neq 0$, then

$$\epsilon = \mathrm{sign}(h_{\mathrm{fac}}) * |\epsilon| \tag{A3}$$

i.e., the sign of $h_{\mathrm{fac}}$ determines which hemisphere is the winter hemisphere (positive $h_{\mathrm{fac}}$: northern hemispheric winter, negative $h_{\mathrm{fac}}$: southern hemispheric winter).

The inverse relaxation time scale in the PK set-up is identical to that in 'HS' set-up. The equilibrium temperature in the PK set-up is similar to the one of HS in the troposphere, but uses the following function in the stratosphere above a given transition pressure $p_{\mathrm{T}}(\phi)$:

$$T_{\mathrm{equ}}^{\mathrm{PK}}(\phi,p) = \begin{cases} \max\left\{ T_{\mathrm{US}}(p_{\mathrm{Ts}}), \left[ T_1 - \delta_y \sin^2(\phi) - \epsilon\sin(\phi) - \delta_z \log\left(\frac{p}{p_0}\right)\cos^2(\phi) \right]\left(\frac{p}{p_0}\right)^k \right\} & \text{for } p \geq p_{\mathrm{T}}(\phi) \\ [1 - W(\phi)]\, T_{\mathrm{US}}(p) + W(\phi)\, T_{\mathrm{US}}(p_{\mathrm{Ts}})\left(\frac{p}{p_{\mathrm{T}}(\phi)}\right)^{\frac{R\gamma}{g}} & \text{for } p < p_{\mathrm{T}}(\phi) \end{cases} \tag{A4}$$

The stratospheric temperature profile is based on the US standard atmosphere temperature profile $T_{US}(p)$ in the summer hemisphere (USA, 1976) and exhibits a temperature decrease with lapse rate $\gamma$ in the winter hemisphere representing the region of the polar vortex. This transition is performed by the weighting function

$$W(\phi) = \frac{1}{2}\left[1 + \text{sign}(h_{\text{fac}})\tanh\left(\frac{\phi - \phi_0}{\delta\phi}\right)\right].\tag{A5}$$

The transition latitude $\phi_0$ is set, similar to $\epsilon$, to $\phi_0 = \text{sign}(h_{\text{fac}}) * |\phi_0|$. The smooth transition between tropospheric and stratospheric temperatures is ensured by bounding the tropospheric temperature to the temperature in the transition region $T_{US}(p_{Ts})$.

As an extension to the original PK set-up, we include the possibility to vary the transition pressure from summer to winter hemisphere, using the weighting function $W(\phi)$:

$$p_T(\phi) = (p_{Tw} - p_{Ts})W(\phi) + p_{Ts}\tag{A6}$$

where $p_{Ts}$ and $p_{Tw}$ are the transition pressures over the summer and winter hemisphere, respectively. Again, all constants can be set in the namelist with default values that correspond to the original PK02 set-up ( i.e., with constant transition pressure $p_T(\phi) \equiv 100\,\text{hPa}$), as detailed in the Supplement (Table 1).

## A2  Rayleigh Friction

The following implemented functions are available for setting the horizontal wind damping coefficient $k_{\text{damp}}$:

1) Damping of the surface layer as specified by HS94 (option 'HS'):

$$k_{\text{damp}} = k_{\text{max}}^{\text{HS}}\max\left(0, \frac{\frac{p}{p_s} - \sigma_0}{1 - \sigma_0}\right)\tag{A7}$$

with default values $k_{\text{max}}^{\text{HS}} = 1.16 \times 10^{-5}\,\text{s}^{-1}$, $\sigma_0 = 0.7$ and $p_s$ the current surface pressure.

2) Damping of a layer at the model top as specified by PK02 (option 'PK'):

$$k_{\text{damp}} = \begin{cases} 0 & \text{for } p > p_{\text{sp}} \\ k_{\text{max}}^{\text{PK}}\left(1.0 - \frac{p}{p_{\text{sp}}}\right)^2 & \text{for } p \leq p_{\text{sp}} \end{cases}\tag{A8}$$

with default values $k_{\text{max}}^{\text{PK}} = 2.3148 \times 10^{-5}\,\text{s}^{-1}$ and $p_{\text{sp}} = 0.5\,\text{hPa}$.

3) Damping of a layer at the model top with the function as implemented in the original ECHAM code (option 'EH'):

$$k_{\text{damp}} = \begin{cases} 0 & \text{for } i_{\text{lev}} > i_{\text{lev}}^{sp} \\ k_{\text{drag}}c^{i_{\text{lev}}^{sp} - i_{\text{lev}}} & \text{for } i_{\text{lev}} \leq i_{\text{lev}}^{sp} \end{cases}\tag{A9}$$

where $i_{\text{lev}}$ is the number of the hybrid level counted from the top of the model for a vertical resolution of L90MA. Thus, the drag $k_{\text{drag}}$ is enhanced by a factor of $c$ for each level going upward. Default values are $c = 1.5238$, $k_{\text{drag}} = 5.02 \times 10^{-7}\,\text{s}^{-1}$ and $i_{\text{lev}}^{sp} = 10$, corresponding to a pressure of 0.43 hPa for the L90MA vertical resolution. If the model is run at a different vertical resolution, the damping coefficients are first calculated according to Eq. A9 for L90MA within the routine, and then interpolated to the current vertical levels. The equivalent pressure levels can be found in the data set that accompanies this publication (see data availability section).

## A3 Diabatic heating

The implemented heating function for the zonal mean heating (*tteh_cc_tropics*), as given by (Butler et al., 2010) reads

$$Q_0(\lambda, \phi, p) = q_0^{cc} \exp\left[-\frac{1}{2}\left(\frac{\phi - \phi_0^{cc}}{\delta_\phi^{cc}}\right)^2 - \frac{1}{2}\left(\frac{p/p_s - \sigma_z^{cc}}{\delta_z^{cc}}\right)^2\right] \tag{A10}$$

with $p_s$ being the surface pressure and default values are set to those by Butler et al. (2010) (see Supplement, Table 3).

The temperature tendency *tteh_waves*, used here for the generation of planetary waves introduced by Lindgren et al. (2018), reads

$$Q_0(\lambda, \phi, p) = \begin{cases} q_0^w \sin(m\lambda) \exp\left[-\frac{1}{2}\left(\frac{\phi - \phi_0^w}{\delta_\phi^w}\right)^2\right] \sin\left(\pi \frac{\log(p/p_\text{bot})}{\log(p_\text{top}/p_\text{bot})}\right) & \text{for } p_\text{top} \leq p \leq p_\text{bot}, \\ 0 & \text{otherwise} \end{cases} \tag{A11}$$

where $\lambda$ is the geographical longitude, and all parameters are set to default values as used by Lindgren et al. (2018), see Supplement Table 3.

The function describing the localized heating field, *tteh_mons*, is given as:

$$Q_\text{loc}(\lambda, \phi, p, t) = Q_\text{temp}(t) Q_\text{pres}(p) Q_\text{lat}(\phi) Q_\text{lon}(\lambda). \tag{A12}$$

Here, the individual factors are used to describe the temporal and spatial (horizontal and vertical) dependence of the heating function. The temporal evolution of the heating is given by:

$$Q_\text{temp}(t) = \begin{cases} \frac{t}{t_s^m} \times (q_0^m + q_\text{temp}^m sin(2\pi \frac{t}{\delta t^m})) & \text{for } 0 \leq t \leq t_s^m, \\ 1 \times (q_0^m + q_\text{temp}^m sin(2\pi \frac{t}{\delta t^m})) & \text{otherwise.} \end{cases} \tag{A13}$$

To slowly increase the heating after the start of the simulation a spin up factor of $\frac{t}{t_s^m}$ is included until the end of the spin up time $(t_s^m)$. After the spin up time $(t_s^m)$ the temporal variation of the heating is only given by a periodic oscillation (period $\delta t^m$) with amplitude $(q_{temp}^m)$ around a constant base heating $(q_0^m)$.

In the vertical the heating is assumed to be of the same form as in Eq. (A11), i.e.:

$$Q_\text{pres}(p) = \begin{cases} sin(\pi \frac{log(p/p_\text{bot}^m)}{log(p_\text{top}^m/p_\text{bot}^m)}) & \text{for } p_\text{top}^m \leq p \leq p_\text{bot}^m, \\ 0 & \text{otherwise.} \end{cases} \tag{A14}$$

Here, $p_{bot}^m$ and $p_{top}^m$ denote the maximum and minimum pressure to which the heating is confined in the vertical. The latitudinal dependence for $\phi \in [-90, 90]$ follows the function suggested by Schubert and Masarik (2006, their Eq. 4.1), and is given as

$$Q_\text{lat}(\phi) = exp\left(-\left(\frac{\phi - \phi_0^m}{\delta\phi^m}\right)^2\right) \tag{A15}$$

Finally, the longitudinal dependence for $\lambda \in [0, 360)$ is given by

$$Q_{\text{lon}}(\lambda) = \begin{cases} 0.5(1 + cos(\pi \frac{g(\lambda, \lambda_0^m)}{\delta \lambda^m})) & \text{if } g(\lambda, \lambda_0^m) \leq \delta \lambda \\ 0 & \text{otherwise} \end{cases} \tag{A16}$$

where $g(\lambda, \lambda_0^m) = min((\lambda - \lambda_0^m) \; mod(360), (\lambda_0^m - \lambda) \; mod(360))$ and the modulo function $mod(360)$ maps $\mathbb{R}$ to $[0, 360)$, i.e., the function returns the smallest angle between the longitude $\lambda$ and the central longitude $\lambda_0^m$ with accounting for the crossing of the $0°$ line. Again the longitudinal function is based on the heating described by Schubert and Masarik (2006, their Eq. 4.1). However, as Schubert and Masarik (2006) were aiming to investigate the Madden-Julian-Oscillation, they included a movement of the localized heat source, which we do not include here (i.e., we use their equation with propagation speed 0). Overall this heating structure is similar to other idealized heatings used for studying monsoon anticyclones (e.g., Siu and Bowman, 2019).

## Appendix B: List of simulations

In Table B1, a list of all simulations presented in this study is given with details on their set-up, resolution and simulation length. The simulations without planetary wave forcing are labeled "WN0", the ones with topographic wavenumber 2 wave-forcing with "WN2T", the ones with diabatic wave forcing with "WN2H". The values of the winter transition pressure $p_{\text{Tw}}$ are given in hPa, and the values of the polar vortex lapse rate $\gamma$ in K km-1. "npv" refers to simulations with "no polar vortex", i.e., the summer stratospheric equilibrium temperature is extended to the winter pole. The values of the parameter in control of the tropical vertical temperature gradient in the troposphere ($\delta_z$) are given in K. As stated in Sec. 2, the reduced value of $\delta_z$ = 10 K /log(10) = 4.34 K in a number of simulations is the result of a previous implementation of the term $\delta_z \log \left( \frac{p}{p_0} \right) \cos^2 \phi$ in Eq. A1 as logarithm to base 10 instead of the natural logarithm. Note that in the accompanied data set (see data availability), the simulations are labeled accordingly as "ln" versus "log10". The upper atmospheric damping coefficients are set to the formulation by PK (see Eq. A8) or to the formulation EH (see Eq. A9). The total length of the simulations is given in the table in days, with the number of analyzed days given in the Figure captions.

## Appendix C: Probability distributions of polar vortex strength and tropospheric jet location

The probability distributions of the polar vortex strength (maximum zonal mean zonal wind at 10 hPa) and the position of the tropospheric jet (latitude of zonal mean zonal wind maximum at 850 hPa and 500 hPa) are shown for a number of selected experiments: for the original PK02 set-up, and the equivalent reduced $\delta_z$ set-up (Fig. C1), for the simulations with a WN2 topography with differing $p_{\text{Tw}}$ (Fig. C2), and for simulations with a WN2 diabatic heating for the standard and reduced $\delta_z$ set-up (Fig. C3).

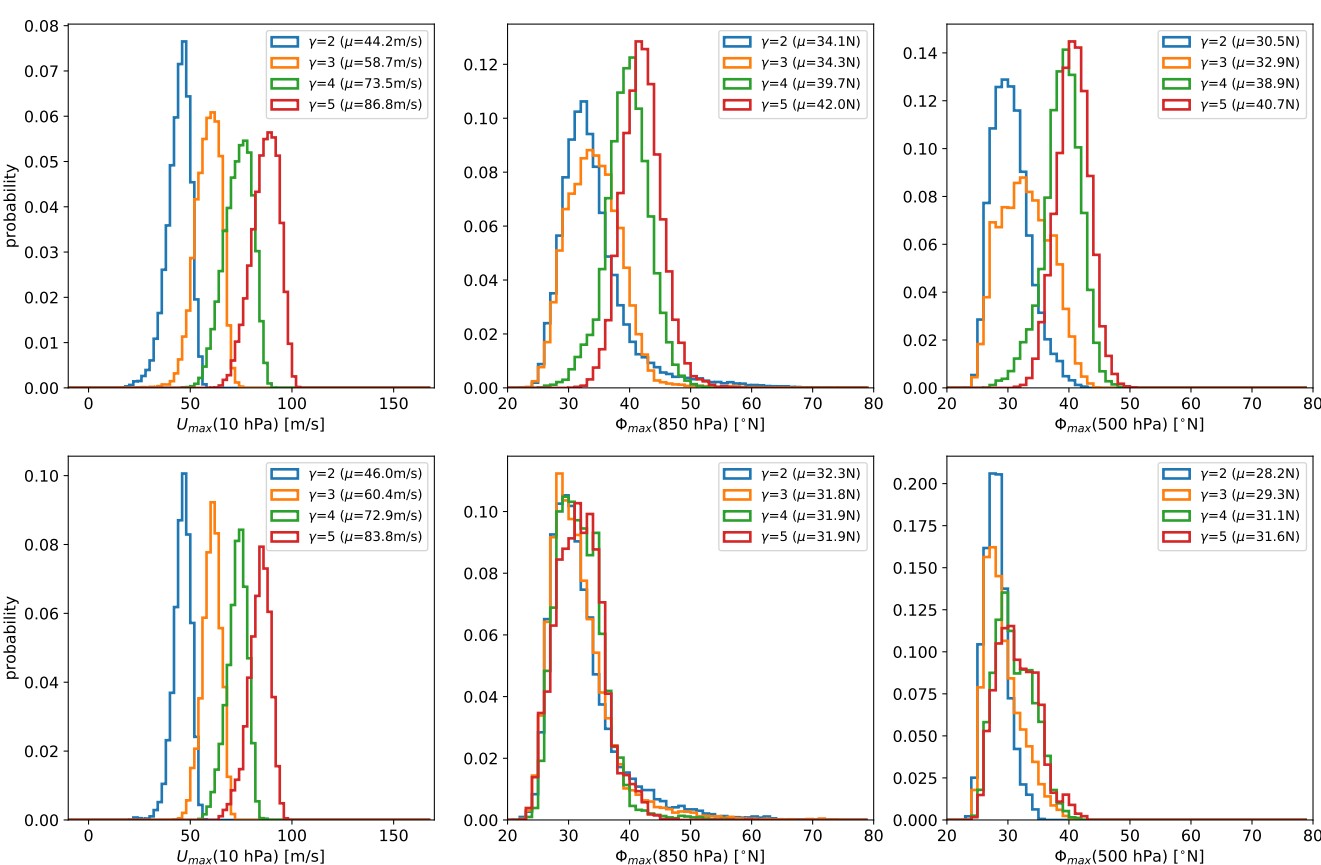

**Figure C1.** Probability distributions of (left) the maximal zonal mean zonal wind at 10 hPa, (middle) the latitude of the maximum of zonal mean zonal winds at 850 hPa, and (right) at 500 hPa for simulations without planetary wave forcing and (top) for the original PK02 set-up, and (bottom) for the modified set-up with reduced $\delta_z$. The mean value of the distribution is given in the legend (denoted as $\mu$).

**Table B1.** List of all simulations presented in this study, with information on all relevant parameter settings and variations, and simulation length in days (see text for details). Simulations with multiple values of $\gamma$ are listed in one row for brevity. * also $p_{Ts}$=200 hPa; # Including additional localized heating with parameter settings given in Sec. 5.2.

| Section/ Figure | WN | PK/HS | $p_{Tw}$ | $\gamma$ | $\delta_z$ | upper sponge | length | resolution |
|---|---|---|---|---|---|---|---|---|
| 3.1 / 3 | WN0 | HS ($\epsilon = 0$) | - | - | 10 | - | 1825 | T63L19 |
| 3.1 / 3 | WN0 | HS ($\epsilon = 0$) | - | - | 10 | - | 1825 | T42L90MA |
| 3.2 / 4, 7 | WN0 | PK ($\epsilon = 10$) | 100 | 4 | 10 | PK | 10957 | T42L90MA |
| 3.2 / 4, 5 | WN2T | PK ($\epsilon = 10$) | 100 | 4 | 10 | PK | 10534 | T42L90MA |
| 4.2.1 / 8, 11, C1 | WN0 | PK ($\epsilon = 10$) | 100 | [npv,1,2,3,4,5] | 10 | PK | 10957 | T42L90MA |
| 4.2.1 / 8, 11, C1 | WN0 | PK ($\epsilon = 10$) | 100 | [npv,1,2,3,4,5] | 4.34 | PK | 3652 | T42L90MA |
| 4.1 / 7 | WN0 | PK ($\epsilon = 10$) | 100 | 4 | 10 | EH | 1825 | T42L90MA |
| 4.2 / 8, 11 | WN2T | PK ($\epsilon = 10$) | 100 | [1,2,3,4] | 4.34 | EH | 1825 | T42L90MA |
| 4.2 / 8 | WN2T | PK ($\epsilon = 10$) | 150 | [1,2,3,4] | 4.34 | EH | 1825 | T42L90MA |
| 4.2 / 8 | WN2T | PK ($\epsilon = 10$) | 200 | [1,2,3,4] | 4.34 | EH | 1825 | T42L90MA |
| 4.2 / 8 | WN2T | PK ($\epsilon = 10$) | 250 | [1,2,3,4] | 4.34 | EH | 1825 | T42L90MA |
| 4.2 / 8 | WN2T | PK ($\epsilon = 10$) | 300 | [1,2,3,4] | 4.34 | EH | 1825 | T42L90MA |
| 4.2 / 8 | WN2T | PK ($\epsilon = 10$) | 350 | [1,2,3,4] | 4.34 | EH | 1825 | T42L90MA |
| 4.2 / 8, 11 | WN2T | PK ($\epsilon = 10$) | 400 | [1,1.5,2,2.5,3,4] | 4.34 | EH | 1825 | T42L90MA |
| 4.2 / 8 | WN2T | PK ($\epsilon = 10$) | 450 | [1,2,3,4] | 4.34 | EH | 1825 | T42L90MA |
| 4.2 / 8, 9, 11, C2 | WN2T | PK ($\epsilon = 10$) | 100 | [1,2,3,4] | 10 | EH | 10957 | T42L90MA |
| 4.2 / 5, 8, 9, 11, C2 | WN2T | PK ($\epsilon = 10$) | 400 | [1,1.5,2,2.5,3,4] | 10 | EH | 10957 | T42L90MA |
| 4.3 / 8 | WN2H | PK ($\epsilon = 0$) | 200* | 4 | 4.34 | EH | 1825 | T42L90MA |
| 4.3 / 8, 11, C3 | WN2H | PK ($\epsilon = 0$) | 400 | [1,1.5,2,2.5,3,3.5,4,4.5] | 4.34 | EH | 1825 | T42L90MA |
| 4.3 / 8 | WN2H | PK ($\epsilon = 0$) | 450 | [1,1.5,2,2.5,3,3.5,4,4.5] | 4.34 | EH | 1825 | T42L90MA |
| 4.3 / 8, 10 | WN2H | PK ($\epsilon = 0$) | 200* | 4 | 10 | EH | 10957 | T42L90MA |
| 4.3 / 8, 10, 11, C3 | WN2H | PK ($\epsilon = 0$) | 400 | [1,2,3,4] | 10 | EH | 10957 | T42L90MA |
| 5.1 / 12, 13 | WN2T | PK ($\epsilon = 10$) | 400 | [1,2,3] | 4.34 | EH | 3285 | T42L90MA |
| 5.2 / 14, 15 | WN0 # | HS ($\epsilon = -10$) | - | - | 10 | - | 1095 | T42L90MA |

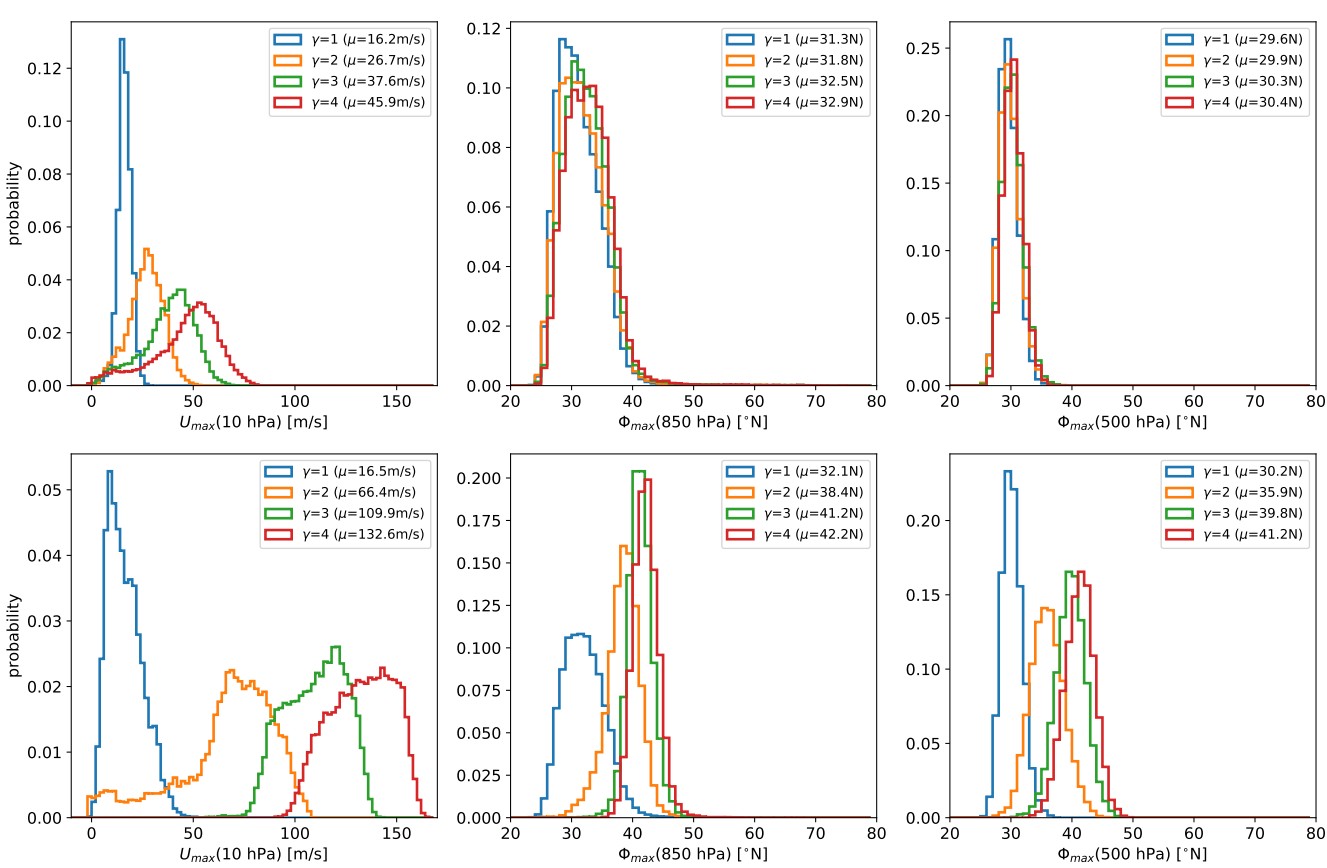

**Figure C2.** As Fig. C1, but for simulations with standard PK set-up and WN2 topography (top) for $p_{\text{Tw}}$= 100 hPa, and (bottom) for $p_{\text{Tw}}$= 400 hPa.

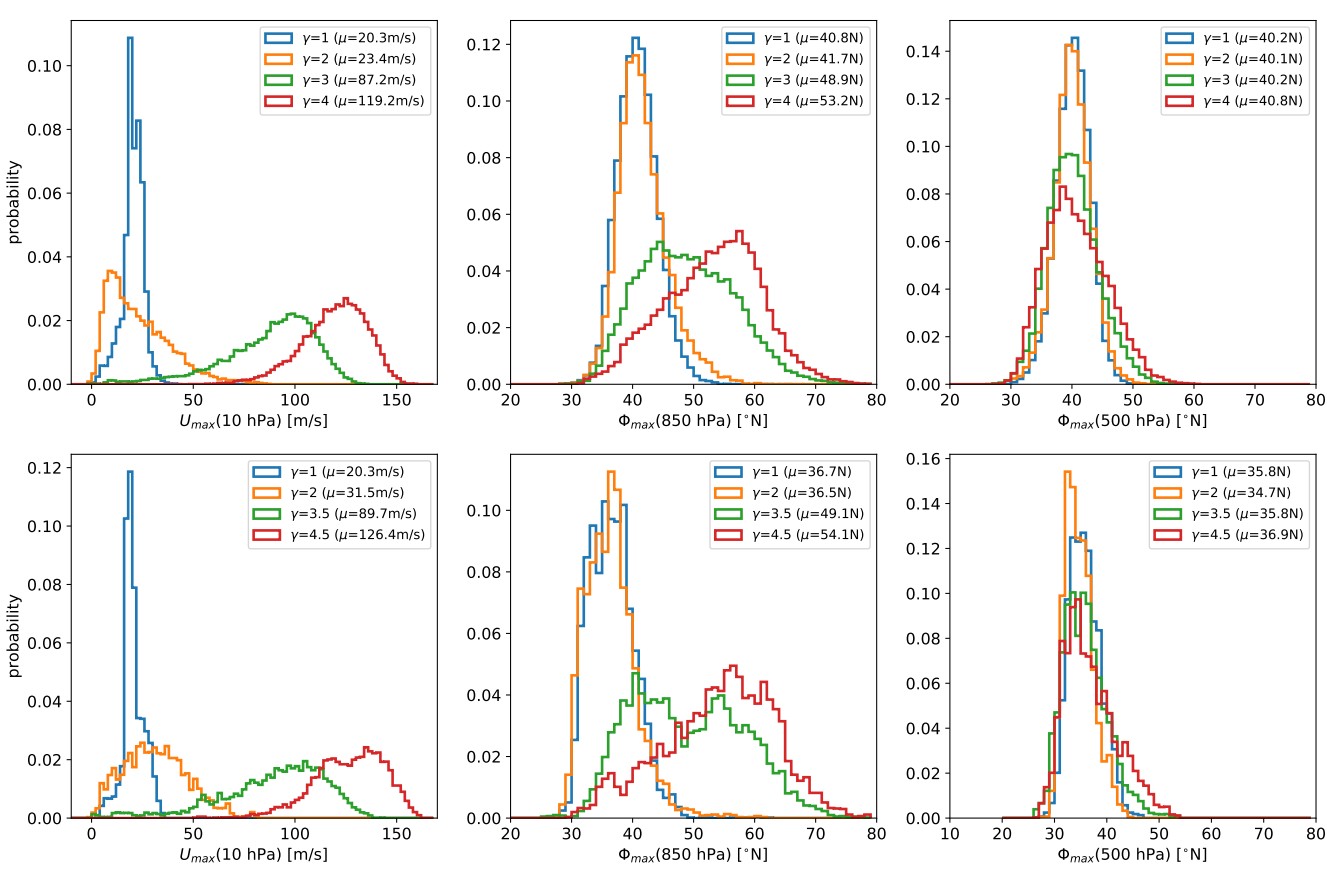

**Figure C3.** As Fig. C1, but for simulations with WN2 heating and $p_{\mathrm{Tw}}= 400\,\mathrm{hPa}$ for (top) the standard set-up and (bottom) the modified set-up with reduced $\delta_z$.

*Author contributions.* HG designed and performed the implementation of the submodel RELAX, performed the test simulations and wrote large parts of the paper. RW strongly contributed to the implementation and conducted, analyzed and described the sensitivity experiments. MN implemented the localized heating function and analyzed and described the monsoon experiments. TB initiated the design of the sensitivity simulations. All authors contributed to the writing of the paper.

5 *Competing interests.* The authors hereby declare that they do not have conflicting interests.

*Acknowledgements.* HG and RW were funded by the Helmholtz Association under grant VH-NG-1014 (Helmholtz-Hochschul- Nachwuchsforschergruppe MACClim). MN received funding from the Initiative and Networking Fund of the Helmholtz Association through the project "Advanced Earth System Modelling Capacity (ESM). The simulations have been performed at the German Climate Computing Centre DKRZ through support from the Bundesministerium für Bildung und Forschung (BMBF). Data was processed using CDO (Climate Data Operators;
10 Schulzweida, 2019). For parts of the data analysis and plotting NCL (NCAR Command Language; NCL, 2018) has been used. The SPARC climatologies are available at ftp://sparc-ftp1.ceda.ac.uk/sparc/ref_clim/randel/temp_wind/. We thank Patrick Jöckel for useful discussion and comments, and Philip Rupp for helpful discussions on idealized modelling of monsoon anticyclones. We thank Penelope Maher, one anonymous reviewer and in particular Edwin Gerber for their very valuable comments on the previous manuscript version.

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
