# Peer review of "Extending the Modular Earth Submodel System (MESSy v2.54) model hierarchy: The ECHAM/MESSy idealized (EMIL) model set-up"

_Geoscientific Model Development, 2019_

## Referee Comment (RC1) · Edwin Gerber (Referee) · 17 Jan 2020

The authors document a new idealized model configuration within the ECHAM/MESSy modeling framework, and demonstrate how it can be used to investigate open questions in the climate sciences, namely chemistry-transport interactions and the monsoonal circulation. I believe that this work is timely and important, and would be of interest to GMD readers. I therefore recommend publication pending consideration of the comments/suggestions below. As my identify might be obvious given my familiarity with the system, I'm signing this review. Ed Gerber

General comments

1) The authors compare the performance of EMIL against a number of benchmark cases that are in the literature. It would be ideal, however, if we could move beyond the "picture norm" for these comparisons – at least in the future. Could you publish the data for these results (or incorporate it within the ECHAM-MESSy distribution), so that in the future, other groups could check their models against yours? The best standard would be to determine whether your integrations are consistent/inconsistent with other benchmark integrations, within the sampling uncertainty. I believe that data can be archived through Zenodo.org, or other structures. You could just provide the zonal mean time mean data needed for the figures.

Another option would be to include the key benchmarks as test cases within ECHAM/MESSy, something that could easily be reproduce by another group. Could you provide a citable link to the model and the required parameter scripts? (That is, a frozen version of the model, as was used to produce this paper, ideally with the same run scripts that you used.) I appreciate that the supplement provides all the parameters, but it would still involve a lot of work (and hence many chances to make a mistake) to reproduce this exactly.

2) I appreciate that the authors have striven to find a balance between detailing a new model set-up for others to use, and presenting new results. I felt that the test cases that were shown at the end in section 5 were very interesting, but could have been more developed. To provide more space, perhaps the earlier sections could be condensed? (The reader might also be a bit exhausted by the time they reach these really interesting results!)

For example, there are a lot of equations and parameters defined in this study, many which are specified in other papers (but also many of which are new). I think some of this detail could best be put in an appendix (e.g., in sections 2.1.1, 2.1.2 and 2.1.3), allowing you to move more quickly to the results.

3) It would help the reader to have a table that defines all the parameters in one place.

It would also help you catch any parameters that are multiply defined. One example is $k\_max$, which appears in the equations (8) and (9) with distinct values. $k\_damp$ is also defined inconsistently between these two equations (though any reasonable reader would understand what is meant). The parameter $\delta\phi$ also appears in multiple equations, e.g., (5) and (16).

Finally, I noticed that $\sigma$ is sometimes used to refer to a vertical coordinate (p/ps), and at other times used width (where I appreciate the motivation is to connect it to the variance of a Gaussian). It might be good to adopt a consistent notation, where $\delta$ is always used for width parameters – but again watching out to make sure all parameters are uniquely defined. (This said, I know that these parameters came from multiple papers in the literature, where the other authors were not consistent with each other!)

4) The paragraph spanning from page 3 line 28 to page 4 line 2 is very interesting, but seems out of place in the introduction. I would consider pushing this to final section, where you could present it as the next step in your research program.

5) Finally, the topic of regimes comes up quite prominently in section 4. I think this is a very interesting (albeit sometimes frustrating) result that could be mentioned in the abstract and introduction. I think these regimes have simmering in idealized models for sometime: as detailed by Gerber and Polvani (2009), the original PK02 result is so dramatic precisely because of a regime switch between their $\gamma$ 2 and 4 integrations. Chan and Plumb (2009, DOI: 10.1175/2009JAS2937.1) and Wang et al. (2012) discuss this in more detail.

The presence of regimes is interesting: if such a thing existed in our atmosphere, we could be in for surprises with global warming (or perhaps when the planet enters an ice age). If it is an artifact of these idealized models, however, it's something that the dynamics community should be wary of. It could lead to unphysical parameter sensitivity or results that are qualitatively disconnected from the real atmosphere, breaking

the link we'd hope to establish through model hierarchies.

Specific comments (largely typographical) by page:line number

1:1 Consider "As models of the Earth system grow in complexity, a need emerges to connect them with simplified systems through model hierarchies in order to improve process understanding."

1:3 consider cutting "with the aim"

1:6 Would you consider ECHAM/MESSy a "model", or rather a "framework" which allows you to build many different models.

1:10 Consider "Test similations with EMIL reproduce benchmarks provided by earlier dry dynamical core studies."

1:19 What do you mean by "the ability to simulate dynamical systems"? Dynamical systems in the broadest sense is a whole field in mathematics. Perhaps you mean "the ability to simulate qualitatively realistic dynamical variability of the circulation"

1:22 Consider something like "Earth system models continue to incorporate more processes to enable a more complete simulation of the climate system, and thus produce the best possible climate projections. In practice, this increases the complexity of model codes as new compartments are added to represent new processes."

I'm not sure if you need that second sentence; my thought was that the goal is to increase the range of processes that are simulated, and this is effected in practice by adding more compartments, modules, etc..

2:9 stray space: "hereafter) ."

2:13 I think the upper level drag is only in the PK02 set up, and not a part of the original HS94 configuation.

2:16 consider a paragraph break before "The functions..."

2:21 "to idealized heating that mimics the thermal response to CO2 increase" I think "climate change" is the response, not the forcing!

2:26 "motivates one to include"

2:29 Jucker and Gerber (2017) were not the first/only one to do this. Consider also referencing:

Merlis, T. M., T. Schneider, S. Bordoni, and I. Eisenman, 2013: Hadley circulation response to orbital precession. Part I: Aquaplanets. J. Climate, 26, 740–753, doi:10.1175/ JCLI-D-11-00716.1.

Tan, Z., T. A. Shaw, and O. Lachmy, 2019: The sensitivity of the jet stream response to climate change to radiative assumptions, J. Advan. Mod. Earth Sys., 10.1029/2018MS001492.

2:35 Here and throughout the text, the quotes seem to be reversed. Perhaps this is set by the journal, but I am used to "hello" as opposed to "hello"

3:19 consider "allows the creation of model hierarchies"

3:20 consider "Earth-system model. Any developments..."

5:9 I found "idealzied localize contrained" to be awkward. Consider just "forced by a simple, localized heating that..."

eqn (1) In HS94 and other papers, it's usually just T_{eq}

5:30 This was a point where I feel you've lost the balance on providing enough technical advice without making the paper too long. Do you need to describe an option that "physicaly of little use"

eqns (204) To make the paper more concise, you could refer the reader to HS94. I appreciate that equation (2) is modified by the inclusion of the \epsilon sin(\phi) term; this was documented by equations A3 and A4 in PK02. A happy medium might

be to reference past work in the paper, highlighting your modifications, and including equations in an appendix.

5:19 $T_{US}$ isn't defined in the paper. The reference is: U.S. Standard Atmosphere, U.S. Government Printing Office, Washington, D.C., 1976. (Which I appreciate isn't so easy to find!)

eqn (6) Aditi Sheshadri did something like this in her 2015 paper, https://doi.org/10.1175/JAS-D-14-0191.1. There she lowered the start of the vortex to 200 hPa. That said, I appreciate the more thorough investigation of the transition height in this study!

Figure 3 and surrounding discussion. It is interesting that the jets shift equatorward when you move from the T63L19 to the T42L90 integrations. I suspect the vertical resolution plays a more important role her than you might suspect. This is consistent with the behavior of GFDL's spectral core, where the jets also shift equatorward when the vertical resolution is increased. See Fig. 4 of Gerber et al. (2008), https://doi.org/10.1175/2007MWR2211.1. This doesn't seem to happen in finite difference or finite volume based cores. [This said, I don't mean for you to add another citation; I think you've already been very generous in referencing my past work.]

11:13 consider a paragraph break after PK02.

11:14 (namely GFDL's spectral dynamica core)

Figure 5: the caption on this figure could be expanded to help a reader who's skimming the paper, for instance, defining the key parameters $p_{Tw}$ and $\gamma$ that are being used. I'll admit I had to remind myself what $p_{Tw}$ represented.

13:16 Along the lines of my general comment on the "picture norm", it would be ideal to be more precise about what you meant by negligible. I think you mean that it is small relative to uncertainties in the cliamatology with resolution (i.e., T63L19 vs. T42L90), but you could also define it relative to sampling uncertainty (i.e., it would take inordinately long integrations for the difference to be significant above sampling noise.)

Figure 6 and discussion. I appreciated this portion of the paper, but a a quick question: is one month of austral hemisphere gravity wave drag enough to nail down the effective damping rate in models? I don't have a good sense how much this rate varies. I assume this includes both orographic and non-orographic drag? Would the effective rate be much different in the boreal hemisphere during winter?

I think it would help to expand the caption, to explain that GWD/u provides an effective damping time scale of the winds when using a full gravity wave drag scheme.

15:4 "cannot"

Figure 8 and following figures. You could possible color the dashed curves which show the equilibrium profiles, to make the comparison with their respective \gamma's easier. For Figure 8 specifically, please specify the location of this profile. Is it right at the pole?

16:7 consider a paragraph break after \gamma.

17:3 In Wang et al. (2012), I think we had to grapple with this same regime behavior. The model switch abruptly from a state with active stratospheric variability and a strong residual mean circulation (which allows the temperature to deviate substantially from T_eq) to a state with an very cold, stable vortex near "radiative" equilibrium. In Wang et al., this regime change was associated with a substantial change in the position of the tropospheric jet. Does that happen here?

17:28 "these two simulations"

Figure 10 Here you are showing results from integrations which exhibit multiple regimes. Based on past experience (e.g., Wang et al. 2012), regime transitions can introduce very long time scales, as the model switches between states. You can see this of this Figure 5 of your text, which corresponds to pTw=400, gamma 2 integration shown in the right panel (I think.) Therefore, you have to be very careful in establishing convergence. Earlier in the text you suggested that runs were done for 1825 days; it

[Figure]

seems that you have longer runs (3000 days are shown in Fig. 5), but I'm not sure that would be sufficient. It would be good to check/comment on the sampling uncertainty in these climatologies.

Figure 12 and discussion. I suspect that the strength of the overturning (difference between T and T_eq) near the model top will be dominated by the drag layer. Hence, it's likely to be determined by \gamma: if you force a stronger vortex, you need a stronger drag. At lower layers, the strength of overturning is dominated by "wave pumping", and so the resolved circulation.

I worried about this a lot in preparing my 2012 paper, but convinced myself that in the mid-to-upper stratosphere, the differences in the residual circulation in response to changing gamma were still being dominated by the waves, and so not an artifact of the sponge layer. I'm not exactly sure how far down you need to go to be free of the sponge layer, but perhaps 10 hPa would be a better choice than 1 hPa? This would be supported by Figure 7, where you find that the spong layer has a negligible impact below 10 hPa. I'd also be curious to see if the nonlinearity in the vortex shown in Figure 11, bottom left, shows up in the overturning at 10 hPa in the model with heating.

22:18 consider a paragraph break after "high."

22:19 consider "high latitudes (north of 60N), driven by the strong wave dissipation that effected the SSW; see the red line in the top panel of Fig. 13. This transports ..."

22:22 consider "latidues, evident in Fig. 13 ... 15 ms-." (no paratheses). I'd also consider breaking the paragraph after this sentence.

22:29 Isolated from what? Consider cutting "in an isolated manner," or to be more specific, e.g., "independent of the annual cycle" or "isolated from all other chemical processes".

Figure 13 Consider reworking the caption, as you first refer to the middle panel.

It might also be nice to include a second axis on the top panel, or to make "w*[10ˆ-5

hPa]" in red

23:1 Consider a paragraph break after "is steep."

23:3 "downwelling is maxized at the"

23:4 same as above

24:10 consider a pargarph break after "state."

24:17 10ˆ20 J sounds like a lot, but could you provide some context? Say, what is the effective heating rate per square meter (W/mˆ2), which could be more easily compared to solar or precipitation forcing. With hope this number is in the ball park for what you'd expect from monsoon precipitation.

24:19 "produced in response to the additional heating"

24:23 consider a paragraph break after "respectively.)"

24:26 Perhaps the anticyclonic centers could be marked/labeled in the figure.

24:30 You could break the paragraph after "2016).

24:30 Consider. "An example of eastward eddy shedding was observed during the second period, as displayed on the right of Fig. 16. This phenomenon has been previously investigated..."

25:7 Your summary opens with a hard sentence to parse. Consider from line 8"... model system is documented. The set-up, denoted EMIL (explain the acronym), is shown to perform consistently with established dry dynamical core benchmarks, both earlier configurations of the ECHAM core, and those developed by other modeling centers."

25:26 "used setups. The polar"

26:1 This is an interesting result, as we see this coupling in observations (i.e., with the ozone hole, or following an SSW). It is my understanding that the tropospheric state

of the Lingren et al. (2018) model is substantially different, and might explain why does not couple to the stratosphere. As you have shown in Figure 10 (right panel), for instance, easterlies are generated in the UTLS region of the winter hemisphere.

26:3 consider "we present, as a proof-of-concept, a"

---

## Short Comment (SC1) · 30 Jan 2020

Dear authors,

as a GMD executive editor checking the manuscript for compliance with the GMD rules and as a MESSy user and developer myself, I am really confused by the MESSy version number provided in the title.

The MESSy user and developer knows, (1) that MESSy v2.55 is not yet released (not even completely fixed at the time of publication of the discussion version of your article) and (2) that your developments will be part of the v2.55 release.

[Figure]

So the code availability section is correct, but MESSy v2.55 could not have been used to produce results. Attentive reading of the title leads to the recognition, yes, the extension of the MESSy model hierarchy is available from MESSy v2.55 on.

Unfortunately, the title promotes the misconception, that the results of the article are produced with MESSy v2.55, what is not yet possible. The introduction of Section 2 further strengthens this wrong impression. Therefore the version number in the title and in the introduction of section2 needs to be "based on v2.54". While in the code availability section the addition important information is "available in MESSy v2.55" (as is already the case).

Note, that the exact code version(s), with which you produced the results shown in this article need(s) to be permanently archived.

Best regards, Astrid Kerkweg
* * *

---

## Author Comment (AC1) · 31 Jan 2020

Dear Astrid Kerkweg,

thank you for your comment on our paper, and thank you for pointing out the misconception on the version numbering of the MESSy versions.

Indeed, you are absolutely right that it should read "based on MESSy v2.54" in the title and the text. The misconception arose from the fact that the new developments will only be available in the next version 2.55 (so my thinking was that a user looking to apply the developments described here would want to use this version, once released).

But of course the development is based on the existing version 2.54 (as we agree is documented correctly in the code availability section), and you are right in that we should not refer to a version that is not released yet. I apologize for the mistake from my side, and will make sure to correct the version number in the revised version of the paper. The code version the results presented in the paper (i.e. the version based on v2.54) is archived and available on request.

Best regards, Hella Garny on behalf of all authors

---

## Referee Comment (RC2) · Penelope Maher (Referee) · 11 Feb 2020

**Review of 'Extending the Modular Earth Submodel System (MESSy v2.55) model hierarchy: The ECHAM/MESSy idealized (EMIL) model set-up'**

*by Hella Garny, Roland Walz, Matthias Nützel, and Thomas Birner*

GMD manuscript number: gmd-2019-330. *Recommendation*: Major Revision

*Reviewer Name*: Penelope Maher

**1 Summary of the Review**

This manuscript describes the implementation of the Held–Suarez configuration, with the Polvani–Kuchner amendment for the stratosphere, within the ECHAM/MESSy modelling framework. From the model description, it seems the model has been implemented in a modular nature which is a credit to the modelling effort (this can be a development nightmare otherwise). The manuscript has a well described parameterisation equation set, and has tested the relevant parameter spaces for the tunable variables and compares their results with the literature. The new model set-up is then used, as a proof of concept, for looking at how CFCs impact the polar vortex and monsoon circulation. In these regards the manuscript is both novel and interesting. There were, however, a number of things that I was confused about and that need further clarification or description. I also feel there are a number of figures that could benefit from further work. This manuscript is well suited for publication in GDM, is written in a way consistent with the journal style and with further revision I believe it will be suitable for publication in GMD. In this review I have used the notation "P$x$L$y$" and this should be interpreted as page $x$ and line $y$.

**2 Major Comments**

1. The introduction is well motivated in terms of the using idealised models in general (the philosophy of idealised models), however, I think more introductory material is needed for describing the need for adding chemistry into the hierarchy and what these styles of models are used for. For example, it may not be clear to readers if/why chemistry models are needed to investigate the polar vortex or monsoons.

2. I felt the abstract, introduction and conclusions did not sufficiently describe what is currently possible within the ECHAM vs MESSy models. I initially assumed this paper was the first to implement the Held-Suarez test case within ECHAM but realised my mistake on page 10 when the authors described the study of Wan et al 2008. I think what options are (or not) previously available needs to be said much earlier or more clearly. I understand the RELAX submodel is new (ie implementing the parameterisations of newtonian cooling and drag), but were changes to the dynamical core needed or where they already avaailable (if it was available, is it the same/similar as Wan et al 2008?)? I am confused by what is new and what was existing in ECHAM.

**3 Minor Comments**

1. The introduction would benefit from a plane language description of ECHAM vs ECHAM/Messy ( ie what is the standard GCM, atmosphere only or ECM).

2. The manuscript would be easier to read to non-ECHAM specialists if there was a table of acronyms with a short description of each model and where if fits in with the other options.

3. From my perspective, sections 1 through to 4 are describing the implementation and the validation of the code. While in section 5, the model infrastructure is now well justified to use with the chemistry models. I think at the beginning of section 5 this should be more clearly communicated to let the reader that we have reached to point of advertising why a model like this is useful.

Figures often reflect personal styles and different perspectives. I have listed quite a few changes to the figures. These are separated into changes I would like to see made (below) and suggestions which I feel would help (these can actioned at your discretion, see clarifications section).

Requesting the following changes be made to the figures/captions:

1. Fig 1: Are there four options on the y-axis or more? I found it hard to interpret this figure and I am not sure which set-up has which chemistry option. What does '/...' in the 3D dynamical core mean?

2. Consistent colour bars are needed. Fig 2, 3, 10 use a yellow-to-red colour bar to describe $T$, $uv$ and $vT$. Suggest Fig 3 has different colour bar for the fluxes. The blue-to-red colour bar is used for diffs in Fig 3, 7 but for $T$ in Fig 4 and $\Phi$ in Fig 15-16. Suggest diffs for blue-to-red, T use yellow-to-red and another colour option for $\Phi$ etc.

3. Many subplots all have the colour bar repeated. Suggest only having one colour bar or legend per plot.

4. Fig 5 caption should explain $P_{TW}$ and $\gamma$ are from the legend and point to relevant equations.

5. Fig 13: The jet colour map is generally considered bad practice and I suggest a different colour. I found it hard to interpret the zonal mean zonal wind in white and it look me a while to identify what the breaks were a SSW (also is this surface wind or aloft?). On first reading I thought the top panel was u divided by w so the title was confusing for me. Are both u and w essential (could it be described instead as the inverse in general)?. I suggest exploring some other formats for this plot to help draw out the features.

**4 Clarifying Comments**

**4.1 Figures**

Suggestion the following changes be made (optional):

1. Fig 1: The y -axis title 'Chemistry' is floating in a way that it feel out of place (either remove or move). I am not sure what the purpose of the two dotted vertical lines are.

2. Fig 2: The title on the fig is not helpful (suggest removing it).

3. Fig 3: I find the left plots very hard to see. Suggest moving the left panel to a new plot and then keep the flux plots together. Alternatively you could consider only plotting 0-90 in one hemisphere given they are symmetric in this case. What does the 'MA' in the caption (and text) mean? The fonts are too small (also in other plots).

4. Fig 4 Suggest subplot titles are larger and also included in bottom panel.

5. Fig 5: Suggest you use the same seaborn colours as in Fig 8 (matching the values of $\gamma$ in fig 8) – this assumes these are in python though I notice a mix a languages used to generate the plots which is fine.

6. Fig 6 (but in general): might want to consider skipping either red or green in your line plots so everyone can easy see it. I would prefer you use the same colour choices as in Fig 7.

7. Fig 7 (and text): add the equilibrium temp to the legend. What do you mean by 'plane' in titles (and text)?

8. Fig 11-12 bottom panels: suggest using colours not already used in the top panels and they are different so as not to confuse latitude, wind speed and temp differences.

9. Fig 14, if this image is a a pdf/png/jpg or similar, then I would suggest replacing the error bars with filled upper and lower intervals with lower alpha values (ie shading). If your using ps/eps this won't work.

10. Fig 14: the line width is not consistent (thinner is nicer) and I suggest removing titles. The choice of black gives this authority (as is commonly done for obs), was this intentional?

**4.2 Abstract**

1. P1L1: I think it would help to explain why you mean by 'a need emerges'. I know what you mean but it might help to explicitly say it.

2. P1L2: I would suggest a more general description instead of 'process understanding', perhaps ' simulations of the climate'

**4.3 Introduction**

1. P2L8-13: In terms of the Held-Suarez description, a reader could easily get confused about a models dynamical core vs the parameterisation set-up of HS. Suggest rewriting L8-13 to make it clearer that HS was designed as a test for the dynamical core.

2. P3L2: The sentence starting 'The motivation of the MESSy framework was' is an excellent sentence that helped ground me in the context of the configurations. Could I requestion you add this (word for word is fine) to the abstract (something similar is already in the abstract but not as clear) and something just as cleanly described for the EMIL.

3. L9: Could you include what MECCA stands for? Is MECCA the chemistry model of EMAC or it more subtle?

**4.4 Model Description**

1. What surface conditions are you using? Is it generally an aquaplanet with 'water' mountains or does it have a land like surface heat capacity?

2. Equ 2: Worth mentioning the extra term ($\epsilon \sin \phi$) that is not in the original HS formulation is from PK.

3. P6L12: suggest replacing $(40\kappa_a)^{-1}$ with $0.025\kappa_s^{-1}$ and $(4\kappa_s)^{-1}$ with $0.25\kappa_s^{-1}$

4. Equ 8: Why use $\sigma_0$ here and $\sigma_b$ in equ 3?

**4.5 Model test cases, sensitivity simulations and application examples**

1. Fig 4: titles and caption have inconsistent window for the averages, were they 10 or 11 years?

2. P15L1: might help to define UTLS and what it's acronym is.

3. P24L5: a key task of what?

4. P24: I think you should state early in section 5.2 that the monsoon simulations are run with the chemistry scheme. I think you should also say if this is usual, and if not, why is the chemistry scheme is helpful.

5. Sect 5.2: Is there temporal variability in these simulations? If it was said then I missed it.

**4.6 Summary and Outlook**

1. P25L7: Suggest removing 'In the paper presented here,' (it works better without it).

2. L2511: suggest replace 'based on the suggestions by HS94 and PK02' with 'as described in HS94 and PK02'. I would then start a new sentence that described what is new in your implementation.

3. P25L25-27: I found the description of 'climate states' confusing. I also suggest rewriting this sentence (or even multiple sentences) as the grammar has gotten complicated.

4. Please add in the acknowledgements where the SPARC and Era-I data can be downloaded from.

**5 Editorial comments**

There were a number of times where latex has compiled with " instead of " (please review).

**5.1 Title and abstract**

I think the title is too technical. By their very nature GMD papers are technical but I think you could make your title easier to read/understand (and remove some of the acronyms where possible).

1. I found the number of acronyms hard to digest. Given the subject matter, I think ECHAM, MESSY and EMIL are probably fine to use but I would suggest removing RELAX and EMAC as they are not essential. I also think referring to the model as only EMIL or ECHAM/Messy idealied model would help readability. There is a lot of switching between model names which makes it hard to read at times for non-ECHAM experts.

2. I don't think you need to citations in the abstract. I think it is fine to say Held-Suarez model.

**5.2   Introduction**

1. P1L1 'more and more processes and compartments' is awkward, suggest changing to 'increasing complexity' or similar

2. P2L13 (or there abouts) suggest stating that the HS model will be described in detail in section 2.

3. P2L32: The description 'currently underway' reads as though these models are yet to be released. Suggest rewording as both Isca and CESM are broadly used for idealised studies.

4. P3L3: suggest adding '0d' before box model.

5. P3L22-27: I very much liked this paragraph. I would suggest moving it earlier in the introduction. Maybe even as the first paragraph.

6. P3L28: What do you mean by 'consistent'?

7. P3L28: I like the description of model hierarchy of chemistry-dynamical coupling and I suggest you use this more often (esp in abstract).

8. P3L28-P4L2: Suggest moving this paragraph to outlook section.

9. P4L5: replace 'to' with 'two' and suggest separating into two sentences, one that talks about section 3 and one for section 4.

10. P3L6: Suggest joining the two paragraphs that describe what is coming up in the paper.

**5.3   Model description**

1. P5L12: Suggest adding convection to the list of parameterisations.

2. P5L13: suggest you mention Held-Suarez in this sentence.

3. P5L17-20: Suggest you add these as dot points rather than a list in a sentence.

4. P5L25: should cite HS in here.

5. P5L27 (and else ware): suggest replace 'local' with 'environmental'.

6. P5L28: suggest adding this paragraph to the one before and list dot points for the ways of implementing $\kappa$ and $T_{equ}$

7. P6L12: The discussion on $h_{frac}$ should start with a description of $\epsilon$.

8. P6L14: suggest replacing 'sign' with $\pm$

9. P8L15: suggest replacing 'employed' with 'added'.

10. P8L18: suggest replacing 'reads' with 'is given by'

**5.4   Sensitivity simulations**

1. P16: Fig 11-12 are referred to before 9-10. You might want to consider moving figs or mentioning 9-10 earlier.

2. P17L6 and Fig 9: Could you explain why there are multiple Lindgren lines on these plots (which variables are changed)?

**5.5   Supplementary**

The tables would benefit from latex hlines and vlines so they look more like tables. Suggest removing the quotations from all variables. I don't think Fig1-2 are needed but I do not feel strongly about this. Fig 3 is a nice aid to see the call sequence (well done).

---

## Referee Comment (RC3) · Anonymous Referee #3 · 27 Feb 2020

The authors introduce a new idealized and modular modeling setup and demonstrate its use in a couple of ways. I believe the paper would benefit from some restructuring – the paper goes back and forth between model setup issues (choices of values for various parameters) and scientific results which could potentially be a bit confusing to a reader. Perhaps the authors might wish to consider splitting the manuscript in two? Would it be possible to set up a github with a downloadable version of the model? I have doubts about reproducibility which the availability of the model would help to dispel. On the science front, I think the authors are up against some regime issues in dynamical core models, which it would be good to clarify. The original PK02 model shows a very large response to stratospheric perturbations in comparison with observations, and in

the absence of a quantitative theory of how stratospheric perturbations affect the tropo-sphere, responses of the model when planetary scale waves are forced by topography are not necessarily the "correct" response. On readability, the manuscript would bene-fit from some proofreading and fixing of minor typos (in particular, the quotation marks all appear reversed?).

———————————————————

---

## Author Comment (AC2) · 28 Apr 2020

**Reply to Reviewer comments on "Extending the Modular Earth Submodel System (MESSy v2.55) model hierarchy: The ECHAM/MESSy idealized (EMIL) model set-up" by Hella Garny et al.**

We thank all reviewers for their very valuable reviews of our manuscript, which led to a major revision of the paper, as detailed in our response (blue) to the individual reviewer comments (black, italic) below.

In the process of the revision, we discovered an inadvertently introduced modification to the original equilibrium temperature set-up. In the formulation of the "Held-Suarez" tropospheric equilibrium temperature function, used in all the simulations presented in the manuscript, the modified formulation led to a weaker vertical temperature gradient in the tropics. We corrected the formulation to the standard set-up in the model implementation, and repeated a number of simulations. In particular the benchmark simulations presented in Section 3 are now performed with the standard set-up to ensure comparability to earlier studies. To study the sensitivity of the results on the modified versus standard set-up, a number of additional simulations was performed, and the simulations with the differing set-ups are presented in concert in Section 4. Most of the results are qualitatively unchanged with the standard versus modified model set-up. One major difference of the two set-ups is the strength of the response of the tropospheric jet to stratospheric polar vortex changes, and we added a new subsection to discuss this result (see new Section 3.3).

**Edwin Gerber (Referee)**

The authors document a new idealized model configuration within the ECHAM/MESSy modeling framework, and demonstrate how it can be used to investigate open questions in the climate sciences, namely chemistry-transport interactions and the monsoonal circulation. I believe that this work is timely and important, and would be of interest to GMD readers. I therefore recommend publication pending consideration of the comments/suggestions below. As my identify might be obvious given my familiarity with the system, I'm signing this review. Ed Gerber

Thank you, Ed, for your very helpful suggestions to improve our manuscript, that we greatly appreciate.

**General comments**

1) The authors compare the performance of EMIL against a number of benchmark cases that are in the literature. It would be ideal, however, if we could move beyond the "picture norm" for these comparisons – at least in the future. Could you publish the data for these results (or incorporate it within the ECHAM-MESSy distribution), so that in the future, other groups could check their models against yours? The best standard would be to determine whether your integrations are consistent/inconsistent with other benchmark integrations, within the sampling uncertainty. I believe that data can be archived through Zenodo.org, or other structures. You could just provide the zonal mean time mean data needed for the figures. Another option would be to include the key benchmarks as test cases within ECHAM/MESSy, something that could easily be reproduce by another group. Could you provide a citable link to the model and the required parameter scripts? (That is, a frozen version of the model, as was used to produce this paper, ideally with the same run scripts that you used.) I appreciate that the supplement provides all the parameters, but it would still involve a lot of work (and hence many chances to make a mistake) to reproduce this exactly.

Thank you for this suggestion, we agree that it would be very beneficial to be able to do quantitative comparisons of model simulations in the future, rather than the comparison to published Figures, as we have done in our paper. Indeed, having data for comparison available would have likely prevented us from using the inadvertently modified model set-up for the last 3.5 years, as we likely would have discovered the above mentioned differences much quicker.

Thus, we decided to provide the simulation data as freely available data set (via Zenodo), to enable future users to reproduce the analyses presented in the paper. The doi to the data set is inserted in the paper in the "data availability" section.

The ECHAM/MESSy code is available upon registration as MESSy user, and the next MESSy release (v2.55) will include the here described implementation of the idealized set-up, together with a sub-set of the namelists for the simulations presented here. Thus, the user will be able to repeat the simulations. Further, to ensure reproducibility, we include now a table with all simulations and information on the set-up (new Table B1), and further the supplement was improved to make the transition from the parameters in the presented Equations to the namelist parameters easier for future users (see improved Tabels 1-3 in supplement, see also answer to comment 3 below).

2) I appreciate that the authors have striven to find a balance between detailing a new model setup for others to use, and presenting new results. I felt that the test cases that were shown at the end in section 5 were very interesting, but could have been more developed. To provide more space, perhaps the earlier sections could be condensed? (The reader might also be a bit exhausted by the time they reach these really interesting results!) For example, there are a lot of equations and parameters defined in this study, many which are specified in other papers (but also many of which are new). I think some of this detail could best be put in an appendix (e.g., in sections 2.1.1, 2.1.2 and 2.1.3), allowing you to move more quickly to the results.

Thank you for this excellent suggestion, we followed your advice and moved all equations to a new Appendix (Appendix A), and only kept a short description in Section 2, in which we particularly stress which formulations are new.

As we have expanded the results on troposphere-stratosphere coupling, we refrained from adding additional material to Section 5, also because we plan future publications for example on the idealized monsoon circulations.

3) It would help the reader to have a table that defines all the parameters in one place. It would also help you catch any parameters that are multiply defined. One example is k\_max, which appears in the equations (8) and (9) with distinct values. k\_damp is also defined inconsistently between these two equations (though any reasonable reader would understand what is meant). The parameter \delta\phi also appears in multiple equations, e.g., (5) and (16).

Finally, I noticed that \sigma is sometimes used to refer to a vertical coordinate (p/ps), and at other times used width (where I appreciate the motivation is to connect it to the variance of a Gaussian). It might be good to adopt a consistent notation, where \delta is always used for width parameters – but again watching out to make sure all parameters are uniquely defined. (This said, I know that these parameters came from multiple papers in the literature, where the other authors were not consistent with each other!)

Thank you for spotting the inconsistent naming of the parameters, we realized the parameters were also not assigned to the namelist variables in a straightforward way. To address the comment and clarify parameters, we:

- made sure to name parameters consistently and uniquely by following your advice to use sigma for the vertical coordinate and delta for width parameters, and by adding superscripts to variables used in multiple equations (see equations A1-A16).

- added to the Tables in the Supplement (Table 1 to 3) detailed descriptions of the namelist parameters, including default values and the corresponding symbol in the defining equations A1-A16. Thus, the information on the default values is removed from Appendix A, enhancing the readability of this section.

- added a table with all simulation details as Appendix B.

4) The paragraph spanning from page 3 line 28 to page 4 line 2 is very interesting, but seems out of place in the introduction. I would consider pushing this to final section, where you could present it as the next step in your research program.

**Good suggestion, the paragraph is moved to the final paragraph of the paper.**

5) Finally, the topic of regimes comes up quite prominently in section 4. I think this is a very interesting (albeit sometimes frustrating) result that could be mentioned in the abstract and introduction. I think these regimes have simmering in idealized models for sometime: as detailed by Gerber and Polvani (2009), the original PK02 result is so dramatic precisely because of a regime switch between their \gamma 2 and 4 integrations. Chan and Plumb (2009, DOI: 10.1175/2009JAS2937.1) and Wang et al. (2012) discuss this in more detail. The presence of regimes is interesting: if such a thing existed in our atmosphere, we could be in for surprises with global warming (or perhaps when the planet enters an ice age). If it is an artifact of these idealized models, however, it's something that the dynamics community should be wary of. It could lead to unphysical parameter sensitivity or results that are qualitatively disconnected from the real atmosphere, breaking the link we'd hope to establish through model hierarchies.

We agree that the regime behavior in the idealized models is a very interesting result, and we agree that it requires careful evaluation whether those regimes are at all relevant for our real atmosphere. Inspired by your comment, we analyzed the regime behavior in our model simulations more closely, which led to the discovery of the modified implementation in the equilibrium temperature (see above). The modified equilibrium temperature, with lower tropical upper tropospheric temperatures, led to an equatorward shift of the tropospheric jet (in agreement with studies prescribing diabatic heating in the upper tropical troposphere), and interestingly the response of the jet location to stratospheric forcing appears to be strongly damped in this set-up compared to the stndard equilibrium temperature (see new Fig. 6 (left)). We added a new subsection (Section 3.3) to the paper to discuss the stratosphere-troposphere coupling and the regime behavior of the tropospheric jet location. The changed tropospheric state in the simulations with modified set-up appears to inhibit the regime-like behavior of the tropospheric jet, possibly because the jet is located further equatorward in the basic state. Thus, while the general result that the tropospheric response is sensitive to the prescribed tropospheric equilibrium temperatures is in line with the study by Chan and Plumb (2009), the reason for the damped response seems to be a different one (in the simulations by Chan and Plumb (2009), the jet was rather located further poleward in the basic state). However, the dynamical reasons for this behavior remain to be analyzed in detail, which is beyond the scope of the present paper.

Moreover, we want to point out that next to the regimes in the location of the near-surface jet, the regimes we had addressed so far in the paper are regimes in the polar vortex strength. We added more discussion on the polar vortex regimes in Sections 4.1-4.2 (including appended Figures of probability distributions, new Figs. C1-3). More detailed analysis of the polar vortex regimes is part of ongoing work that we plan to publish in a follow-on paper.

Overall, the following changes with respect to discussion of the regimes and of stratospheretroposphere coupling were made:

- added statement to Abstract
- added paragraph to Introduction
- new Section 3.3 and new Fig. 6

- added new Fig. 12, showing relation of tropospheric jet location and stratospheric polar vortex strength, and Figures C1-3 with probability distribution functions of polar vortex strength and free tropospheric and near-surface jet location, and discussion thereof in Sections 4.2 and 4.3 - added paragraph with discussion to Section 6

Specific comments (largely typographical) by page:line number

1:1 Consider "As models of the Earth system grow in complexity, a need emerges to connect them with simplified systems through model hierarchies in order to improve process understanding." Done

1:3 consider cutting "with the aim" Done

1:6 Would you consider ECHAM/MESSy a "model", or rather a "framework" which allows you to build many different models.

Good point, MESSy is definitely a framework, ECHAM/MESSy is one instance of this framework. We decided to change "model" to "framework" here.

1:10 Consider "Test similations with EMIL reproduce benchmarks provided by earlier dry dynamical core studies."

Done (and changed title of Section 3 accordingly to "Model benchmark tests").

1:19 What do you mean by "the ability to simulate dynamical systems"? Dynamical systems in the broadest sense is a whole field in mathematics. Perhaps you mean "the ability to simulate qualitatively realistic dynamical variability of the circulation"

True, this was a misnomer, we rephrased to "circulation systems".

1:22 Consider something like "Earth system models continue to incorporate more processes to enable a more complete simulation of the climate system, and thus produce the best possible climate projections. In practice, this increases the complexity of model codes as new compartments are added to represent new processes." I'm not sure if you need that second sentence; my thought was that the goal is to increase the range of processes that are simulated, and this is effected in practice by adding more compartments, modules, etc..

Done, thanks for the suggestion, we decided to keep the addition on the compartments. *2:9 stray space: "hereafter"*."

Done.

2:13 I think the upper level drag is only in the PK02 set up, and not a part of the original HS94 configuation.

True, removed "and upper level" here.

2:16 consider a paragraph break before "The functions..."

Done.

2:21 "to idealized heating that mimics the thermal response to CO2 increase" I think "climate change" is the response, not the forcing!

True, and changed.

2:26 "motivates one to include"

Changed to "motivates the expansion of ..."

2:29 Jucker and Gerber (2017) were not the first/only one to do this. Consider also referencing: Merlis, T. M., T. Schneider, S. Bordoni, and I. Eisenman, 2013: Hadley circulation response to orbital precession. Part I: Aquaplanets. J. Climate, 26, 740–753, doi:10.1175/ JCLI-D-11-00716.1. Tan, Z., T. A. Shaw, and O. Lachmy, 2019: The sensitivity of the jet stream response to climate change to radiative assumptions, J. Advan. Mod. Earth Sys., 10.1029/2018MS001492. Thanks for pointing those references out, we added them here.

2:35 Here and throughout the text, the quotes seem to be reversed. Perhaps this is set by the journal, but I am used to "hello" as opposed to "hello"

Thanks for spotting this, and corrected.

3:19 consider "allows the creation of model hierarchies"

Done.

3:20 consider "Earth-system model. Any developments..." Done.

5:9 I found "idealzied localize contrained" to be awkward. Consider just "forced by a simple, localized heating that..."

Done.

eqn (1) In HS94 and other papers, it's usually just T\_{eq}

Done, replaced throughout the paper.

5:30 This was a point where I feel you've lost the balance on providing enough technical advice without making the paper too long. Do you need to describe an option that "physicaly of little use" eqns (204) To make the paper more concise, you could refer the reader to HS94. I appreciate that equation (2) is modified by the inclusion of the \epsilon sin(\phi) term; this was documented by equations A3 and A4 in PK02. A happy medium might be to reference past work in the paper, highlighting your modifications, and including equations in an appendix.

As detailed above, the section has been reworked by moving equations and details to the appendix, while only a general description is left in the main part of the paper. Thanks for the suggestions.

5:19 T\_{US} isn't defined in the paper. The reference is: U.S. Standard Atmosphere, U.S. Government Printing Office, Washington, D.C., 1976. (Which I appreciate isn't so easy to find!) Thanks, and added.

eqn (6) Aditi Sheshadri did something like this in her 2015 paper, https://doi.org/10.1175/JAS-D-14-0191.1. There she lowered the start of the vortex to 200 hPa. That said, I appreciate the more thorough investigation of the transition height in this study!

Thanks for pointing this out, we added a sentence in section 4.2.

Figure 3 and surrounding discussion. It is interesting that the jets shift equatorward when you move from the T63L19 to the T42L90 integrations. I suspect the vertical resolution plays a more important role her than you might suspect. This is consistent with the behavior of GFDL's spectral core, where the jets also shift equatorward when the vertical resolution is increased. See Fig. 4 of Gerber et al. (2008), https://doi.org/10.1175/2007MWR2211.1. This doesn't seem to happen in finite difference or finite volume based cores. [This said, I don't mean for you to add another citation; I think you've already been very generous in referencing my past work.]

Interesting that there is a similar behavior in the GFDL model. Newer results of ours also indicate that the vertical resolution might be more important here than we had assumed. We changed the text to "The jets are shifted equatorward in the T42L90MA resolution, and eddy variance is generally reduced. This is likely a combined effect of lower horizontal and higher vertical resolution, in agreement with Wan2008."

11:13 consider a paragraph break after PK02.

We chose to keep the text in one paragraph as still the same Figure is discussed.

11:14 (namely GFDL's spectral dynamica core)

Done.

Figure 5: the caption on this figure could be expanded to help a reader who's skimming the paper, for instance, defining the key parameters  $p_{TW}$  and \gamma that are being used. I'll admit I had to remind myself what  $p_{TW}$  represented.

**Done.**

13:16 Along the lines of my general comment on the "picture norm", it would be ideal to be more precise about what you meant by negligible. I think you mean that it is small relative to uncertainties in the cliamatology with resolution (i.e., T63L19 vs. T42L90), but you could also define it relative to sampling uncertainty (i.e., it would take inordinately long integrations for the difference to be significant above sampling noise.)

True, the statement on the differences due to the different sponge set-up was rather vague so far, and not based on a statistical evaluation. We added a significance test on the differences, to clarify whether the differences are negligible with respect to sampling uncertainty. We based the t-test on slices of 30-day means, assuming that there is no correlation between those 30-day time-slices (given a decorrelation time-scale in those simulations of about 30 days, this should be about right). The addition of the significance test revealed a weak (significant) downward extension of zonal wind differences into the troposphere, that we had not noticed so far. We rewrote the description of the differences to be more precise and quantitative.

Figure 6 and discussion. I appreciated this portion of the paper, but a a quick question: is one month of austral hemisphere gravity wave drag enough to nail down the effective damping rate in models? I don't have a good sense how much this rate varies. I assume this includes both orographic and non-orographic drag? Would the effective rate be much different in the boreal hemisphere during winter? I think it would help to expand the caption, to explain that GWD/u provides an effective damping time scale of the winds when using a full gravity wave drag scheme. True, only one month of data for one hemisphere was a very thin data basis to argue with. The "effective damping time-scales" for both the NH and SH are added to the Figure now, including values from 50 winters in each hemisphere. There is considerable variability between different winters, and the damping is on average stronger in the SH (possibly because of the lower planetary wave activity?). Overall, the chosen damping time-scales in the new sponge layer implementation lie well within the variability. We also added text to the legend to be more clear. 15:4 "cannot"

Done.

Figure 8 and following figures. You could possible color the dashed curves which show the equilibrium profiles, to make the comparison with their respective \gamma's easier. For Figure 8 specifically, please specify the location of this profile. Is it right at the pole?

Done (colored dashed lines for Figures with multiple T\_eq values).

16:7 consider a paragraph break after \gamma.

Obsolete due to re-writing of paragraph (emphasizing on vortex regimes).

17:3 In Wang et al. (2012), I think we had to grapple with this same regime behavior. The model switch abruptly from a state with active stratospheric variability and a strong residual mean circulation (which allows the temperature to deviate substantially from T\_eq) to a state with an very cold, stable vortex near "radiative" equilibrium. In Wang et al., this regime change was associated with a substantial change in the position of the tropospheric jet. Does that happen here?

Yes, and the tropospheric jet shift is discussed in more detail now in Section 4.2 (new Fig. 12 and discussion thereof). Whether the stratospheric polar vortex regime shift and the tropospheric jet location regimes are (necessarily) connected is, to my understanding, a question to be clarified. *17:28 "these two simulations"*

Obsolete, as we removed old Fig. 10 (climatologies), as we felt that they do not add much value, and wanted to compensate for the new additional Figures.

Figure 10 Here you are showing results from integrations which exhibit multiple regimes. Based on past experience (e.g., Wang et al. 2012), regime transitions can introduce very long time scales, as the model switches between states. You can see this of this Figure 5 of your text, which corresponds to pTw=400, gamma 2 integration shown in the right panel (I think.) Therefore, you have to be very careful in establishing convergence. Earlier in the text you suggested that runs were done for 1825 days; it seems that you have longer runs (3000 days are shown in Fig. 5), but I'm not sure that would be sufficient. It would be good to check/comment on the sampling uncertainty in these climatologies.

Agreed, and indeed we find very long time-scales in the simulations with regime transitions (the simulation with gamma = 2 and pTw = 400 presented in old Fig.5 has a decorrelation time scale of ~100 days). We agree that an integration length of 1825 days is thus far too short to establish convergence (indeed, for a simulation with regime transitions, convergence might be never reached). We commented on this issue at the beginning of Section 3, and further in Section 4.2. We have extended the newly simulations (with the standard set-up, see top) to ~10000 days to test for the robustness of the results, but are not able to extend all simulations to this length. However, the new extended simulations do show a generally similar behavior than the old (short) simulations, letting us believe that the sampling uncertainty does not influence our conclusions majorly.

Figure 12 and discussion. I suspect that the strength of the overturning (difference between T and T\_eq) near the model top will be dominated by the drag layer. Hence, it's likely to be determined by \gamma: if you force a stronger vortex, you need a stronger drag. At lower layers, the strength of overturning is dominated by "wave pumping", and so the resolved circulation. I worried about this a lot in preparing my 2012 paper, but convinced myself that in the mid-to-upper stratosphere, the differences in the residual circulation in response to changing gamma were still being dominated by the waves, and so not an artifact of the sponge layer. I'm not exactly sure how far down you need to go to be free of the sponge layer, but perhaps 10 hPa would be a better choice than 1 hPa? This would be supported by Figure 7, where you find that the spong layer has a negligible impact below 10 hPa. I'd also be curious to see if the nonlinearity in the vortex shown in Figure 11, bottom left, shows up in the overturning at 10 hPa in the model with heating. Thanks for pointing this out, and we decided to change the figure to show the 10 hPa results, which also provide more interesting insights – as you suspected, the non-linearity in the vortex is in agreement with the deviation from T eq at 10 hPa.

22:18 consider a paragraph break after "high."

**Done.**

22:19 consider "high latitudes (north of 60N), driven by the strong wave dissipation that effected the SSW; see the red line in the top panel of Fig. 13. This transports ..." Done.

22:22 consider "latidues, evident in Fig. 13 ... 15 ms-." (no paratheses). I'd also consider breaking the paragraph after this sentence.

Done.

22:29 Isolated from what? Consider cutting "in an isolated manner," or to be more specific, e.g., "independent of the annual cycle" or "isolated from all other chemical processes". Moved sentence to the end of the section, and added more specific statement. Figure 13 Consider reworking the caption, as you first refer to the middle panel. It might also be nice to include a second axis on the top panel, or to make "w\*[10^-5 hPa]" in red Reworked figure, and done.

23:1 Consider a paragraph break after "is steep."

We chose to keep the text in one paragraph as still the same Issue is discussed.

23:3 "downwelling is maxized at the"

Done. 23:4 same as above

Done.

24:10 consider a pargarph break after "state."

Done.

24:17 1020 J sounds like a lot, but could you provide some context? Say, what is the effective heating rate per square meter (W/m2), which could be more easily compared to solar or precipitation forcing. With hope this number is in the ball park for what you'd expect from monsoon precipitation.

Thank you for this comment. We added the following sentence to the text: "This heating is of the same order of magnitude as the idealized heat source of \$6\times 10^{19}\$\unit{J} prescribed in Siu and Bowman (2019) to model the North American monsoon anticyclone (see their experiments 5a-5e)." Keeping in mind e.g. that the Asian monsoon anticyclone is clearly more pronounced than the North American monsoon anticyclone, the higher energy input in our study seems reasonable. *24:19 "produced in response to the additional heating"*

Done.

24:23 consider a paragraph break after "respectively.)"

Done.

24:26 Perhaps the anticyclonic centers could be marked/labeled in the figure.

Done.

24:30 You could break the paragraph after "2016).

Done.

24:30 Consider. "An example of eastward eddy shedding was observed during the second period, as displayed on the right of Fig. 16. This phenomenon has been previously investigated..." Done.

25:7 Your summary opens with a hard sentence to parse. Consider from line 8"...model system is documented. The set-up, denoted EMIL (explain the acronym), is shown to perform consistently with established dry dynamical core benchmarks, both earlier configurations of the ECHAM core, and those developed by other modeling centers."

Thanks, and Done.

25:26 "used setups. The polar"

Done.

26:1 This is an interesting result, as we see this coupling in observations (i.e., with the ozone hole, or following an SSW). It is my understanding that the tropospheric state of the Lingren et al. (2018) model is substantially different, and might explain why does not couple to the stratosphere. As you have shown in Figure 10 (right panel), for instance, easterlies are generated in the UTLS region of the winter hemisphere.

We added substantial discussion of the tropospheric jet response in the different set-ups of the model, mentioning also that from observational evidence we expect a vertically coherent response of the tropospheric jet (which is not seen in the "Lindgren" set-up).

26:3 consider "we present, as a proof-of-concept, a" Done.

**Reviewer Name: Penelope Maher**

**Summary of the Review**

This manuscript describes the implementation of the Held–Suarez configuration, with the Polvani– Kuchner amendment for the stratosphere, within the ECHAM/MESSy modelling framework. From the model description, it seems the model has been implemented in a modular nature which is a credit to the modelling effort (this can be a development nightmare otherwise). The manuscript has a well described parameterisation equation set, and has tested the relevant parameter spaces for the tunable variables and compares their results with the literature. The new model set-up is then used, as a proof of concept, for looking at how CFCs impact the polar vortex and monsoon circulation.

Thank you for the positive description of our modeling efforts.

Unfortunately, we realize that a general misunderstanding arose: in the model version we describe in this paper, it is possible to analyze the impact of dynamical variability and forced changes ON tracer distributions, including diagnostic chemical tracers as shown for the CFC example (Section 5.1), but NOT the impact of e.g. CFCs on dynamics. No feedback of the chemical tracers on dynamics exists currently in the model.

Thanks to your following comments and to avoid this misunderstanding, we reworked the model description and motivation (in Abstract, Introduction and Summary), as detailed below.

In these regards the manuscript is both novel and interesting. There were, however, a number of things that I was confused about and that need further clarification or description. I also feel there are a number of figures that could benefit from further work. This manuscript is well suited for publication in GDM, is written in a way consistent with the journal style and with further revision I believe it will be suitable for publication in GMD. In this review I have used the notation "PxLy" and this should be interpreted as page x and line y.

**2 Major Comments**

1. The introduction is well motivated in terms of the using idealised models in general (the philosophy of idealised models), however, I think more introductory material is needed for describing the need for adding chemistry into the hierarchy and what these styles of models are used for. For example, it may not be clear to readers if/why chemistry models are needed to investigate the polar vortex or monsoons.

The motivation to implement the idealized model set-up in the framework of a chemistry-climate model system is, for the current set-up, the ability to study the impact of idealized dynamical variability and forced changes on to the distribution of (chemically active) tracers. This model set-up is motivated by a large number of research questions on the distribution of chemical substances in the atmosphere, e.g. the question how changes in the circulation in a changing climate will affect stratospheric ozone, and how important the monsoon systems are in transporting tracers from the troposphere to the stratosphere. As detailed in the paper, it will be the next step to couple the chemistry and dynamics, a task that will be possible to perform within the chemistry-climate model system framework we use. This second step is motivated by the research question on how circulation-induced anomalies in radiative trace gases (e.g. ozone and water vapor) feed back on the dynamics, a question that is relevant both on climate time-scales as well as in intra-seasonal timescales (e.g. during sudden warmings).

We added the motivation for the inclusion of diagnostic (chemical) tracers in the Introduction (see p3, line 16ff), and motivation for the next step in the hierarchy (including the coupling), that we moved to last paragraph of the paper (see also Ed Gerbers major comment 4).

2. I felt the abstract, introduction and conclusions did not sufficiently describe what is currently possible within the ECHAM vs MESSy models. I initially assumed this paper was the first to implement the Held-Suarez test case within ECHAM but realised my mistake on page 10 when the authors described the study of Wan et al 2008. I think what options are (or not) previously

available needs to be said much earlier or more clearly. I understand the RELAX submodel is new (ie implementing the parameterisations of newtonian cooling and drag), but were changes to the dynamical core needed or where they already available (if it was available, is it the same/similar as Wan et al 2008?)? I am confused by what is new and what was existing in ECHAM.

Thank you for pointing out the fact that we need to state much clearer the difference of our model set-up, i.e. using ECHAM as base model within the MESSy framework, to the original ECHAM model. MESSy is a framework that allows to link a base model (i.e. a dynamical core, here ECHAM) to submodels (e.g., physical parameterizations, diagnostics, and among others the chemistry scheme). While Wan et al. (2008) used a dynamical core version of ECHAM in their paper, this version was to our knowledge only used for testing purposes of the model core and is not part of the general model distribution of ECHAM. The implementation we performed here is new in that it was developed within the MESSy framework. One advantage of MESSy is that the implementation of the model was possible simply through the implementation of a new submodel for the relaxation, and the other physical parameterizations could be simply switched off. Thus, no changes to the dynamical core had to be made. Within MESSy, it is now possible to run the dynamical core model with the same executable as more complex versions of the model, and the idealized model set-up is available for all model users. Moreover, the full infrastructure on tracer set-up, transport and chemical reactions, available in the MESSy framework, can be exploited also with the dynamical core model.

We added text on the distinction and motivation of our implementation of the dry dynamical core model within MESSy to the Introduction (p4, I20 ff) and Abstract (p1, line 6-7).

**3 Minor Comments**

1. The introduction would benefit from a plane language description of ECHAM vs ECHAM/Messy (ie what is the standard GCM, atmosphere only or ECM).

We expanded the description of the MESSy framework in the Introduction, to state that ECHAM is (one possible) dynamical core used within the MESSy framework:

"The MESSy framework couples a base model (dynamical core) to submodels, that contain the physical parametrizations as well as diagnostics. Among other base models, the ECHAM dynamical core is available in MESSy." (p3, line 33)

2. The manuscript would be easier to read to non-ECHAM specialists if there was a table of acronyms with a short description of each model and where if fits in with the other options. While we do see that the number of Acronyms are confusing (in particular for non-MESSy users), we do not feel like we can omit any of them, as the paper also serves as documentation. However, we did try to reduce the usage of the acronyms in the text as much as possible to increase readability, and explain the model framework in more detail (which we hope serves the purpose more than adding a table).

3. From my perspective, sections 1 through to 4 are describing the implementation and the validation of the code. While in section 5, the model infrastructure is now well justified to use with the chemistry models. I think at the beginning of section 5 this should be more clearly communicated to let the reader that we have reached to point of advertising why a model like this is useful.

Thanks for the suggestion, and we added a sentence at beginning of section 5.

Figures often reflect personal styles and different perspectives. I have listed quite a few changes to the figures. These are separated into changes I would like to see made (below) and suggestions which I feel would help (these can actioned at your discretion, see clarifications section). Requesting the following changes be made to the figures/captions:

1. Fig 1: Are there four options on the y-axis or more? I found it hard to interpret this figure and I am not sure which set-up has which chemistry option. What does '/...' in the 3D dynamical core mean?

The figure caption was expanded to clarify the meaning of the axis, and the Figure was slightly reworked to emphasize on the model set-ups with /without coupled chemistry.

2. Consistent colour bars are needed. Fig 2, 3, 10 use a yellow-to-red colour bar to describe T, uv and vT. Suggest Fig 3 has different colour bar for the fluxes. The blue-to-red colour bar is used for diffs in Fig7 but for T in Fig 4 and  $\Phi$  in Fig 15-16. Suggest diffs for blue-to-red, T use yellow-to-red

**and another colour option for $\Phi$ etc.**

We reworked these Figures according to your suggestion to use consistent colour bars (Figs. 3, 4, 15, 16 (now 17, 18))

3. Many subplots all have the colour bar repeated. Suggest only having one colour bar or legend per plot.

Done.

4. Fig 5 caption should explain PT W and y are from the legend and point to relevant equations. We added the description of p\_Tw and gamma in the caption.

5. Fig 13: The jet colour map is generally considered bad practice and I suggest a different colour. I found it hard to interpret the zonal mean zonal wind in white and it look me a while to identify what the breaks were a SSW (also is this surface wind or aloft?). On first reading I thought the top panel was divided by w so the title was confusing for me. Are both u and w essential (could it be described instead as the inverse in general)?. I suggest exploring some other formats for this plot to help draw out the features.

The plot was reworked by changing the color map, and adding an additional y-axis in the upper panel to avoid confusion on the time-series of u and w\*. The caption now clarifies that the wind contours in the middle panel are displaying winds at 50 hPa.

**4 Clarifying Comments**

**4.1 Figures**

Suggestion the following changes be made (optional):

1. Fig 1: The y -axis title 'Chemistry' is floating in a way that it feel out of place (either remove or move). I am not sure what the purpose of the two dotted vertical lines are.

The label "Chemistry" was moved, and the caption expanded to make the purpose of the vertical lines more obvious.

2. Fig 2: The title on the fig is not helpful (suggest removing it).

3. Fig 3: I find the left plots very hard to see. Suggest moving the left panel to a new plot and then keep the flux plots together. Alternatively you could consider only plotting 0-90 in one hemisphere given they are symmetric in this case. What does the 'MA' in the caption (and text) mean? The fonts are too small (also in other plots).

The plot is redone and resized to make it more visible, The "MA" is omitted from the title to avoid confusion, and explained in the text (MA="Middle Atmosphere", i.e. high-top version of model levels).

4. Fig 4 Suggest subplot titles are larger and also included in bottom panel.

Added titles to subplots, and omitted upper panels to keep balance of number of figures (two new additions).

5. Fig 5: Suggest you use the same seaborn colours as in Fig 8 (matching the values of  $\gamma$  in fig 8) – this assumes these are in python though I notice a mix a languages used to generate the plots which is fine.

Well spotted, there is indeed a mixture of languages due to the different habits of the authors. Most figures are reworked now (in python), as is Fig, 5. Colors of Fig. 8 (new Fig. 9) and the time-series in Fig. 5 are consistent now.

6. Fig 6 (but in general): might want to consider skipping either red or green in your line plots so everyone can easy see it. I would prefer you use the same colour choices as in Fig 7. Usage of Red and Green lines in Fig. 6 (new 7) is omitted now.

7. Fig 7 (and text): add the equilibrium temp to the legend. What do you mean by 'plane' in titles (and text)?

The title "plane surface" is omitted to avoid confusion (without topography was meant) and the equ. temp. line is explained in the caption.

8. Fig 11-12 bottom panels: suggest using colours not already used in the top panels and they are different so as not to confuse latitude, wind speed and temp differences.

Due to the addition of the new experiments, the coloring changed and is unique in each set of panels (top versus bottom).

9. Fig 14, if this image is a a pdf/png/jpg or similar, then I would suggest replacing the error bars with filled upper and lower intervals with lower alpha values (ie shading). If your using ps/eps this won't work.

Decided to leave figure as is (it is an eps file).

10. Fig 14: the line width is not consistent (thinner is nicer) and I suggest removing titles. The choice of black gives this authority (as is commonly done for obs), was this intentional? Fixed (thinner lines in all panels, title removed). Black was intentional as this is the base case presented in Fig. 13.

**4.2 Abstract**

1. P1L1: I think it would help to explain why you mean by 'a need emerges'. I know what you mean but it might help to explicitly say it.

Thank you for the suggestion. As the reason for the "emerged need" is detailed in the first sentences in the introduction, we felt that adding it to the Abstract as well would blow up the Abstract too much. Also, we feel that the addition "to improve process understanding" does explain already why we need the model hierarchies.

2. P1L2: I would suggest a more general description instead of 'process understanding', perhaps ' simulations of the climate'

As detailed above, we think that the reason for the simplified models is exactly the seek for process understanding, while the complex models are the ones that are used for the best possible "simulations of the climate". Therefore, we would rather keep the formulation as is. See first paragraph of introduction.

**4.3 Introduction**

1. P2L8-13: In terms of the Held-Suarez description, a reader could easily get confused about a models dynamical core vs the parameterisation set-up of HS. Suggest rewriting L8-13 to make it clearer that HS was designed as a test for the dynamical core.

True, and reformulated to make it more clear that we mean a "Held-Suarez" type model here, not the particular functions for the equilibrium temperature they propose.

2. P3L2: The sentence starting 'The motivation of the MESSy framework was' is an excellent sentence that helped ground me in the context of the configurations. Could I requestion you add this (word for word is fine) to the abstract (something similar is already in the abstract but not as clear) and something just as cleanly described for the EMIL.

Thanks for the suggestion, we revised the paragraph in the Abstract to describe the MESSy hierarchy and the motivation for the implementation of the dry dynamical core model within this framework more clearly:

"The Modular Earth Submodel System (MESSy) was developed to incorporate chemical processes into an Earth System model. It provides an environment to allow for model configurations and setups of varying complexity, and as of now the hierarchy reaches from a chemical box model to a fully coupled Chemistry-Climate model. Here, we present a newly implemented dry dynamical core model set-up within the MESSy framework, denoted as ECHAM/MESSy IdeaLized (EMIL) model set-up. EMIL is developed with the aim to provide an easily accessible idealized model set-up that is consistently integrated in the MESSy model hierarchy. The implementation in MESSy further enables the utilization of diagnostic chemical tracers."

3. L9: Could you include what MECCA stands for? Is MECCA the chemistry model of EMAC or it more subtle?

Yes, MECCA is the chemistry module used in EMAC. The full name is added to the text. *4.4 Model Description*

1. What surface conditions are you using? Is it generally an aquaplanet with 'water' mountains or does have a land like surface heat capacity?

As in the dry dynamical core model, the only interaction with the ground is via the prescribed friction, the ground does not actually have any heat capacity (it is implicitly included in the prescribed equilibrium temperature).

2. Equ 2: Worth mentioning the extra term ( $\phi \sin \phi$ ) that is not in the original HS formulation is from *PK*.

We added the remark that the asymmetry term was added is from the PK study (see equ. A1, moved to appendix in response to Reviewer Ed Gerber).

**3. P6L12: suggest replacing (40ка)-1 with 0.025к-1 s and (4кs) -1 with 0.25к-1 s**

The default values have been omitted from the text, and moved to the new Table 1 in the supplement.

4. Equ 8: Why use  $\sigma$ 0 here and  $\sigma$ b in equ 3?

As those are two distinct parameters, we chose to keep the distinct labeling to be able to keep them apart.

4.5 Model test cases, sensitivity simulations and application examples

1. Fig 4: titles and caption have inconsistent window for the averages, were they 10 or 11 years? True, indeed 11 years as on the Figure titles were used. However, as we have chosen to remove the panels showing ERA-Interim climatologies to compensate for the addition of new Figures, this is obsolete.

2. P15L1: might help to define UTLS and what it's acronym is

Thanks and done.

**3. P24L5: a key task of what?**

Rephrased.

4. P24: I think you should state early in section 5.2 that the monsoon simulations are run with the chemistry scheme. I think you should also say if this is usual, and if not, why is the chemistry scheme is helpful.

In the monsoon simulations presented in the paper, no chemical tracers were included (see clarifications at top of review). However, for future applications, we plan to use diagnostic tracers to study tracer transport in the idealized model framework, one of the major research questions about monsoon circulation systems (see addition to the Introduction, new p3, line 20 ff).

5. Sect 5.2: Is there temporal variability in these simulations? If it was said then I missed it. The forcing term is set constant, so there is no variability in the forcing. This is described in the second paragraph of Section 5.2.

**4.6 Summary and Outlook**

1. P25L7: Suggest removing 'In the paper presented here,' (it works better without it). Done.

2. L2511: suggest replace 'based on the suggestions by HS94 and PK02' with 'as described in HS94 and PK02'. I would then start a new sentence that described what is new in your implementation.

Done.

3. P25L25-27: I found the description of 'climate states' confusing. I also suggest rewriting this sentence (or even multiple sentences) as the grammar has gotten complicated. Done.

4. Please add in the acknowledgements where the SPARC and Era-I data can be downloaded from.

Done (for SPARC, ERA-I is not used directly anymore).

**5 Editorial comments**

There were a number of times where latex has compiled with " instead of " (please review). Done.

**5.1 Title and abstract**

I think the title is too technical. By their very nature GMD papers are technical but I think you could make your title easier to read/understand (and remove some of the acronyms where possible). We tried to reduce the usage of acronyms in the text of the paper, including the Abstract, which hopefully is now much more readable also to non-MESSy -users. However, we think the title needs to include the model system/ set-up names (indeed, it is requited by GMD to include version numbers), so we decided to keep it as is.

1. I found the number of acronyms hard to digest. Given the subject matter, I think ECHAM, MESSY and EMIL are probably fine to use but I would suggest removing RELAX and EMAC as they are not essential. I also think referring to the model as only EMIL or ECHAM/Messy idealied model would help readability. There is a lot of switching between model names which makes it hard to read at times for non-ECHAM experts.

We agree, and as said above, we tried to reduce the usage of the acronyms in the text, and hopefully explain the nature of the model framework more clearly now. However, we don't think we can remove any of the acronyms completely, as they are important informations to the model

users.

2. I don't think you need to citations in the abstract. I think it is fine to say Held-Suarez model. References are omitted.

5.2 Introduction

1. P1L1 'more and more processes and compartments' is awkward, suggest changing to 'increasing complexity' or similar

Rephrased (see Ed Gerbers review).

2. P2L13 (or there abouts) suggest stating that the HS model will be described in detail in section 2.

Done.

3. P2L32: The description 'currently underway' reads as though these models are yet to be released. Suggest rewording as both Isca and CESM are broadly used for idealised studies. Rephrased to "are aiming to…".

4. P3L3: suggest adding '0d' before box model.

Done.

5. P3L22-27: I very much liked this paragraph. I would suggest moving it earlier in the introduction. Maybe even as the first paragraph.

Thanks, and we moved to paragraph to an earlier place (before the whole MESSY framework is explained, to make it clear why we use it).

6. P3L28: What do you mean by 'consistent' ?

With "consistent", we here refer to a model hierarchy which adds processes in a logical order to study chemistry-dynamics interactions. The paragraph has been moved to the end of the paper, where it is put more in context.

7. P3L28: I like the description of model hierarchy of chemistry-dynamical coupling and I suggest you use this more often (esp in abstract).

See above, the paragraph has been moved to the very end of the paper, to make it more clear that this is an outlook.

8. P3L28-P4L2: Suggest moving this paragraph to outlook section.

Done.

9. P4L5: replace 'to' with 'two' and suggest separating into two sentences, one that talks about section 3 and one for section 4.

Done.

10. P3L6: Suggest joining the two paragraphs that describe what is coming up in the paper. Done.

5.3 Model description

1. P5L12: Suggest adding convection to the list of parameterisations.

Done.

2. P5L13: suggest you mention Held-Suarez in this sentence.

Done.

3. P5L17-20: Suggest you add these as dot points rather than a list in a sentence.

Done.

*4. P5L25:* should cite HS in here. As this is the generic equation for temperature relaxation, we don't think the reference in necessary here – it is cited again two sentences later.

5. P5L27 (and else ware): suggest replace 'local' with 'environmental'.

Changed to "actual temperature"

6. P5L28: suggest adding this paragraph to the one before and list dot points for the ways of implementing  $\kappa$  and Tequ

Obsolete due to restructuring of section.

7. P6L12: The discussion on hf rac should start with a description of *q*.

Done.

8. P6L14: suggest replacing 'sign' with ±

We keep the "sign" function here, as this function is returning the sign of the given parameter. 9. P8L15: suggest replacing 'employed' with 'added'. Done. 10. P8L18: suggest replacing 'reads' with 'is given by' Kept as is to avoid duplication of "given by"

5.4 Sensitivity simulations

1. P16: Fig 11-12 are referred to before 9-10. You might want to consider moving figs or mentioning 9-10 earlier.

Consistent order of Figures is ensured now.

2. P17L6 and Fig 9: Could you explain why there are multiple Lindgren lines on these plots (which variables are changed)?

The differences in the set-up (i.e. in the Equilibrium temperature) are explained in the second paragraph of the section, and in the figure legends.

5.5 Supplementary

The tables would benefit from latex hlines and vlines so they look more like tables. Suggest removing the quotations from all variables. I don't think Fig1-2 are needed but I do not feel strongly about this. Fig 3 is a nice aid to see the call sequence (well done). The supplement, including tables, has been revised.

**Anonymous Referee #3**

The authors introduce a new idealized and modular modeling setup and demonstrate its use in a couple of ways. I believe the paper would benefit from some restructuring – the paper goes back and forth between model setup issues (choices of values for various parameters) and scientific results which could potentially be a bit confusing to a reader. Perhaps the authors might wish to consider splitting the manuscript in two?

We thank the reviewer for the suggestion. Following Ed Gerber's review, we restructured the paper to enhance readability (moved considerable part of the technical descriptions to Appendix), and added more scientific discussion on the results (namely, stratosphere-troposphere coupling and dynamical regimes). We agree with the reviewer in that the scientific results might be expandable, and a second paper on the dynamical regimes of the polar vortex, that we find in the parameter sensitivity experiments, is in preparation.

Would it be possible to set up a github with a downloadable version of the model? I have doubts about reproducibility which the availability of the model would help to dispel.

The MESSy model is available upon obtaining a licence (see code availability section), so a freely downloadable version at github is not possible. As detailed in response to Ed Gerber's review, the next model release will contain sample namelists for the experiments conducted in the current study, so that the reproducibility will be ensured. Further, we added a new table with the specifications of the simulations, and the supplement contains now more detailed instructions for setting the parameters. Also, we decided to provide the data of the presented experiments via zenodo to enable future comparisons to other models.

On the science front, I think the authors are up against some regime issues in dynamical core models, which it would be good to clarify. The original PK02 model shows a very large response to stratospheric perturbations in comparison with observations, and in the absence of a quantitative theory of how stratospheric perturbations affect the troposphere, responses of the model when planetary scale waves are forced by topography are not necessarily the "correct" response. We agree, and added discussion on regimes and stratosphere-troposphere coupling in our simulations (for details, see response to Ed Gerbers general comment 5).

On readability, the manuscript would benefit from some proofreading and fixing of minor typos (in particular, the quotation marks all appear reversed?).

We implemented the suggested changes by the other reviewers, and revised the manuscript with fixing all typos etc. we were able to identify.

---

## Editor Decision (ED1)

I appreciate the authors' efforts to respond to my comments, and those of the other reviewers, and feel that the manuscript has improved considerably. I'm also very glad that they were able to catch the coding bug. (If it is any consolation, on my very first paper I was double checking a calculation during the revision phase when I horrifyingly discovered that the radius of the Earth in my code was off by a factor of 10! Fortunately I was able to redo all the calculations and the key results were unaffected.)

My chief concern with the manuscript is that it is still very long, such that some of the very interesting results (particularly the new applications at the end) might get lost. I've tried to offer suggestions on how it could be shortened, and other typographical suggestions, but leave the final decisions to the authors.

Ed Gerber

Minor comments and suggestions

1) I think it would be easier to describe the log10 coding error in terms of the parameter \delta_z in equation A1. In coding the natural log as a base 10 log, the authors' effectively reduced the amplitude of \delta_z by a factor of ln(10)=2.3, so that instead of the default value of 10 K, it was just 4.3 K. I would still explain why this happened (i.e., that it was an error in the representation of the logarithm), but you could refer to the alternative integrations as \delta_z=4.3 and fit them easily in the Held-Suarez framework. To me, this gives me a more physical sense of error: it was a reduction of the tropical stratification by about 6 K, from 10 to 4 K

I think that it would be better to introduce this alternative configuration earlier in the manuscript. An appropriate location would be when discussing the Newtonian cooling in Section 2.1, at the bottom of section 6. Here you explain the HS94 and PK02 profiles, and could mention that (inadvertently) the \delta_z parameter was varied in some of the integrations, revealing interesting behavior that will be discussed later in the manuscript.

2) This comment is about how the change in delta_z parameter may impacts the circulation. The authors indicated that it increases the equator-to-pole temperature gradient, but I think it's primary affect might be through the stratification of the tropics and subtropics. It is true that modifying this parameter does increase the equator-to-pole temperature gradient in the upper troposphere, but the effect is on the order of a few degrees, compared to parameter \delta_y which is 60K. As I understand, this parameter was designed to increase the stratification in the tropics and subtropics, mimicking the impact of moisture. (Moist convection drives the tropics to a state of constant moist potential temperature, which imparts a dry stratification which increases with moisture content, which follows the surface temperature).

The stratification in the tropics is important in allowing the Hadley Cell to transport energy (it increase the gross stability, and hence reduces the need to transport mass to transport a given amount of energy). It may also affect midlatitude eddies, as baroclinic instability is sensitive to the stratification, particularly in the subtropics. Reducing the stratification in the subtropics would favor baroclinic instability further equatorward, and so could help explain the shift in the

jet.

3) I think the paper reads better with the equations moved to the supplement but as I mentioned above, it is still very long. A number of early figures are needed just to compare against earlier results (Figs 3, 4, 6a). I do see that these are essential, but unfortunately they not very exciting, compared to the results at the end. My concern is that by the time the reader gets to the interesting new case studies in section 5, they might be rather exhausted.

There's also a great deal of information on the parameter sweep experiments in Figs 5, 9-13, and C1-C3 (which are substantially referred to in the text). It's hard for me to pinpoint what might be less important, but I would urge the authors to consider trimming or consolidating plots as much as possible, as to highlight the results you want people to remember.

To help summarize/focus the paper, would you agree that these are the key variations of the model that you consider:
i) \delta_z (stratification)
ii) stationary waves (topography vs. thermal)
iii) sponge layer

Of these, the sponge layer had the least impact on the dynamics of the stratosphere and troposphere. Perhaps this could be put in an Appendix, along with Figs 7 and 8), if you wanted to move the text faster. (This is not to say, however, that the exponential profile isn't an improvement.)

The two key control parameters that you vary are:
a) p_Tw (depth of the vortex)
b) \gamma (strength of the vortex)
These two knobs are not independent, although they do have different impacts on the UTLS region.

In terms of the results, it seems that the model's tropospheric jet stream tends to fall into one of two states: a fairly stable jet located around 30 degrees or fairly stable jet located around 40 degrees. Increasing gamma or increasing p_Tw tends to pull the jet further poleward, towards the high latitude state. When a control parameter tends to shift the jet between these states, however, you can observe regime behavior with a more variable distribution. But if the jet start too stably in the 30 degree state (i.e., with reduced \delta_z) or too stably in the 40 degree state (i.e., with thermal topography), there is no transition and the control parameters have a limited impact on the jet.

In the stratosphere it seems that three states are possible. When the vortex is very weak, there is limited variability of the vortex. When it is increased to moderate levels, the vortex is highly variable (at least ina relative sense) and the temperature is far from the equilibrium state due to a strong overturning circulation. And finally, when the vortex becomes sufficiently strong, it tends to hover near the equilibrium state, with reduced variability.

These stratospheric transitions are not necessarily tied to the tropospheric jet stream transition (as was the case in Wang et al. 2012). This does make it trickier to explain all the behavior. But if the figures could be organized to highlight these results, you might find it easier to focus the paper and trim.

4) I feel the case study in section 5.1 is ideal for highlighting why this new EMIL model will be so useful. To my knowledge, this examination of the impact of dynamics on photolysis would simply not be possible in other models. This was a very interesting section.

I felt the monsoon case in section 5.2, however, to be less developed. For example, I found myself wondering about the statistical robustness of the eddy shedding and splitting in Figure 17. More importantly, it seems that this sort of work could be done with any dry dynamical core with HS94 style forcing. I would suggest to consider saving this monsoon case to another paper, when you can highlight the troposphere-stratosphere tracer transport, a topic that would be much harder to explore with a traditional dry dynamical core model. Saving this example for a later paper would also permit you to reduce the appendix, cutting equations A12-16, which include description of the temporal parameters that are not used in this manuscript.

Very minor comments and typographical suggestions by page:line number

2:7 Maher et al. 2019 is now published.

2:13 ...for testing the dynamical cores of atmospheric models, the...

2:16 consider "all thermodynamical processes, e.g., radiation and convection, by a relaxation toward a prescribed equilibrium temperature profile, and the surface boundary layer by Rayleigh friction ...

3:18 changes on the transport of

3:20 this question has received a lot of

6:2 I believe i.e. should generally be followed by a comma, here and other places in the manscript

6:5 consider change the last sentences to "We describe the submodel in the next subsection, and provide technical details of the model setup (namelist choices, etc.) and implementation in the supplement.

6:14 consider removing the parentheses here.

6:16 you could cut this last sentence

6:25 Consider moving the sentence starting with "In section 4.2," up one sentence, to follow "winter hemisphere."

Figure 2 -- as I noted in the major comments, figure 6b might fit here, as a second panel. I appreciate that you would want to use a linear pressure, but otherwise it would put all the information on T_eq in one place.

7:4 There are a lot of parentheses in this sentence, which with the numbers 1) 2) and 3) were hard to parse.

7:11 The function names imply the intent of these diabatic heating/cooling options (i.e., to simulate climate change, stationary waves, and a monsoonal circulation), but these goals aren't fully explained in the paragraph. It might be good to explain both what each option allows and how it has been used in the past. For example:

The tteh_cc_tropics option allows the user to apply a zonal mean heating tendency with a Gaussian shape in latitude and pressure, as detailed in equation A10. The default values are taken from Butler et al. (2010) to approximate the impact of global warming on the atmospheric circulation.

Note that I would not call this exponential decay, which I would associated with exp(-x), not exp(-x^2).

7:12 (diabatic heating and cooling)

8:13 ... specified for each simulation in Table B1 and figure captions.

8:13 Consider rephrasing the sentence about it being favorable to convergence of the climatology is reached. I appreciate that the authors had to optimize their use of computational resources to obtain reasonable confidence about their results. This is particularly challenging for integrations with regime behavior, and is reflected in large uncertainties in some of the integrations.

8:20-25 You could also compare your results to the plots in the original HS94 paper, or at least refer the reader to the paper, which show some of these statistics as well.

8:21 It might be good to define the T63L19 notation here; I think it's the first time we've seen it. [You do define in the next paragraph.]

8:26 with a higher top

8:29 equatorward with T42L90MA resolution, and the eddy variance is generally reduced.

Figure 3: Consider using red-blue color scale for panels b,c. If panels e and f had the same color scale, it would give a better impression of the relative error.

9:1 Could you briefly explain what you mean by "resolution of convergence"? In Gerber et al. 2008 (which does not need to be cited) we found that the aspect ratio of resolution appears to matter, so that some of these effects (e.g., the shift in the jet stream) still appear at high horizontal resolution if the vertical resolution is sufficiently high.

10:13 It might be good to direct the reader to the relevant details on the mountain, e.g., equation (1) of Gerber and Polvani 2009.

8:30 What do you mean by "in a agreement with Wan et al. (2008)" Did they also show these effects of horizontal and/or vertical resolution?

Figure 5 caption and legend. I was initially confused about the meaning of \mu here, thinking that it was some parameter. Consider defining it here and the first time it is used in the text. For the caption, you could say "the average windspeed at this location, denoted \mu, is given in the legend".

12:12 Consider a paragraph break after (Butler et al. 2010).

13:3 free tropospheric jet (see Fig. C1).

13:9 (2009), however, the jet was shifted to higher latitudes in the simulation with a weak response. Thus, ....

13:21 What do you mean by non-physical? Gravity waves due end up providing a sink of momentum in the stratosphere, so there is some physical basis for this. But they do not behave like a Rayleigh friction, particularly when you start modifying the climate with parameter gamma, etc.. Shepherd and Shaw (2005) show why it's important that gravity wave parameterizations conserve momentum, but I think this is a finer point than you intended here.

13:27 but generally increases exponentially with decreasing presssure, not quadratically.

Could Figure 6b be incorporated into Figure 2?

14:1 set-ups naturally differ within

14:2 temperature also extend

14:3 There are also a lot of differences in the tropics, where you get jets. Are these well converged? Consider adding ... in particular the high latitudes and the tropics, where alternating jets form, reminiscent of Quasi-Biennial Oscillation winds albeit fixed in time.

14:11 since for both a flat surface and

Figure 8. The stippling dots were rather large here! (I don't mean too be picky, but you could describe these as polka dots!)

15:5 This is the first time you introduce the SPARC climatology, but it was used in Figure 8. You could just move this up to the preceding paragraph.

5:14 I'm not sure what is meant by unrealistic "step". I think the main problem is that the PK02 damped the winds to the climatology, but as the authors describe in other places in the manuscript, the climatology is warmer than radiative equilibrium in the high latitudes due to the overturning circulation (in the TEM framework, or equivalently due to meridional heat fluxes in the traditional, Eulerian framework).

17:17 I had trouble parsing this paragraph. I've made some suggestions below, but consider reworking it, as it is highlighting an interesting result.

17:19 polar vortex increases nonlinearly with increasing \gamma. In line ... at 10 hPa a the polar vortex accelerates more strongly with when \gamma exceeds a critical value (...

17:22 the polar stratosphere, i.e., change in parameter \gamma, the polar vortex...

17:28 This is an interesting paragraph, but I think you might be able to explain the regime transition more clearly in the context of the literature. There are two regimes here: for a moderate vortex, the winds are westerly, allowing Rossby wave propagation (Charney and Drazin 1961). This is allows waves to propagate up, breaking up the vortex and raising the temperature considerably above radiative equilibrium. When the vortex becomes two strong, however, stationary waves (or quasi-stationary waves) can no longer propagate fast enough to keep up with the flow, preventing waves from enter the vortex. Hence there's no dynamic tendency to offset the radiative forcing (at least until you get to the gravity wave drag layer much higher).

In my major comment above, I thought there might also be a third regime, when there is no vortex and easterlies prevent any wave propagation; this would a summer vortex situation, but again, you end up pretty close to equilibrium for the lack of wave driving circulation.

Figs 10, 11, 12. It would aid the reader to adopt a natural progression in the color scale, say a rainbow from red to purple as you transition along a parameter. That is, ptw=100 could be red, ptw=250, ptw=450 would be purple. When you have multiple configurations (ln vs. ln10) would be to use dashes for one and solid lines for the other.

21:3 In general I find that footnotes overemphasize the point, as the reader stops and moves down to figure it out. I have actually worried about this issue when varying the tropopause height in other simulations (a slightly different strategy, but with similar impact). It is good to warn the reader that if you lower the vortex too much, the distortion in T_eq could be a real issue. [My worry here is that you have a nice parameter knob in the code, but people could get into trouble if they use it as a black box.]

22:8-9 The transition is a bit awkward here. You first emphasize that other aspects of the integrations are different. But then you appear to talk about the simulations "also" exhibiting the

same behavior.

22:9 Might be good to refer the reader back to Figure 10c here.

22:17 in the case of the

23:2 hPa, respectively

23: first paragraph. I had trouble parsing the argument here, but I think the authors primarily mean to say that for these integraions, the surface jet latitude is very sensitive to the vortex strength (Fig. 12b) but the 500 hPa jet is not (Fig. 12a)

Fig 14 Caption

First line: \overline{u} at (missing space in caption, and perhaps you could put the bar to emphasize zonal mean)

Later on, you refer to "sudden polar vortex decelerations" and SSWs, presumably referring to the same events. Please consider a consistent nomenclature.

Figure 14 right panel and surrounding analysis. I think these results are very interesting, but the time period choices seem somewhat arbitrary, and I worry about the statistical significance of the result. As this is a very interesting result, it would be nice if you could take a number of SSW events and consistently average before, during, and after the event to obtain the profiles with more statistical confidence.

[I appreciate that the authors mean for this to be a case study, rather than a proper research study, but this is one of the most novel results of the paper, and would benefit from more rigor.]

24 The footnote isn't necessary here. You could just state that these were \delta_z =4.3 integrations.

25:3 Where are the mixing ratios anomalously high? Do you mean in the high latitudes, or in the tropics? (Or everywhere?)

25:23-24 I was a bit confused here. How can the integration have weaker upwelling in the lower stratosphere as well as stronger downwelling in the midlatitudes? I think the authors just need to be a bit more descriptive here. There is weaker upwelling in the tropics, and all the downwelling in the extratropics is pushed to the vortex edge.

Section 5.2. Please do consider saving these results for future publications, as I felt that they were underdeveloped. Here are a couple small comments.

28:2 investigated in several publications

Figure 16 and surrounding discussion. Are splitting / westward eddy shedding events associated with eastward eddy shedding events, as implied by this single case study, or was this just a fortitous coincidence.

33:23 and equation A9. Would this drag work if the user specified a different set of vertical levels, other than L90MA. It might be good to include a reference to L90MA here as well.

34:15 If the model is starting from a cold start, I'm curious if you really need to spin up the monsoon slowly? [This is a very minor point!]

Table B1 The parenthesis after * and # confused me at first. Consider omitting them.

Figures C1-3 Consider also a natural progression with the color scale, though with just 4 curves, it's not so hard for the reader to parse.

Fig C2 and C3, panels d. I'm curious what clips the vortex at 150 m/s? Was this just how you plotted the curves (in which case you could just prevent the curve from falling to zero), or does the model prevent it from going faster? I do appreciate that this is 540 kph, pretty far beyond what you'd ever expect to see in the real world! I wonder what would happen when the vortex closes in on the speed of sound? (Clearly not a discussion for this paper!)

---

## Author Response (AR2)

**Reply to second Reviewer comments on**
**"Extending the Modular Earth Submodel System (MESSy v2.55) model hierarchy:**
**The ECHAM/MESSy idealized (EMIL) model set-up" by Hella Garny et al.**

We thank all reviewers for their very valuable comments on our revised manuscript. The response to the individual comments by reviewer Ed Gerber, and changes made to the manuscript are detailed below.

**Reviewer Ed Gerber**
I appreciate the authors' efforts to respond to my comments, and those of the other reviewers, and feel that the manuscript has improved considerably. I'm also very glad that they were able to catch the coding bug. (If it is any consolation, on my very first paper I was double checking a calculation during the revision phase when I horrifyingly discovered that the radius of the Earth in my code was off by a factor of 10! Fortunately I was able to redo all the calculations and the key results were unaffected.)
My chief concern with the manuscript is that it is still very long, such that some of the very interesting results (particularly the new applications at the end) might get lost. I've tried to offer suggestions on how it could be shortened, and other typographical suggestions, but leave the final decisions to the authors.
Ed Gerber

Thank you for the positive evaluation of our changes to the manuscript, and for the additional very helpful suggestions.
We agree in that the paper is still very long, and fully understand that there is a lot to digest. The difficulty stems from the paper being on the one hand a model description paper, thus it includes (mainly in the first part) the description of implemented functions (Section 2) and the evaluation in terms of benchmark tests (Section 3). In the spirit of a model description paper, it further aims to give a taste of possible application that will be possible in the future, without necessarily presenting new results (in Section 5). On the other hand, we included the range of sensitivity experiments that do actually lead to interesting new results on dynamical regimes and coupling (in Section 4). While it could be a solution to move for example the model description / benchmark tests to the appendix, we feel that it should stay in the paper, because it is a model description paper (and the journal is GMD).
Therefore, we decided to leave all material in the paper, but restructured it slightly (see below), and in particular make sure that we point out that the different sections are stand alone parts of the paper, thus the reader does not necessarily have to go through all the sections in the given order, but can choose the Section that she/he is interested in.

Minor comments and suggestions

1) I think it would be easier to describe the log10 coding error in terms of the parameter \delta_z in equation A1. In coding the natural log as a base 10 log, the authors' effectively reduced the amplitude of \delta_z by a factor of ln(10)=2.3, so that instead of the default value of 10 K, it was just 4.3 K. I would still explain why this happened (i.e., that it was an error in the representation of the logarithm), but you could refer to the alternative integrations as \delta_z=4.3 and fit them easily in the Held-Suarez framework. To me, this gives me a more physical sense of error: it was a reduction of the tropical stratification by about 6 K, from 10 to 4 K
I think that it would be better to introduce this alternative configuration earlier in the manuscript. An appropriate location would be when discussing the Newtonian cooling in Section 2.1, at the bottom of section 6. Here you explain the HS94 and PK02 profiles, and could mention that (inadvertantly) the \delta_z parameter was varied in some of the integrations, revealing interesting behavior that will be discussed later in the manuscript.

Thank you for pointing this out, it is an excellent suggestion and we changed the description of the "sensitivity simulations" accordingly. We also moved the description and the Figure of the modified

equilibrium Temperature to Section 2.1, as you suggested.

2) This comment is about how the change in delta_z parameter may impacts the circulation. The authors indicated that it increases the equator-to-pole temperature gradient, but I think it's primary affect might be through the stratification of the tropics and subtropics. It is true that modifying this parameter does increase the equator-to-pole temperature gradient in the upper troposphere, but the effect is on the order of a few degrees, compared to parameter \delta_y which is 60K. As I understand, this parameter was designed to increase the stratification in the tropics and subtropics, mimicking the impact of moisture. (Moist convection drives the tropics to a state of constant moist potential temperature, which imparts a dry stratification which increases with moisture content, which follows the surface temperature).
The stratification in the tropics is important in allowing the Hadley Cell to transport energy (it increase the gross stability, and hence reduces the need to transport mass to transport a given amount of energy). It may also affect midlatitude eddies, as baroclinic instability is sensitive to the stratification, particularly in the subtropics. Reducing the stratification in the subtropics would favor baroclinic instability further equatorward, and so could help explain the shift in the jet.

Thank you also for pointing this mechanism out, we agree that this is very likely the more appropriate explanation for the equatorwards shifted jet in the "reduced delta_z" simulations. We decided to add a dedicated subsection at the end of Section 4 to discuss the effects of the different parameter variations in concert (see response to next comment), and we discuss the effect of stratification, as you pointed out, in this new subsection.

3) I think the paper reads better with the equations moved to the supplement but as I mentioned above, it is still very long. A number of early figures are needed just to compare against earlier results (Figs 3, 4, 6a). I do see that these are essential, but unfortunately they not very exciting, compared to the results at the end. My concern is that by the time the reader gets to the interesting new case studies in section 5, they might be rather exhausted.

Agreed, but as stated above, we feel that with this being a model description paper the evaluation is actually an essential part of the manuscript. We hope that clearer structuring and pointing out the different aims of the sections will avoid the exhaustion of the reader by making her/him pick the section of interest (and the evaluation might not be the first pick, unless someone is particularly interested in the model performance, e.g. a future user).

There's also a great deal of information on the parameter sweep experiments in Figs 5, 9-13, and C1-C3 (which are substantially referred to in the text). It's hard for me to pinpoint what might be less important, but I would urge the authors to consider trimming or consolidating plots as much as possible, as to highlight the results you want people to remember.
To help summarize/focus the paper, would you agree that these are the key variations of the model that you consider:
i) \delta_z (stratification)
ii) stationary waves (topography vs. thermal)
iii) sponge layer
Yes, plus the change in winter polar LS temperatures (p_Tw)

Of these, the sponge layer had the least impact on the dynamics of the stratosphere and troposphere. Perhaps this could be put in an Appendix, along with Figs 7 and 8), if you wanted to move the text faster. (This is not to say, however, that the exponential profile isn't an improvement.)
It is true that the impact of the sponge is weakest, and thus dynamically least interesting. However, for the same reasons detailed above we choose to leave this in the paper.

The two key control parameters that you vary are:
a) p_Tw (depth of the vortex)

b) \gamma (strength of the vortex)
... and delta_z.
These two knobs are not independent, although they do have different impacts on the UTLS region.
Absolutely.
In terms of the results, it seems that the model's tropospheric jet stream tends to fall into one of two states: a fairly stable jet located around 30 degrees or fairly stable jet located around 40 degrees. Increasing gamma or increasing p_Tw tends to pull the jet further poleward, towards the high latitude state. When a control parameter tends to shift the jet between these states, however, you can observe regime behavior with a more variable distribution. But if the jet start too stably in the 30 degree state (i.e., with reduced \delta_z) or too stably in the 40 degree state (i.e., with thermal topography), there is no transition and the control parameters have a limited impact on the jet.
In the stratosphere it seems that three states are possible. When the vortex is very weak, there is limited variability of the vortex. When it is increased to moderate levels, the vortex is highly variable (at least ina relative sense) and the temperature is far from the equilibrium state due to a strong overturning circulation. And finally, when the vortex becomes sufficiently strong, it tends to hover near the equilibrium state, with reduced variability.
These stratospheric transitions are not necessarily tied to the tropospheric jet stream transition (as was the case in Wang et al. 2012). This does make it trickier to explain all the behavior. But if the figures could be organized to highlight these results, you might find it easier to focus the paper and trim.

Thank you for the nice summary of the different states of tropospheric and stratospheric state. Given your comments, we re-structured the paper as follows:

Section 2: Model descriptions
- as before, but including the description of the "delta_z" sensitivity

Section 3: Model benchmark tests
- as before, but removed Section 3.3 (Strat-Trop coupling). The result of reproduction of the original PK02 study is now mentioned in subsection 3.2 (PK set-up), and the results on the sensitivity runs are put into Section 4. Thus, this section now is purely reproducing previous results as benchmark tests, and is marked accordingly.

Section 4: Sensitivity of coupled troposphere-stratosphere dynamics to modified set-ups
- includes now in addition a subsection on the delta_z experiments. Further,  the effects on troposphere-stratosphere dynamics of all the parameter variations are discussed in concert in a new discussion subsection (Sec. 4.4). Therewith, this section aims at readers that are interested in the dynamical sensitivities.

Section 5: Application examples
- as before. We decided to leave the section as is, keeping all the results in the paper despite them not being necessarily new, as this section rather aims to point towards possible future application examples rather than presenting ready new results (see also next comment).

4) I feel the case study in section 5.1 is ideal for highlighting why this new EMIL model will be so useful. To my knowledge, this examination of the impact of dynamics on photolysis would simply not be possible in other models. This was a very interesting section.
I felt the monsoon case in section 5.2, however, to be less developed. For example, I found myself wondering about the statistical robustness of the eddy shedding and splitting in Figure 17. More importantly, it seems that this sort of work could be done with any dry dynamical core with HS94 style forcing. I would suggest to consider saving this monsoon case to another paper, when you can highlight the troposphere-stratosphere tracer transport, a topic that would be much harder to explore with a traditional dry dynamical core model. Saving this example for a later paper would also permit you to reduce the appendix, cutting equations A12-16, which include description of the temporal parameters that are not used in this manuscript.

Agreed in that the results are not new, and neither directly highlight the additional capabilities of EMIL. However, firstly as pointed out above, this is really rather meant to provide an example of future applications. Secondly, as a model description paper it aims to document the development of the submodel, which now includes the possibility of inserting a localized heat source, rather than presenting new results. Hence, the documentation of this capability should be part of this paper. However, we made sure to include additional notes about the future use of this capability and development state of the analyses.

Very minor comments and typographical suggestions by page:line number

2:7 Maher et al. 2019 is now published.
Thanks, changed.
2:13 ...for testing the dynamical cores of atmospheric models, the…
Thanks, changed.
2:16 consider "all thermodynamical processes, e.g., radiation and convection, by a relaxation toward a prescribed equilibrium temperature profile, and the surface boundary layer by Rayleigh friction …
Thanks, changed.
3:18 changes on the transport of
Thanks, changed.
3:20 this question has received a lot of
Thanks, changed.
6:2 I believe i.e. should generally be followed by a comma, here and other places in the manscript
Thanks,  a comma is inserted after all "i.e." and "e.g." now.
6:5 consider change the last sentences to "We describe the submodel in the next subsection, and provide technical details of the model setup (namelist choices, etc.) and implementation in the supplement.
Thanks, changed.
6:14 consider removing the parentheses here.
Changed to ".. , as described in the Supplement."
6:16 you could cut this last sentence
Decided to keep as is.
6:25 Consider moving the sentence starting with "In section 4.2," up one sentence, to follow "winter hemisphere."
Moved the sentence to the end of the paragraph, which should serve the same purpose, but reversed (first describe implemented variation, then say what we did with it).
Figure 2 -- as I noted in the major comments, figure 6b might fit here, as a second panel. I appreciate that you would want to use a linear pressure, but otherwise it would put all the information on T_eq in one place.
Yes, we moved Fig. 6b as second panel to Fig. 2, but still kept the linear pressure axis – I'd think it is ok to combine a log and linear pressure axis plot in one Figure.
7:4 There are a lot of parentheses in this sentence, which with the numbers 1) 2) and 3) were hard to parse.
True, changed to a list formatting.
7:11 The function names imply the intent of these diabatic heating/cooling options (i.e., to simulate climate change, stationary waves, and a monsoonal circulation), but these goals aren't fully explained in the paragraph. It might be good to explain both what each option allows and how it has been used in the past. For example:
The tteh_cc_tropics option allows the user to apply a zonal mean heating tendency with a Gaussian shape in latitude and pressure, as detailed in equation A10. The default values are taken from Butler et al. (2010) to approximate the impact of global warming on the atmospheric circulation.
Note that I would not call this exponential decay, which I would associated with exp(-x), not exp(-x^2).

Good point, this paragraph is changed to three bullet points for the individual heating functions, and the goal of their application are stated for all three.
7:12 (diabatic heating and cooling)
True, changed.
8:13 ... specified for each simulation in Table B1 and figure captions.
This paragraph is moved to the start of Section 4 (because for Section 3, the length of simulations is more straight forward to describe now). There, the suggested change is implemented.
8:13 Consider rephrasing the sentence about it being favorable to convergence of the climatology is reached. I appreciate that the authors had to optimize their use of computational resources to obtain reasonable confidence about their results. This is particularly challenging for integrations with regime behavior, and is reflected in large uncertainties in some of the integrations.
We changed the paragraph to the following (see beginning of Section 4):
"… To reduce the uncertainty of the results, it would be favorable to extend each simulation until convergence of the climatologies is reached. In particular for climate states with multiple dynamical regimes, this would, however, require very long integration times.
To reduce computational and data storage costs, we used the strategy of variable simulation length, i.e., we extended only a chosen set of simulations to test for the robustness of the results. In the shorter simulations, considerable uncertainty in the climatologies due to variability can be present. However, as shown in the following, the results are qualitative robust when comparing the short and long simulations. "
8:20-25 You could also compare your results to the plots in the original HS94 paper, or at least refer the reader to the paper, which show some of these statistics as well.
Good pointed, added: "Furthermore, the EMIL temperature and wind climatologies also compare well to the simulations shown in the original HS94 study (see their Fig. 1 and 2). "
8:21 It might be good to define the T63L19 notation here; I think it's the first time we've seen it. [You do define in the next paragraph.]
Done.
8:26 with a higher top
Done.
8:29 equatorward with T42L90MA resolution, and the eddy variance is generally reduced.
Done.
Figure 3: Consider using red-blue color scale for panels b,c. If panels e and f had the same color scale, it would give a better impression of the relative error.
Done.
8:30 What do you mean by "in a agreement with Wan et al. (2008)" Did they also show these effects of horizontal and/or vertical resolution?
Yes, they did, as is specified in the text now.
9:1 Could you briefly explain what you mean by "resolution of convergence"? In Gerber et al. 2008 (which does not need to be cited) we found that the aspect ratio of resolution appears to matter, so that some of these effects (e.g., the shift in the jet stream) still appear at high horizontal resolution if the vertical resolution is sufficiently high.
True, the result that convergence of climatology is reached with T85 in Wan et al. was obtained only for rather low vertical resolutions (by modern standard) of L19 and L31. Indeed, differences in T85 to T106 climatologies are reported to be larger for the higher vertical resolution L31. When fixing horizontal resolution to T85 and varying vertical resolution, they find convergence already at L31 – however, that is in a set-up with the upper level at 10 hPa, which might play a role.
As the resolution "convergence" likely is a more complicated matter than the text indicated, we decided to remove the statement on the resolution of convergence, and instead state more generally that the results might in general be dependent on the chosen resolution.
10:13 It might be good to direct the reader to the relevant details on the mountain, e.g., equation (1) of Gerber and Polvani 2009.
Good idea, done.
Figure 5 caption and legend. I was initially confused about the meaning of \mu here, thinking that it was some parameter. Consider defining it here and the first time it is used in the text. For the caption, you could say "the average windspeed at this location, denoted \mu, is given in the legend".

Added a more appropriate description to the caption.

12:12 Consider a paragraph break after (Butler et al. 2010).

No longer applies due to re-structured text.

13:3 free tropospheric jet (see Fig. C1).

No longer applies due to re-structured text.

13:9 (2009), however, the jet was shifted to higher latitudes in the simulation with a weak response. Thus, ….

No longer applies due to re-structured text.

13:21 What do you mean by non-physical? Gravity waves due end up providing a sink of momentum in the stratosphere, so there is some physical basis for this. But they do not behave like a Rayleigh friction, particularly when you start modifying the climate with parameter gamma, etc.. Shepherd and Shaw (2005) show why it's important that gravity wave parameterizations conserve momentum, but I think this is a finer point than you intended here.

True, the effects of GW is to decelerate the zonal wind, but they would not act as pure damping, as you wrote – in particular, they would not damp the whole wind fields irrespective of wave number, but maybe rather only the zonal mean wind I'd think a detailed discussion of those matters would go too far here, but to make the point more clear, the text is changed to:
"The simplified manner of damping the entire horizontal wind fields introduces a non-physical sink of momentum, as not only the zonal mean wind, but also all waves are damped."

13:27 but generally increases exponentially with decreasing pressure, not quadratically.

Thanks, corrected.

Could Figure 6b be incorporated into Figure 2?

Yes, done.

14:1 set-ups naturally differ within

Thanks, corrected.

14:2 temperature also extend

Thanks, corrected.

14:3 There are also a lot of differences in the tropics, where you get jets. Are these well converged? Consider adding ... in particular the high latitudes and the tropics, where alternating jets form, reminiscent of Quasi-Biennial Oscillation winds albeit fixed in time.

Interestingly, the tropical winds show oscillations with periods of about 5 years or so. Thus, in the shorter simulations, they are not converged, and we added a statement on the tropical winds.

14:11 since for both a flat surface and

Thanks, corrected.

Figure 8. The stippling dots were rather large here! (I don't mean too be picky, but you could describe these as polka dots!)

True, changed.

15:5 This is the first time you introduce the SPARC climatology, but it was used in Figure 8. You could just move this up to the preceding paragraph.

Done.

15:14 I'm not sure what is meant by unrealistic "step". I think the main problem is that the PK02 damped the winds to the climatology, but as the authors describe in other places in the manuscript, the climatology is warmer than radiative equilibrium in the high latitudes due to the overturning circulation (in the TEM framework, or equivalently due to meridional heat fluxes in the traditional, Eulerian framework).

True, the source of the problem are the equilibrium temperatures that are too warm here, but also the uniform equilibrium temperatures over the whole UTLS region are likely not realistic (as evident for example from Fig. 5 of Jucker et al., 2013 – indeed this Figure nicely shows that the lowered p_Tw is indeed more realistic).

17:17 I had trouble parsing this paragraph. I've made some suggestions below, but consider reworking it, as it is highlighting an interesting result.

17:19 polar vortex increases nonlinearly with increasing \gamma. In line ... at 10 hPa a the polar vortex accelerates more strongly with when \gamma exceeds a critical value (...

17:22 the polar stratosphere, i.e., change in parameter \gamma, the polar vortex...

Paragraph is reworked to hopefully make it easier to read.

17:28 This is an interesting paragraph, but I think you might be able to explain the regime transition more clearly in the context of the literature. There are two regimes here: for a

moderate vortex, the winds are westerly, allowing Rossby wave propagation (Charney and Drazin 1961). This is allows waves to propagate up, breaking up the vortex and raising the temperature considerably above radiative equilibrium. When the vortex becomes two strong, however, stationary waves (or quasi-stationary waves) can no longer propagate fast enough to keep up with the flow, preventing waves from enter the vortex. Hence there's no dynamic tendency to offset the radiative forcing (at least until you get to the gravity wave drag layer much higher).

In my major comment above, I thought there might also be a third regime, when there is no vortex and easterlies prevent any wave propagation; this would a summer vortex situation, but again, you end up pretty close to equilibrium for the lack of wave driving circulation.

Yes, this is very likely the backbone of the explanation of why we find the regime behavior of the vortex. To verify this, we would have to analyse the wave fluxes in greater detail, but I'd think we agree in that we do not want to extend the current study.

I'd say the interesting bit here is the sudden transition between the weak and strong state – while the polar cooling can be compensated by enhanced wave driving for weak increases of gamma (so apparently the conditions for wave propagation/ forcing improve), at some point the wave driving can't keep up with the cooling and the vortex strengthens suddenly, both because of the gamma change and the sudden "shut down" of wave propagation (or rather, breaking).

A paragraph on this line of argumentation was added to the new Discussion Section 4.4.

Figs 10, 11, 12. It would aid the reader to adopt a natural progression in the color scale, say a rainbow from red to purple as you transition along a parameter. That is, ptw=100 could be red, ptw=250, ptw=450 would be purple. When you have multiple configurations (ln vs. ln10) would be to use dashes for one and solid lines for the other.

Done for new Fig. 8 (combined former Fig. 10 and 11), but not possible for former Fig. 12 (new Fig. 11), as here multiple set-ups rather than a progression in a parameter are shown.

21:3 In general I find that footnotes overemphasize the point, as the reader stops and moves down to figure it out.

You might be right, but in this case I'd say that the footnote is a good place for this kind of information.

I have actually worried about this issue when varying the tropopause height in other simulations (a slightly different strategy, but with similar impact). It is good to warn the reader that if you lower the vortex too much, the distortion in T_eq could be a real issue. [My worry here is that you have a nice parameter knob in the code, but people could get into trouble if they use it as a black box.]

That is certainly true, but I'd say the issue of the "black box" applies to many parameters (in many kinds of models). I hope our analysis makes it rather clear that the climate state is sensitive to the way you set up your transition pressure, and that only paying attention to the mean polar temperature profile is problematic.

22:8-9 The transition is a bit awkward here. You first emphasize that other aspects of the integrations are different. But then you appear to talk about the simulations "also" exhibiting the same behavior.

True, and we reformulated the statements accordingly.

22:9 Might be good to refer the reader back to Figure 10c here.

Done.

22:17 in the case of the

Done.

23:2 hPa, respectively

Done.

23: first paragraph. I had trouble parsing the argument here, but I think the authors primarily mean to say that for these integraions, the surface jet latitude is very sensitive to the vortex strength (Fig. 12b) but the 500 hPa jet is not (Fig. 12a)

This text is moved to Sec. 4.4 and rephrased.

Fig 14 Caption

First line: \overline{u} at (missing space in caption, and perhaps you could put the bar to emphasize zonal mean)

Done.

Later on, you refer to "sudden polar vortex decelerations" and SSWs, presumably referring to the same events. Please consider a consistent nomenclature.

True, in the text we used the term "SSW", but in the caption "sudden decelerations" - this was because it makes more sense in this context to define the "central date" (i.e., the marker in Fig. 4) at the time of strongest deceleration rather than the zero-wind crossing, that appears quite a few days later. We added text to the caption, and when the Figure is first discussed, to make this clear.

Figure 14 right panel and surrounding analysis. I think these results are very interesting, but the time period choices seem somewhat arbitrary, and I worry about the statistical significance of the result. As this is a very interesting result, it would be nice if you could take a number of SSW events and consistently average before, during, and after the event to obtain the profiles with more statistical confidence.

[I appreciate that the authors mean for this to be a case study, rather than a proper research study, but this is one of the most novel results of the paper, and would benefit from more rigor.]

You are of course absolutely right, that this analysis could be done much more rigorous, for example by including composites of SSW events. We refer back to the beginning, where we said that the purpose of this Section is rather to highlight the possibilities with this model then to be a full study on their own. Hopefully, this will be the subject of a follow-on paper!

24 The footnote isn't necessary here. You could just state that these were \delta_z =4.3 integrations.

Done.

25:3 Where are the mixing ratios anomalously high? Do you mean in the high latitudes, or in the tropics? (Or everywhere?)

At high latitudes, added (and rephrased, because the mixing ratios are increased compared to the episodes directly following the SSW, but not compared to the "strong vortex" episodes).

25:23-24 I was a bit confused here. How can the integration have weaker upwelling in the lower stratosphere as well as stronger downwelling in the midlatitudes? I think the authors just need to be a bit more descriptive here. There is weaker upwelling in the tropics, and all the downwelling in the extratropics is pushed to the vortex edge.

Description is changed accordingly.

Section 5.2. Please do consider saving these results for future publications, as I felt that they were underdeveloped. Here are a couple small comments.

See statement to general comments.

28:2 investigated in several publications

Done.

Figure 16 and surrounding discussion. Are splitting / westward eddy shedding events associated with eastward eddy shedding events, as implied by this single case study, or was this just a fortitous coincidence.

The co-occurence of westward and eastward shedding events is indeed rather a coincidence, that we made use of to reduce the number of Figures. Added "Coincidentally" to make this clear.

33:23 and equation A9. Would this drag work if the user specified a different set of vertical levels, other than L90MA. It might be good to include a reference to L90MA here as well.

No, in the EMIL implementation this is done automatically. But admittedly, it would be better to have a more generalized formulation here. I appreciate that it would be hard to reproduce this without knowing the exact model/pressure levels, thus I added: "The equivalent pressure levels can be found in the data set that accompanies this publication (see data availability section)."

34:15 If the model is starting from a cold start, I'm curious if you really need to spin up the monsoon slowly? [This is a very minor point!]

Good question, probably not.

Table B1 The parenthesis after * and # confused me at first. Consider omitting them.

Done.

Figures C1-3 Consider also a natural progression with the color scale, though with just 4 curves, it's not so hard for the reader to parse.

We'd decide to leave it as is for simplicity.

Fig C2 and C3, panels d. I'm curious what clips the vortex at 150 m/s? Was this just how you plotted the curves (in which case you could just prevent the curve from falling to zero), or does the model prevent it from going faster? I do appreciate that this is 540 kph, pretty far beyond what you'd ever expect to see in the real world! I wonder what would happen when the vortex

closes in on the speed of sound? (Clearly not a discussion for this paper!)
This was due to a bad choice of the chosen intervals – it is fixed now, showing that wind speeds up to about 170 m/s are found (well below the speed of sound!)

[revised manuscript text omitted]